# Inhibition of oxygen dimerization by local symmetry tuning in Li-rich layered oxides for improved stability

Fanghua Ning[1], Biao Li [1], Jin Song[1], Yuxuan Zuo[1], Huaifang Shang[1], Zimeng Zhao[2], Zhen Yu[3], Wangsheng Chu[3✉], Kun Zhang[1], Guang Feng[1], Xiayan Wang [2✉] & Dingguo Xia[1,4✉]

Li-rich layered oxide cathode materials show high capacities in lithium-ion batteries owing to the contribution of the oxygen redox reaction. However, structural accommodation of this reaction usually results in O–O dimerization, leading to oxygen release and poor electrochemical performance. In this study, we propose a new structural response mechanism inhibiting O–O dimerization for the oxygen redox reaction by tuning the local symmetry around the oxygen ions. Compared with regular $Li_2RuO_3$, the structural response of the as-prepared local-symmetry-tuned $Li_2RuO_3$ to the oxygen redox reaction involves the telescopic O–Ru–O configuration rather than O–O dimerization, which inhibits oxygen release, enabling significantly enhanced cycling stability and negligible voltage decay. This discovery of the new structural response mechanism for the oxygen redox reaction will provide a new scope for the strategy of enhancing the anionic redox stability, paving unexplored pathways toward further development of high capacity Li-rich layered oxides.

[1] Beijing Key Laboratory of Theory and Technology for Advanced Batteries Materials, College of Engineering, Peking University, Beijing 100871, People's Republic of China. [2] Department of Chemistry and Chemical Engineering, Beijing University of Technology, Beijing 100124, People's Republic of China. [3] National Synchrotron Radiation Laboratory, University of Science and Technology of China, Hefei, Anhui 230026, People's Republic of China. [4] Beijing Innovation Center for Engineering Science and Advanced Technology, Peking University, Beijing 100871, People's Republic of China. ✉email: chuws@ustc.edu.cn; xiayanwang@bjut.edu.cn; dgxia@pku.edu.cn

The development of energy storage devices for portable electronics, electric vehicles, and large-scale renewable energy requires lithium-ion batteries (LIBs) with high energy density, long lives, and high safety[1–4]. Cathode materials are considered to be the bottleneck in improving the electrochemical performance of LIBs[5]. Compared with commercial cathode materials, Li-rich layered oxides deliver high discharge capacities of more than 250 mAh g$^{-1}$ owing to the involvement of the oxygen redox reaction[6–9]. Thus, these materials have attracted considerable global interest as important cathode material candidates for next-generation high-energy-density LIBs[10,11].

However, the oxygen redox reaction in Li-rich layered oxides usually results in a structural response involving O–O dimerization $(2O^{2-} \rightarrow O_2{}^{n-})$[12–14]. As a result, $O_2$ release and the migration of transition metal (TM) ions occur during charge–discharge[15–18], rendering a low cycling stability, voltage decay, and safety concerns for high-energy-density LIBs[19–23]. These drawbacks have hindered the commercial development of Li-rich layered oxide cathode materials. To overcome these problems, many approaches[24–27], such as bulk doping[28–30] and surface coating[31–34], have been investigated to improve the cycling performance by suppressing oxygen loss. Although considerable achievements have been made, to meet practical application requirements, further investigations of the mechanism of electrochemical performance evolution and novel strategies for enhancing the electrochemical performance are still required.

In this regard, Tarascon et al.[35] reported that the $d$–$sp$ hybridization associated with the reductive coupling mechanism results in good cycling behavior in $Li_2Ru_{0.75}Sn_{0.25}O_3$ materials. Ceder et al.[36] found that local structural defects can promote metal–oxygen decoordination, which stabilizes anionic redox reactions in the $Li_{2-x}Ir_{1-y}Sn_yO_3$ model system. Zhou et al.[13] demonstrated that a $Li_2Ni_{1/3}Ru_{2/3}O_3$ cathode in the Fd-3m space group has more O–TM percolation networks and shows good cycling performance.

To date, such strategies for enhancing the performance of Li-rich layered oxides have focused on stabilizing the O–O dimer to suppress oxygen release. However, as O–O dimerization is enhanced at increased capacities, oxygen release will always occur when the capacity provided by the oxygen redox reaction is high enough. Therefore, it is necessary to explore new structural response modes to the oxygen redox reaction other than O–O dimerization to enhance the inherent stability of the oxygen redox reaction in Li-rich layered oxide cathodes.

Herein, we propose a new structural response mechanism inhibiting O–O dimerization for the oxygen redox reaction by tuning the local symmetry around the oxygen ions in the Li-rich layered oxide. Using $Li_2RuO_3$ as a model Li-rich layered oxide cathode material, we prepare a local-symmetry-tuned $Li_2RuO_3$ cathode by disordering the TM/Li arrangement in the TM layer, which is defined as intralayer disordered (ID)-$Li_2RuO_3$. The local-symmetry-tuned material demonstrate significantly enhanced cycling stability and negligible voltage decay compared with regular (R)-$Li_2RuO_3$. Density functional theory (DFT) calculations show that the oxygen redox reaction in the local-symmetry-tuned ID-$Li_2RuO_3$ exhibits a structural response of telescopic O–Ru–O configurations without O–O dimerization. Gas analysis by in situ differential electrochemistry mass spectrometry (DEMS) show that no oxygen is released from the local-symmetry-tuned ID-$Li_2RuO_3$ cathode during the charge process. This novel structural response mechanism for the oxygen redox reaction based on local symmetry tuning without O–O dimerization can significantly enhance the cycling stability of high-capacity Li-rich layered oxides, which provides new scope for developing high-capacity cathode materials for LIBs.

## Results

**Prediction of O–O dimerization suppressed by symmetry tuning.** Figure 1a shows the honeycomb arrangement of cations in the $[Li_{1/3}TM_{2/3}]O_2$ slab of a regular Li-rich layered oxide (R-$Li_2TMO_3$), within which there are two oxygen-centered octahedrons in axial symmetry with respect to the O–O axis, as shown schematically in Fig. 1b. When oxygen participates in the charge compensation during delithiation, O ions inevitably approach Ru ions along the direction of the O–O axis owing to the local symmetry around oxygen, resulting in O–O dimerization and subsequent $O_2$ release. This loss of oxygen leads to poor cycling stability, as reported in many previous studies[19–23].

As the O–O dimerization response hinges on the local symmetry around oxygen, we imagine that the stability of the

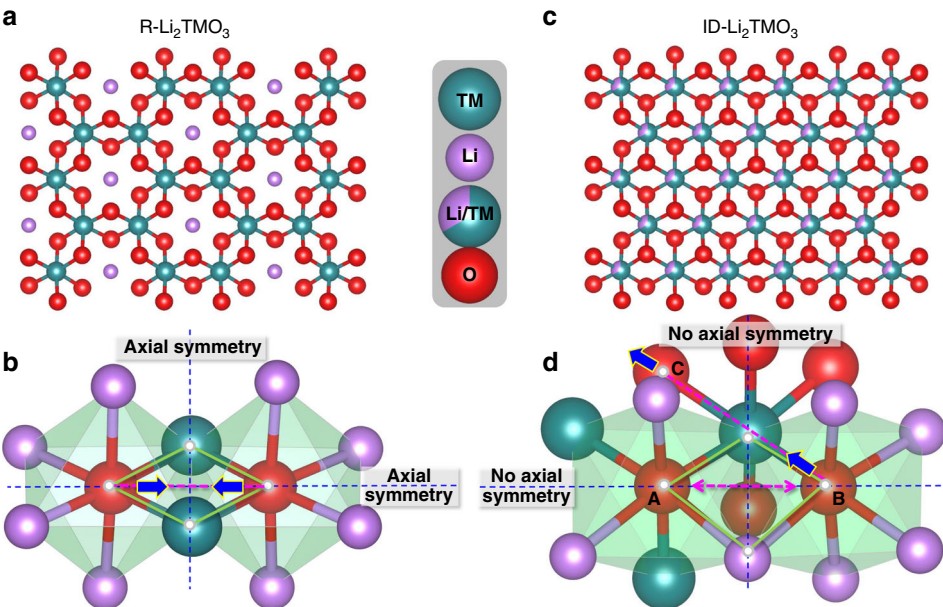

**Fig. 1 Layered structures and structural response modes. a, b** Crystal structure (**a**) and structural response mode (**b**) of R-$Li_2TMO_3$ to the oxygen redox reaction. **c, d** Crystal structure (**c**) and of structural response mode (**d**) of ID-$Li_2TMO_3$ to the oxygen redox reaction during delithiation.

oxygen redox process can be enhanced intrinsically by tuning this symmetry. Based on this consideration, we constructed a $Li_2TMO_3$ material with a disordered Ru/Li distribution in the transition metal layer (i.e., intralayer disordered (ID)-$Li_2TMO_3$) to break the local symmetry around oxygen, as shown in Fig. 1c, while keeping all other factors, such as the type of cationic ions and anionic ions, unchanged. The two oxygen-centered octahedrons in ID-$Li_2TMO_3$, in which the axial symmetry is broken, are shown schematically in Fig. 1d. Unlike the O–O dimerization process during the oxygen redox reaction for R-$Li_2TMO_3$, the structural response of ID-$Li_2TMO_3$ to the oxygen redox reaction is not constrained along the direction of the O–O axis during delithiation as the local axial symmetry is broken, thus O–O dimerization may be suppressed. Further, in the ID-$Li_2TMO_3$ system, oxygen ions with different coordination environments could be oxidized to different extents. As is shown in Fig. 1d, there are three kinds of octahedrally coordinated oxygen ions: $O_{center}[Ru_3Li_3]$ in oxygen site A, $O_{center}[Ru_1Li_5]$ in oxygen site B, and $O_{center}[Ru_2Li_4]$ in oxygen site C. As the octahedral with $O_{center}[Ru_1Li_5]$ coordination has two Li–O–Li configurations, whereas the octahedral with $O_{center}[Ru_2Li_4]$ and $O_{center}[Ru_3Li_3]$ coordination have only one and no Li–O–Li configuration, respectively, the oxygen ion in site B should be more easily oxidized than that in site A or C. Thus, the structural response to charge compensation of the $TM-O_B$ bond will be larger than that of the $TM-O_A$ bond or the $TM-O_C$ bond. Considering that the TM–O bond energy (ionic bond) is usually much larger than that of an O–O bond[37], O ions are expected to approach the TM ions along the O–TM–O bond direction to accommodate the oxygen redox reaction.

Based on these analyses, Li-rich layered $Li_2RuO_3$ was chosen as a model material to investigate the effect of the local symmetry around oxygen on the structural accommodation mode for the oxygen redox reaction. The single type of TM atom in this material and one-electron valence change during the oxygen redox reaction make $Li_2RuO_3$ convenient for tracking geometric and electronic structural changes. Fig. 2a and b show the optimized structures before and after lithium removal from R-$Li_2RuO_3$ and ID-$Li_2RuO_3$, respectively. The final structures for R-$Li_{2-x}RuO_3$ and ID-$Li_{2-x}RuO_3$ ($x = 0, 0.5, 1, 1.5, 1.75, 2$) are shown in Supplementary Figs. 1 and 2, which were tested to be the lowest energy structures among the multiple Li ordering (Supplementary Fig. 3, Supplementary Table 1–3). All the Ru–O bond lengths decrease and O–O dimerization occurs following the delithiation of R-$Li_2RuO_3$, as previously reported[38]. However, for ID-$Li_2RuO_3$, a very interesting telescopic O–Ru–O configuration is observed in the fully delithiated state. The lengths of some Ru–O bonds increase, whereas the lengths of other Ru–O bonds decrease. As for the short Ru–O bonds, the crystal orbital overlap population (COOP) analysis was performed to study the interaction between Ru and O, as shown in Supplementary Fig. 4. The integrated COOP of the short Ru–O bonds in ID-$Li_0RuO_3$ below Fermi level increases by 51% when compared with Ru–O bonds in R-$Li_0RuO_3$, implying that the net bond order of the short Ru–O bonds in ID-$Li_0RuO_3$ is higher than that of Ru–O bonds in R-$Li_0RuO_3$. Considering the higher net bond order and the bond length of 1.67 Å that is close to the previously reported bond lengths of $Ru^{5+}=O$ double bond (1.63 Å[39], 1.676 Å[40], 1.697 Å[40], and 1.70 Å[41]), this terminal Ru–O short bond can be regarded as quasi $Ru^{5+}=O$ double bond with a π-type hybridization between with Ru ($t_{2g}$) and O ($2p$). This is similar to the previous proposed Ir–O π bonds in $Li_2Ir_{1-x}Sn_xO_3$ system after TM ions migration to Li layer[36]. Further, the distance between the oxygen atoms involved in deep charge compensation is far greater than that in R-$Li_2RuO_3$, indicating that oxygen dimerization should be more difficult. As O–O dimerization

causes $O_2$ release, the prevention of O–O dimerization by the telescopic O–TM–O configuration in ID-$Li_2RuO_3$ may provide greater stability against oxygen release during deep delithiation than in the case of R-$Li_2RuO_3$. The enhancement of the oxygen stability was further confirmed by DFT calculations. The ΔG for oxygen release (defined in Supplementary Note 1, according to previous work[33]) with respected to the Li content is shown in Fig. 2c. The oxygen release energy for R-$Li_2RuO_3$ becomes negative after deep delithiation ($x > 1$ for $Li_{2-x}RuO_3$), which means that the oxygen is unstable and prone to release. Interestingly, the oxygen release energies for $O_{center}[Ru_1Li_5]$ coordination (green dashed line), $O_{center}[Ru_2Li_4]$ coordination (purple dashed line) and $O_{center}[Ru_3Li_3]$ coordination (blue dashed line) in ID-$Li_2RuO_3$ are all more positive than that for R-$Li_2RuO_3$ after deep delithiation, which is related to the total energy influenced by overall structural evolution of the systems, indicating that the oxygen is more stable in ID-$Li_2RuO_3$. The oxygen release energies are positive at all Li contents for $O_{center}[Ru_3Li_3]$ coordination. For $O_{center}[Ru_1Li_5]$ coordination, the oxygen release energies are also positive for $x < 1.75$ and close to zero for $x = 2.0$. Thus, oxygen release should be suppressed by the oxygen local symmetry breaking realized by TM/Li-intralayer disordering. In addition, since TM migration to Li layer would be promoted by oxygen release, the energy to form antisite defects of Ru in Li layer is calculated (Supplementary Fig. 5), which shows a much higher formation energy in ID-$Li_2RuO_3$ than in R-$Li_2RuO_3$. Thus, the Ru migration should be much more difficult in ID-$Li_2RuO_3$ than in R-$Li_2RuO_3$. In short, the structural response to the oxygen redox reaction in the R-$Li_2RuO_3$ system is O–O dimerization, whereas the oxygen redox reaction is structurally accommodated by the telescopic O–Ru–O configuration in ID-$Li_2RuO_3$ system. The telescopic O–TM–O configuration that inhibits O–O dimerization is a new structural accommodation mode for oxygen redox reactions, which would show good stability against oxygen release.

**Preparation and characterization of ID-$Li_2RuO_3$.** As $Na_2RuO_3$ shows a TM/Li-intralayer disordered characteristics[42], the ID-$Li_2RuO_3$ sample was prepared by Li/Na-ion exchange of $Na_2RuO_3$. The scanning electron microscopy (SEM) images of ID-$Li_2RuO_3$ and R-$Li_2RuO_3$ samples (Supplementary Fig. 6a, b) show that both samples consist of micrometer-scale particles. The X-ray diffraction (XRD) patterns of the $Na_2RuO_3$ and ID-$Li_2RuO_3$ samples are shown in Supplementary Fig. 7. The XRD pattern and refinement results of the as-prepared ID-$Li_2RuO_3$ sample are shown in Fig. 3a, Supplementary Tables 4 and 5. R-$Li_2RuO_3$ was also prepared for comparison, and the XRD pattern agrees well with that of regular $Li_2RuO_3$ with space group C2/m, as shown in Fig. 3b. Further, the refined crystallographic parameters and atomic coordinates of the R-$Li_2RuO_3$ sample are listed in Supplementary Tables 4 and 6, respectively. Unlike R-$Li_2RuO_3$, the ID-$Li_2RuO_3$ sample exhibit negligible superstructure reflection peaks (such as the peaks in the 2θ range of 20°–35°, highlighted in Fig. 3b), which suggests that TM/Li-intralayer disordering within TM layer exists in ID-$Li_2RuO_3$ sample. Specifically, according to refinement results of ID-$Li_2RuO_3$, the Ru and Li occupancies are 0.701515 (Ru) and 0.298485 (Li) at 4 h site, and 0.596268 (Ru) and 0.403732 (Li) at 2d site, which are close to 0.667 (Ru) and 0.333 (Li) of the Ru and Li occupancies at both 4 h and 2d site in the ideal TM/Li-intralayer disordered $Li_2RuO_3$. Thus, the structure of ID-$Li_2RuO_3$ sample was similar to the ideal intralayer disordered $Li_2RuO_3$. In order to evaluate the extent of intralayer disordering, two phase including regular $Li_2RuO_3$ and ideal intralayer disordered $Li_2RuO_3$ were used for refinement, which shows that the ratio of regular $Li_2RuO_3$ and

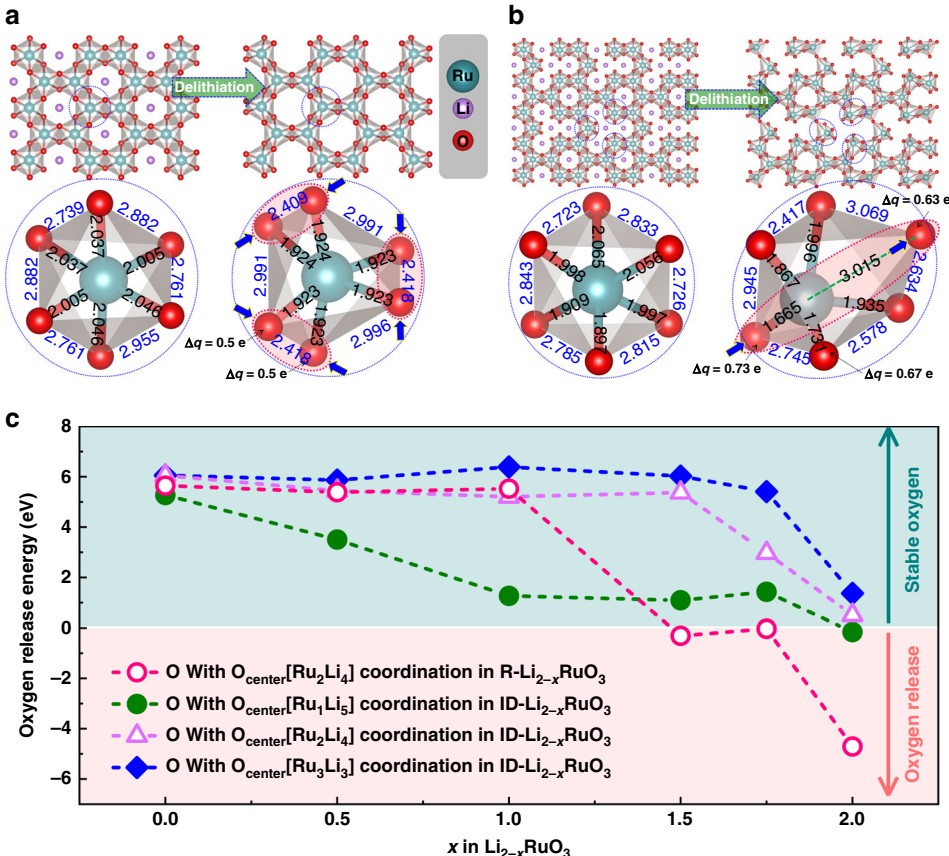

**Fig. 2 Crystal structures and oxygen stability upon delithiation. a, b** Optimized crystal structures and local $RuO_6$ octahedrons of $Li_2RuO_3$ and the corresponding delithiated state ($Li_0RuO_3$) for R-$Li_2RuO_3$ (**a**) and ID-$Li_2RuO_3$ (**b**). The values (in angstrom) on the local structures are the Ru–O bond lengths and O–O distances. **c** Oxygen release energy for R-$Li_{2-x}RuO_3$ and ID-$Li_{2-x}RuO_3$ systems.

idea intralayer disordered $Li_2RuO_3$ phases is about 35: 1. The percentage of the idea intralayer disordered $Li_2RuO_3$ phase is 97.1% (discussed in Supplementary Note 2), confirming that the ID-$Li_2RuO_3$ sample is almost the ideal intralayer disordered $Li_2RuO_3$ phase. Thus, the intralayer disordered $Li_2RuO_3$ was achieved successfully.

Furthermore, High-angle annular dark-field scanning transmission electron microscopy (HAADF-STEM) images of the as-prepared ID-$Li_2RuO_3$ sample were used to verify the TM/Li intralayer disorder in the transition metal layer on the atomic short-range scale (Fig. 3c, d). In these images, TM atoms appear as bright dots whereas oxygen and lithium atoms are nearly invisible. As shown in the HAADF-STEM image of ID-$Li_2RuO_3$ along the [100] zone axis (Fig. 3c), there are regular domains characterized by a periodic arrangement with one dark spot followed by two bright dots. Moreover, Li concentrated domains with continuous dark spots and Ru concentrated domains with continuous bright dots also exist, indicating TM/Li-intralayer disorder in the transition metal layer. The HAADF-STEM image of ID-$Li_2RuO_3$ sample along the [001] zone axis (Fig. 3d) also shows regular honeycomb domains, Li concentrated domains, and Ru concentrated domains. Thus, the HAADF-STEM images confirmed the disordered arrangement of the TM/Li intralayer on short-range scale in the as-prepared ID-$Li_2RuO_3$ sample. The observed and simulated selected area electron diffraction (SAED) patterns (Supplementary Fig. 8) were also given to analyze the structure on long-range scale. The ID-$Li_2RuO_3$ and R-$Li_2RuO_3$ structures with C2/m space group used for SAED simulation are taken from the XRD refinements. The observed SAED patterns of the as-prepared ID-$Li_2RuO_3$ sample shown in Supplementary

Fig. 8a, b that characterized with the marked weaker diffraction spots (red cycles) are consistent with the simulated SAED patterns of ID-$Li_2RuO_3$ structure model along [100] (Supplementary Fig. 8c) and [001] (Supplementary Fig. 8d) zone axes, respectively. Therefore, the intralayer disordering is verified by SAED patterns on long-range scale. Neutron powder diffraction (NPD) patterns were also obtained to further analyze the structural properties of the ID-$Li_2RuO_3$ sample. As shown in Supplementary Fig. 9, the results of NPD refinement (details are listed in Supplementary Tables 4 and 7) show Ru/Li-intralayer disordering, which is similar to XRD refinement. Hence, the TM/Li-intralayer disordered arrangement in the ID-$Li_2RuO_3$ sample was further confirmed by NPD results.

**Electrochemical performance of ID-$Li_2RuO_3$.** The electrochemical performance of the ID-$Li_2RuO_3$ was tested by galvanostatic charge−discharge in the voltage range of 2.0–4.8 V at a current density of 30 mA g$^{-1}$, as shown in Fig. 4a. It delivers a specific capacity of 230 mAh g$^{-1}$ in the first discharge, which is larger than the theoretical capacity of 164 mAh g$^{-1}$, estimated through the redox reaction of Ru$^{4+}$/Ru$^{5+}$. The voltage platform at ~ 4.55 V for the first charge may be related with the oxygen redox as reported from previous studies. The extra capacity could be assigned to the contribution of the oxygen redox. The charge−discharge curves of R-$Li_2RuO_3$ in the voltage range of 2.0–4.8 V at a current density of 30 mA g$^{-1}$ that agrees well with previous reports[43,44] were given for comparison (Fig. 4b), showing an initial specific discharge capacity of 289 mAh g$^{-1}$. The initial specific discharge capacity of ID-$Li_2RuO_3$ with average discharge voltage of 3.33 V is lower than that of R-$Li_2RuO_3$ with average

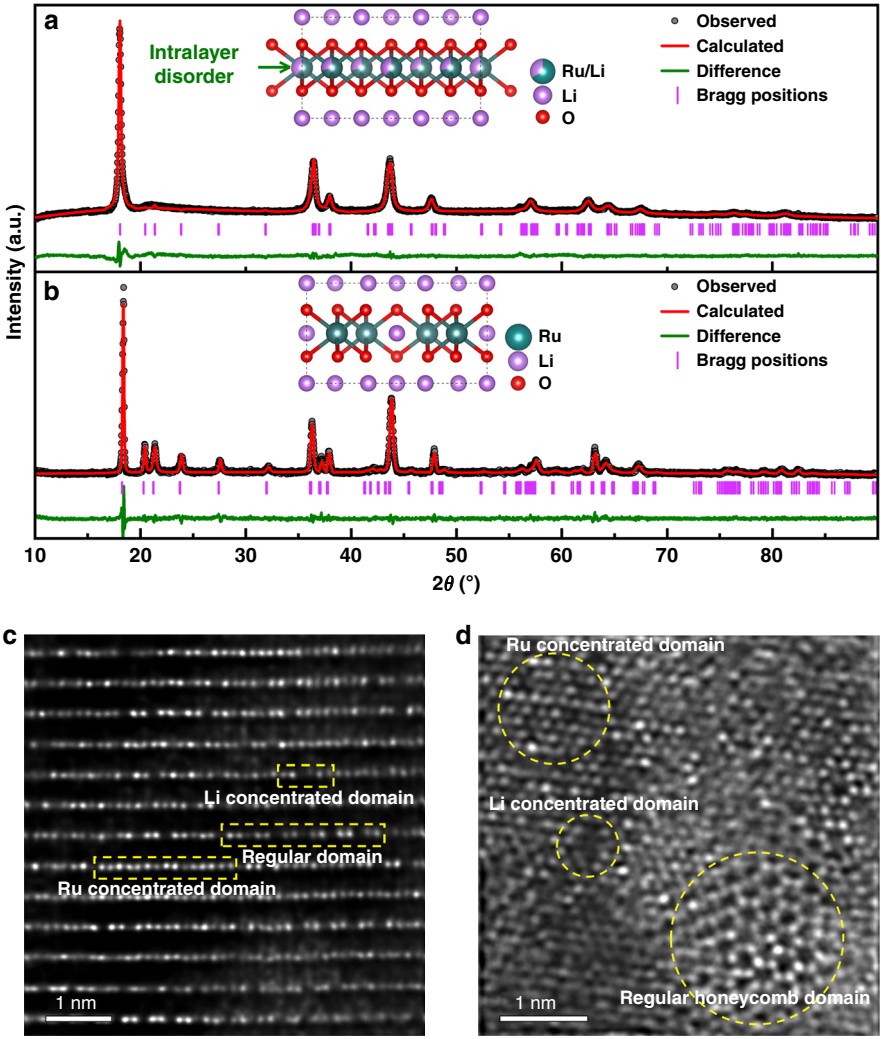

**Fig. 3 Structural characterization. a, b** XRD patterns of ID-Li$_2$RuO$_3$ (**a**) and R-Li$_2$RuO$_3$ (**b**). The insets show the corresponding crystal models after refinement. **c, d** HAADF-STEM images of the ID-Li$_2$RuO$_3$ sample along the [100] (**c**) and [001] (**d**) zone axes.

discharge voltage of 3.24 V within the same voltage range of 2.0–4.8 V, which can be explained by the higher voltage platform of ID-Li$_2$RuO$_3$. Indeed, the dQ/dV curves (Supplementary Fig. 10) indicate that charge and discharge voltage platform of ID-Li$_2$RuO$_3$ are both higher than that of R-Li$_2$RuO$_3$. Figure 4c compare the cycling performance of the ID-Li$_2$RuO$_3$ and R-Li$_2$RuO$_3$ electrodes. ID-Li$_2$RuO$_3$ demonstrates a discharge capacity of 221 mAh g$^{-1}$ with a capacity retention of 96% after 80 cycles, which are significantly higher than the 57 mAh g$^{-1}$ discharge capacity and 20% capacity retention of R-Li$_2$RuO$_3$. Furthermore, the cycling performance of the ID-Li$_2$RuO$_3$ and R-Li$_2$RuO$_3$ electrodes was also evaluated in different voltage ranges. As shown in Supplementary Fig. 11a, the capacity retention of ID-Li$_2$RuO$_3$ is significantly higher than that of R-Li$_2$RuO$_3$ in all cases, even when the initial specific discharge capacity of ID-Li$_2$RuO$_3$ (260 mAh g$^{-1}$ for 2.0–5.0 V) turns higher than that of R-Li$_2$RuO$_3$ (246 mAh g$^{-1}$ for 2.0–4.2 V). The relatively low capacity retention of R-Li$_2$RuO$_3$ is consistent with previous literature reports[44–47]. Thus, we conclude that the ID-Li$_2$RuO$_3$ electrode is more stable than the R-Li$_2$RuO$_3$ electrode upon cycling, as predicted above.

Furthermore, the voltage decay of ID-Li$_2$RuO$_3$ based on the midpoint discharge voltages is only 0.07 V after 80 cycles, which is much lower than that of 1.13 V for R-Li$_2$RuO$_3$, as shown in Fig. 4d. In addition, less voltage decay is observed for ID-Li$_2$RuO$_3$

than that for R-Li$_2$RuO$_3$ in several other voltage ranges (Supplementary Fig. 11b), even when the corresponding initial specific discharge capacity of ID- Li$_2$RuO$_3$ turns higher than that of R-Li$_2$RuO$_3$. That means the voltage decay in ID-Li$_2$RuO$_3$ is significantly suppressed.

The rate capability of ID-Li$_2$RuO$_3$ was estimated by progressive charging and discharging between the voltages of 2.0 V and 4.8 V in serial stages at various current rates from 0.1 C (30 mA g$^{-1}$) to 5 C (1500 mA g$^{-1}$), as shown in Fig. 4e. A capacity of 145 mAh g$^{-1}$ was maintained at 5 C, corresponding to 63.0% of the capacity at 0.1 C. As shown by the progressive charging and discharging test for R-Li$_2$RuO$_3$ in Fig. 4f, the capacity of 93 mAh g$^{-1}$ at 5 C was 31.7% of that at 0.1 C. Thus, although the rate capability of ID-Li$_2$RuO$_3$ is moderate, it is better than that of R-Li$_2$RuO$_3$. Furthermore, the capacity retention for the cycle at 0.1 C after the progressive charging and discharging tests were 100% and 78.8% in the ID-Li$_2$RuO$_3$ and R-Li$_2$RuO$_3$ systems, respectively, further confirming the excellent cycling stability of ID-Li$_2$RuO$_3$.

**Electronic structure changes.** Changes in the Ru oxidation state in ID-Li$_2$RuO$_3$ were determined by examining the ex situ X-ray absorption near edge structure (XANES) spectra of the Ru K-edge, as shown in Fig. 5a. The Ru K-edge continuously shifts to a higher energy below 4.3 V, indicating continuous oxidation of Ru, whereas the Ru K-edge remains unchanged when charging from

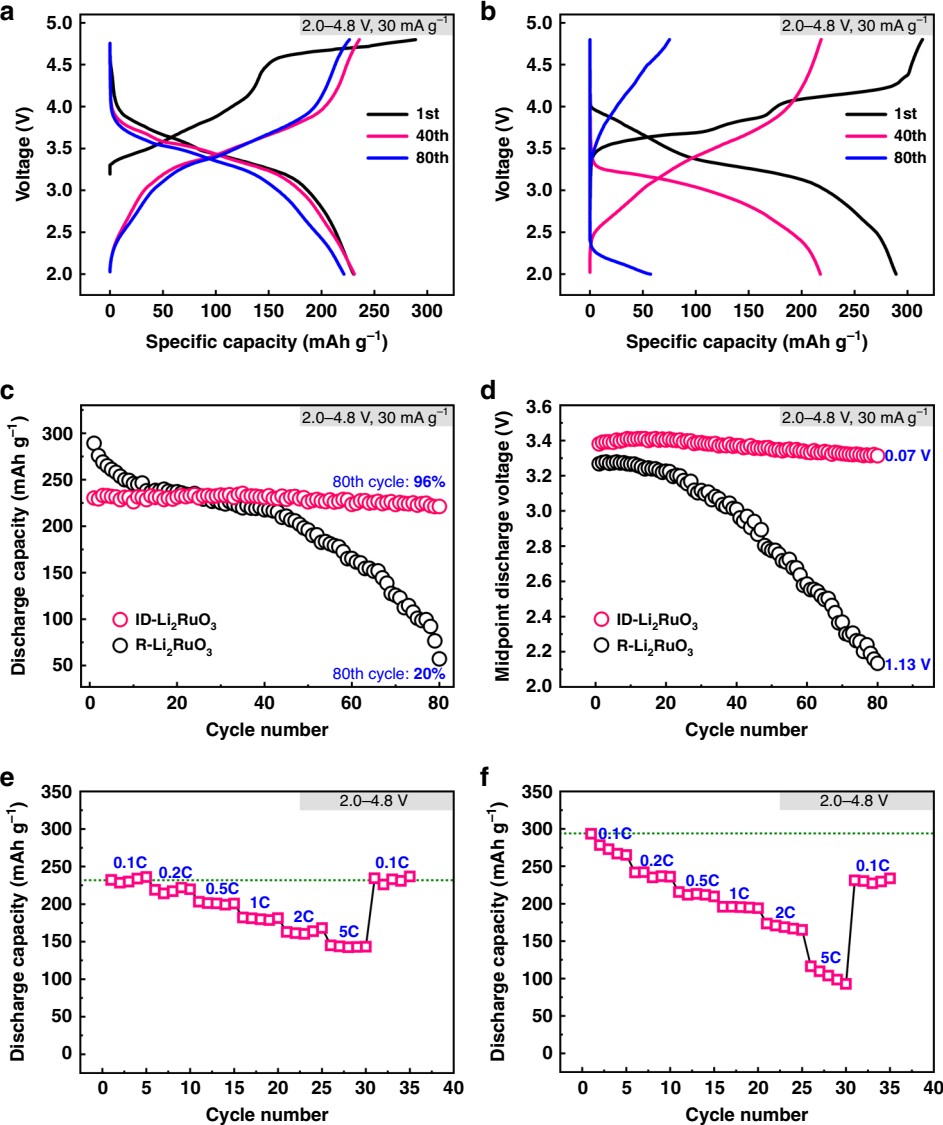

**Fig. 4 The comparative electrochemical performance. a**, **b** The charge−discharge profiles of ID-Li$_2$RuO$_3$ (**a**) and R-Li$_2$RuO$_3$ (**b**). **c** Cycling performance of ID-Li$_2$RuO$_3$ and R-Li$_2$RuO$_3$ in the voltage range of 2.0–4.8 V at a current density of 30 mA g$^{-1}$ (0.1 C). **d** Midpoint discharge voltages of the ID-Li$_2$RuO$_3$ and R-Li$_2$RuO$_3$ during cycling. **e**, **f** The progressive charging and discharging of the ID-Li$_2$RuO$_3$ (**e**) and R-Li$_2$RuO$_3$ (**f**) electrode in serial stages at various current rates from 0.1 C (30 mA g$^{-1}$) to 5 C (1500 mA g$^{-1}$) in the voltage range of 2.0–4.8 V.

4.3 V to 4.8 V. This behavior differs from the Ru K-edge XANES spectra of R-Li$_2$RuO$_3$ (Supplementary Fig. 12). R-Li$_2$RuO$_3$ presents a shift of absorption edge back to lower energy at the end charging (4.1–4.6 V), i.e., the reductive coupling mechanism (RCM), as reported previously for Li$_2$Ru$_{0.75}$Sn$_{0.25}$O$_3$ and regular Li$_2$RuO$_3$ material[35,38], which is known as a process where anionic redox is triggered that O ions are oxidized and structurally accommodated by O–O dimerization. However, for ID-Li$_2$RuO$_3$, the Ru K-edge shifts to a higher energy without shifting back during charging, showing the absence of RCM and thus O–O dimerization in ID-Li$_2$RuO$_3$. The O K-edge XANES spectra of ID-Li$_2$RuO$_3$ in Fig. 5b (more detailed results are shown in Supplementary Fig. 13) show a continuous increase in intensity of the first peak for the first and second charge processes, which corresponds to the hybridization of the 2p orbital of O and the 4d–t$_{2g}$ orbital of Ru. As no Ru oxidation occurred above ~ 4.3 V, this continuous increase in intensity of the O K-edge above ~ 4.3 V can be attributed to the anionic oxygen redox reaction. During the discharge process, the absorption edges in the Ru and O

K-edge XANES spectra show a gradual shift back to lower energies. Further, the evolution of both the Ru and O K-edges for the charge process in the second cycle is similar to that in the first cycle, confirming the reversibility of the Ru and O electronic structure changes.

First-principles calculations were conducted to reveal the origin of the excellent reversibility of ID-Li$_2$RuO$_3$ during delithiation. The charge variations on the Ru ions and O ions during the delithiaton processes for the R-Li$_2$RuO$_3$ and ID-Li$_2$RuO$_3$ systems obtained from Bader charge analysis are shown in Fig. 5c and d, respectively. The electronic structure variations during the delithiation processes for the R-Li$_2$RuO$_3$ and ID-Li$_2$RuO$_3$ systems were studied theoretically by comparing the density of states (DOS) for different Li contents (Li$_2$RuO$_3$, Li$_1$RuO$_3$, and Li$_0$RuO$_3$), as shown in Supplementary Fig. 14. Generally, the electronic structure variations are similar for R-Li$_2$RuO$_3$ and ID-Li$_2$RuO$_3$. The average charge on the Ru ions in Li$_{2-x}$RuO$_3$ decreases for $x < 1$, then remains almost unchanged for $x > 1$. The average charge on the O ions in Li$_{2-x}$RuO$_3$

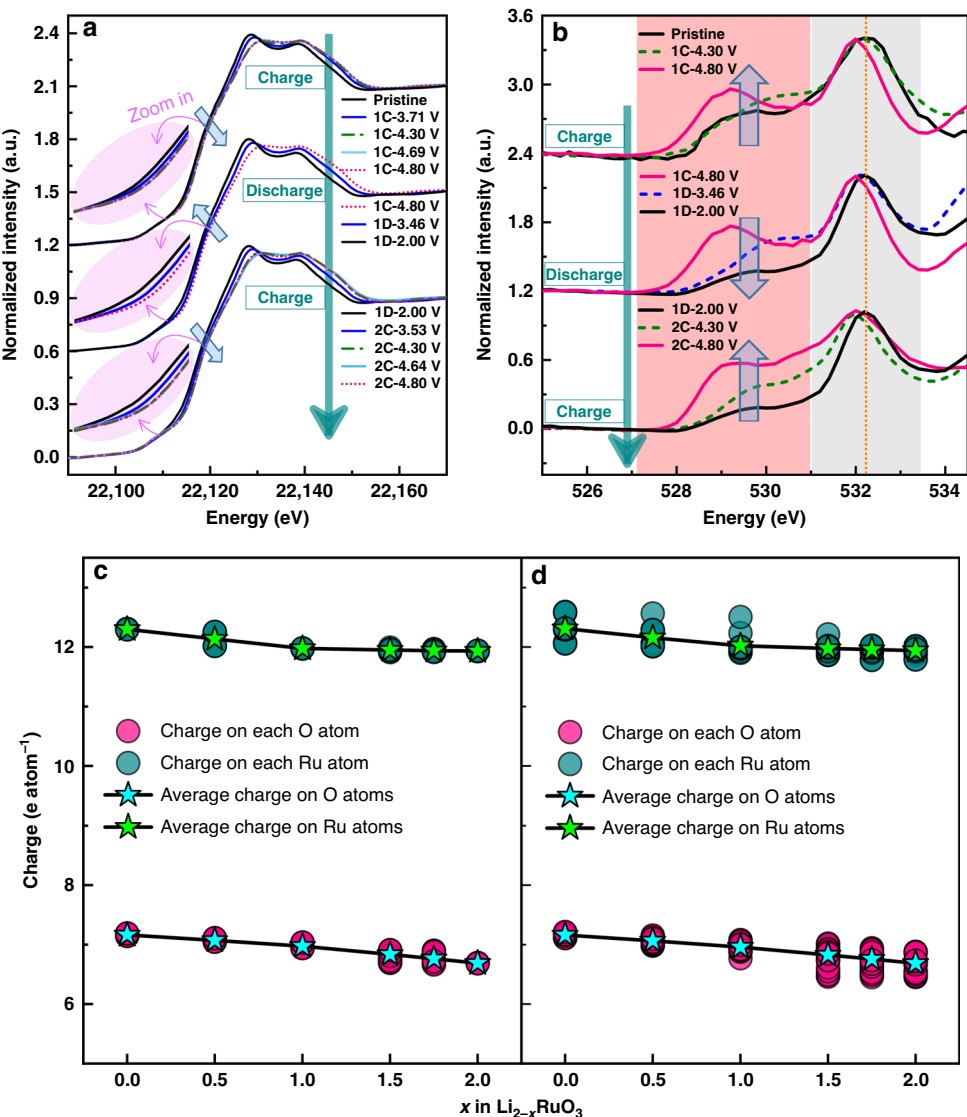

**Fig. 5 Electronic structure changes. a, b** Ex situ Ru K-edge (**a**) and O K-edge (**b**) XANES spectra of ID-Li$_2$RuO$_3$ upon charging and discharging. 1 C, 1D and 2 C represent the first charge, first discharge and second charge, respectively. **c, d** Charge and average charge on Ru ions and O ions in R-Li$_2$RuO$_3$ (**c**) and ID-Li$_2$RuO$_3$ (**d**) with respected to the Li content.

decreases with a higher slope for $x > 1$ than for $x < 1$. Based on the charge variation shown in Fig. 5c, d and the DOS variation shown in Supplementary Fig. 14, we conclude that Ru in Li$_{2-x}$RuO$_3$ mainly participates in charge compensation at $x < 1$, whereas charge compensation can mainly be attributed to the oxygen redox reaction at $x > 1$ in both the R-Li$_2$RuO$_3$ and ID-Li$_2$RuO$_3$ systems, which is consistent with the X-ray absorption spectroscopy (XAS) results. Furthermore, Bader charge analysis revealed the same magnitude of charge on all the oxygen atoms in the R-Li$_0$RuO$_3$ system (Supplementary Fig. 15a), whereas a nonuniform charge distribution was observed for the oxygen atoms in the ID-Li$_0$RuO$_3$ system (Supplementary Fig. 15b). This finding indicates that the extent of the oxygen redox reaction is homogeneous in R-Li$_2$RuO$_3$ but inhomogeneous in ID-Li$_2$RuO$_3$.

**Enhancement of oxygen redox stability**. An in situ XRD analysis was conducted to reveal the long-range structural evolution of ID-Li$_x$RuO$_3$ during the charge–discharge processes. The corresponding charge–discharge profile is given in Fig. 6a. The contour plot of the XRD patterns in the range of $2\theta = 16°–19°$ related to (001) peak is shown in Fig. 6b, where the diffraction intensity is

represented by the color depth. Figure 6c shows the XRD patterns from the direct observations. The peaks marked with stars are attributed to the beryllium X-ray input window of the in situ cell. Generally, the peak variations observed during cycling are reversible, indicating the good reversibility of the long-range structural evolution. The first charge process of ID-Li$_2$RuO$_3$ shows a two-phase transition feature for the (001) peak. However, for R-Li$_x$RuO$_3$, a continuous three-phase transition feature is observed for the (001) peak in the first charge process, as has been reported previously[38]. Combining with the charge–discharge curves, ID-Li$_x$RuO$_3$ shows two stages with a slope-like plateau (3.2–4.3 V) and a flat plateau (4.3–4.8 V), whereas R-Li$_x$RuO$_3$ shows three stages with relatively flat plateaus, which matches the phase transition revealed by in situ XRD. According to the refinement of XRD patterns of the 4.8 V charged ID-Li$_2$RuO$_3$, we find that ID-Li$_2$RuO$_3$ kept in C2/m phase with lattice parameter changed during delithiation, as shown in Supplementary Fig. 16, Supplementary Table 8 and 9. The $\beta$ was changed from 108.5870° to 90.0097°, indicating that the layered structure was altered from O3- to O1-type C2/m phase[12,36]. As shown clearly in Supplementary Fig. 17, the phase changed gradually from O3- to O1-

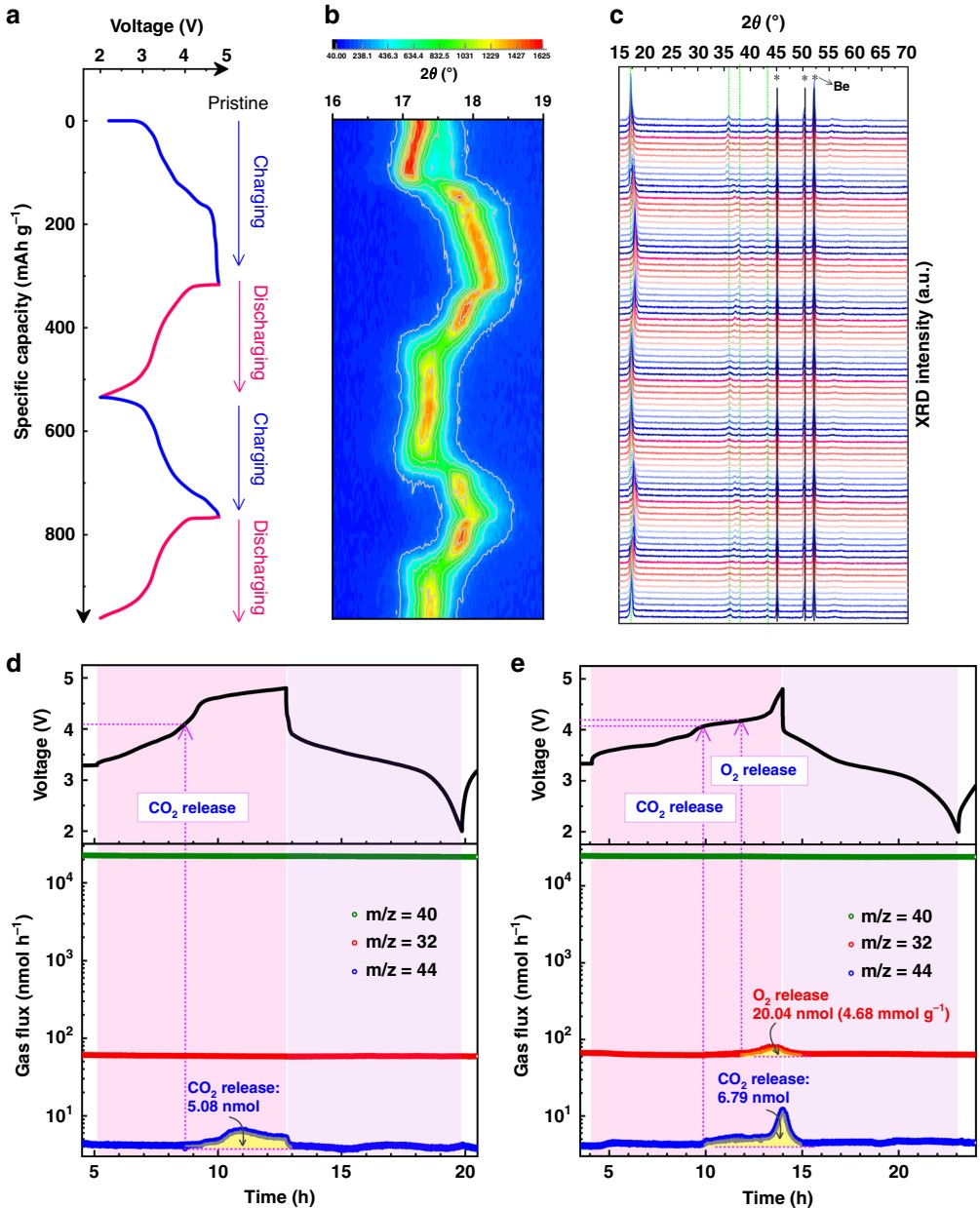

**Fig. 6 In situ XRD and in situ DEMS results. a** Voltage profiles used for in situ XRD analysis for ID-$Li_2RuO_3$ at a current density of 30 mA g$^{-1}$. **b** Contour plot of in situ XRD patterns in the range of $2\theta = 16°$–$19°$. The diffraction intensity is represented by the color depth. **c** in situ XRD patterns from the direct observations. The peaks marked with stars are attributed to the beryllium X-ray input window of the in situ cell. **d**, **e** Gas evolution at a current density of 30 mA g$^{-1}$ in the ID-$Li_2RuO_3$ (**d**) and R-$Li_2RuO_3$ (**e**) vs. Li cells from in situ DEMS analyses.

type structure during charge process, then almost returned back to O3-type structure of the pristine during discharge process. Hence, the long-range structure of ID-$Li_2RuO_3$ is reversible during charge and discharge processes. In addition, the migration of Ru to Li layer is almost absent according to the XRD refinement as the occupancies of Ru in Li layer are about 0.023% and 0.025% of the total Li site in Li layer for pristine and charged (4.8 V) ID-$Li_{2-x}RuO_3$, respectively, which is consistent with the results of the formation energy of Ru anti-site defects (Supplementary Fig. 5). In contrast, the R-$Li_2RuO_3$ undergoes an irreversible phase transition, as shown in Supplementary Fig. 18. The XRD patterns variation of our R-$Li_2RuO_3$ during charge and discharge processes are similar to the results that reported by Inaguma et al.[43] As revealed by Inaguma et al., the structure changed from C2/c phase to a mixed phase of R-3 and C2/c when charged to

3.8 V, then the structural transition with oxygen evolution occurs when further charged to 4.8 V, and the corresponding structure is unknown[43]. Similar to the reference[43], the structure of R-$Li_2RuO_3$ cannot be recovered to the pristine case during discharge processes. In short, the long-range structure of ID-$Li_2RuO_3$ is reversible during charge and discharge processes, in contrast to the irreversible processes of R-$Li_2RuO_3$, resulting in better cycling stability.

In situ DEMS measurements were carried out to evaluate the stability of oxygen, as is shown in Fig. 6d and e. The argon flux (carrier gas, $m/z = 40$) was stable, indicating that a stationary background was achieved. $CO_2$ ($m/z = 44$) release occurred once the charge voltage reached 4.1 V for both ID-$Li_2RuO_3$ (5.600 mg active material) and R-$Li_2RuO_3$ (4.356 mg active material) electrode assembled cell, corresponding to electrolyte

decomposition, which is similar to the DEMS results in previous reports[16,48–50]. More importantly, $O_2$ ($m/z = 32$) release from ID-$Li_2RuO_3$ was not detected, as is shown in Fig. 6d, which is in accordance with the reversible XRD evolution during charging/discharging. Thus, the local-symmetry-tuned ID-$Li_2RuO_3$ shows excellent cycling stability since oxygen release is avoided. However, evolution of $O_2$ from R-$Li_2RuO_3$ was observed during charging when the charge voltage approached ~ 4.2 V, as shown in Fig. 6e, which is consistent with the previous in situ DEMS result for R-$Li_2RuO_3$[16]. In addition, a sharp increase of $CO_2$ generation at ~ 4.3 V for R-$Li_2RuO_3$ was occurred as the electrolyte decomposition was promoted by $O_2$ that generated in the cell once $O_2$ evolution reached a certain high rate, as reported previously[48]. The $O_2$ release demonstrated here is in accordance with the irreversible XRD evolution of R-$Li_2RuO_3$ during charging/discharging. Thus, the R-$Li_2RuO_3$ exhibit poor cycling stability, especially when charged to higher voltage. In addition, the gas evolution for higher charge voltage (2.0–5.0 V) from an ID-$Li_2RuO_3$ electrode with 5.512 mg active material (higher than 4.356 mg in the case of R-$Li_2RuO_3$) was further evaluated by in situ DEMS (Supplementary Fig. 19). Notably, no oxygen release occurred, even at a high charge voltage of 5.0 V, confirming the absence of oxygen release from ID-$Li_2RuO_3$. Thus, the telescopic O–Ru–O configuration increases the cycling stability related to the oxygen redox reaction by suppressing oxygen release.

In order to reveal the structural evolution on the local-range scale, annular bright-field scanning transmission electron microscopy (ABF-STEM) image of 4.8 V charged ID-$Li_2RuO_3$ along [001] zone axis was obtained (Fig. 7a–c). It should be noted that the viewing direction is ascertained by the SAED and FFT patterns (Supplementary Fig. 20a, b), securing the reliability of such analysis. Based on the structure model of a O1-type layered structure with a space group of C2/m obtained from the XRD refinement of 4.8 V charged ID-$Li_{2-x}RuO_3$ as mentioned above, the theoretical SAED patterns are simulated (Supplementary Fig. 20c). The observed SAED (Supplementary Fig. 20a) and FFT Patterns (Supplementary Fig. 20b) are consistent well with the simulated SAED of this O1-type ID-$Li_xRuO_3$ along the [001] zone axis (Supplementary Fig. 20c). Thus, the [001] zone axis is confirmed. The theoretical atomic structure along the [001] zone axis is shown in Fig. 7d and e. Within the ABF-STEM image (Fig. 7a), Ru ions appear as dark black dots, and oxygen and lithium ions appear as light black dots. There are regular honeycomb domains, Li/vacancy concentrated domains, and Ru concentrated domains, as marked in Fig. 7a. If the structural response of the charged ID-$Li_2RuO_3$ behaves in a similar manner with the R-$Li_2TMO_3$, i.e., O–O dimerization which have been demonstrated by ABF-STEM image and Raman spectroscopy previously[12,14], we should observe it directly from the Ru–O arrangement along the [001] zone axis that is schematically presented in Fig. 7e, where the Ru–O bond are rotated slightly with six equal projected distances, with the O–O dimerization being nicely visualized. However, the ABF-STEM image of the charged ID-$Li_2RuO_3$ shows a very different projected Ru–O arrangement when compared with the R-$Li_2TMO_3$ case. The projected distances of the Ru–O bonds along b1, b2, and b3 directions (marked with white dotted arrows) were evaluated by the gray value of the ABF-STEM image, as shown in Fig. 7b (b1–b3). The corresponding projected Ru–O distances of the red hexagon marked $RuO_6$ are shown in Fig. 7c, where the two Ru–O projected distances along the b1 and b2 directions are not equal, and the two Ru–O projected distances along the b3 direction are equal. Therefore, the inhomogeneous Ru–O bonds with specific O–Ru–O configuration around the Ru ions are observed, in contrast to the homogeneous Ru–O bonds with O–O

dimerization that would take place in R-$Li_2RuO_3$. Thus, the telescopic O–Ru–O configuration of ID-$Li_2RuO_3$ was visualized by ABF-STEM image.

Raman analysis was also performed to confirm the structural response mode. The Raman spectra of the 4.8 V charged ID-$Li_2RuO_3$ and R-$Li_2RuO_3$ were obtained with excitation light of a He-Ne laser at 633 nm wavelength, as shown in Fig. 7f. The Raman stretch of O–O dimer $(O_2)^{n-}$ at 847 cm$^{-1}$ (in accordance with ~ 850 cm$^{-1}$ reported previously[14]) was observed in charged R-$Li_2RuO_3$ sample while not in charged ID-$Li_2RuO_3$. Hence, unlike the R-$Li_2RuO_3$, the O–O dimerization didn't occur in ID-$Li_2RuO_3$ during charge process, coinciding with our prediction from DFT calculation and Ru K-edge XANES spectra.

The magnitude of the Fourier transform of the $k^2$-weighted extended X-ray absorption fine structure (EXAFS) oscillations, $|\chi(R)|$, along with the fitting results of R-$Li_2RuO_3$ (Supplementary Fig. 21) and ID-$Li_2RuO_3$ (Supplementary Fig. 22) are both given for comparison. Based on the presence of two crests in the Ru K-edge XANES spectra shown in Fig. 5a, two group of Ru–O bonds were considered during fitting. The variation in the Ru–O shell from the fitting results of R-$Li_2RuO_3$ (Supplementary Fig. 21) is given in Supplementary Fig. 23a with the detailed values listed in Supplementary Table 10. The Ru–O bond length decreases during charge process then increased during discharge process. The total coordination number of the Ru–O bonds dramatically decreased when charged to high voltage (4.1–4.6 V). However, the total coordination number of the first Ru–O shell was not recovered to the pristine during the discharge process (Fig. 7g), indicating that the structural variation is irreversible during charge and discharge processes. This irreversible coordination number might be related to $O_2$ release during charging, which is consistent with the irreversible XRD and in situ DEMS results. In contrast, the fitting results of ID-$Li_2RuO_3$ show a reversible variation, as shown in Supplementary Figs. 22, 23b and Supplementary Table 11. Generally, the Ru–O bond length decreased during charging then increased during discharging. The coordination number of the long bonds dramatically decreased whereas that of the short bonds increased slightly during charging from 4.3 V to 4.8 V. We infer that a small portion of the long bonds was shortened and some long bonds were stretched to such an extent that the stretched bonds were no longer counted as part of the first Ru–O shell. Furthermore, as shown in Supplementary Fig. 23a, b, the difference between two group of Ru–O bond length is much larger than that in R-$Li_2RuO_3$, showing more inhomogeneous Ru–O bond lengths. Thus, the telescopic O–Ru–O configuration, including both shortened and stretched portions, occurs in response to the oxygen redox reaction during the charge process, which agrees well with the results of the DFT calculation, ABF-STEM image. The total coordination number of the first Ru–O shell was recovered during the discharge process (Fig. 7h), indicating that the telescopic O–Ru–O configuration is reversible. As is mentioned above, this structural response based on the reversible telescopic O–Ru–O configuration is responsible for the enhanced cycling stability of ID-$Li_2RuO_3$.

## Discussion

Based on all the above results, the theoretical prediction of local symmetry tuning as a strategy to achieve a structural response of telescopic O–TM–O configuration that avoiding oxygen dimerization upon charging/discharging is confirmed in a model Li-rich layered cathode material, $Li_2RuO_3$. In order to verify whether this telescopic O–TM–O mechanism works for the other cathode Li-rich layered cathode material related to first row TM, the $Li_2MnO_3$ system is investigated by DFT calculation. As shown in

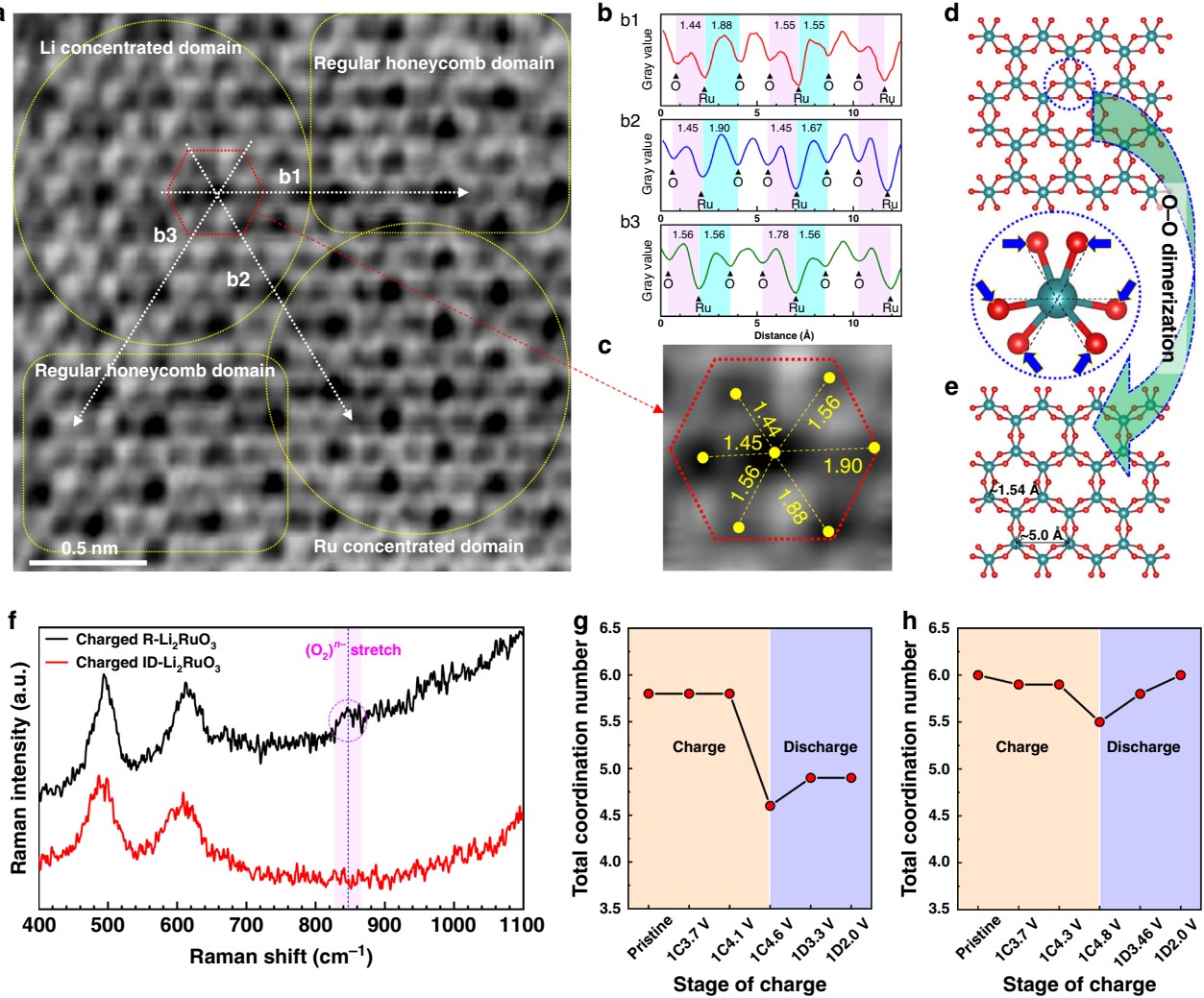

**Fig. 7 The local structural changes upon charging and discharging. a** The ABF-STEM image of 4.8 V charged ID-Li$_2$RuO$_3$ along [001] zone axis. **b** The gray value variation of ABF-STEM image along b1, b2, and b3 directions (marked with white dotted arrows). **c** The enlarged ABF-STEM image of the of the red hexagon marked RuO$_6$, the value between dark black dot and light black dot are the corresponding projected Ru–O distances. **d**, **e** The schematic Ru–O arrangement of Li$_{2-x}$RuO$_3$ before (**d**) and after (**e**) O–O dimerization. **f** The Raman spectra of 4.8 V charged ID-Li$_2$RuO$_3$ and R-Li$_2$RuO$_3$. **g**, **h** The variation of total coordination number of R-Li$_2$RuO$_3$ (**g**) and ID-Li$_2$RuO$_3$ (**h**) during charge and discharge processes, obtained from EXAFS fitting. 1C and 1D represent the first charge and discharge, respectively.

Supplementary Fig. 24, similar to the ID-Li$_2$RuO$_3$ system, the delithiated state of local symmetry tuned ID-Li$_2$MnO$_3$ also responds with telescopic O–Mn–O configurations. The O–TM–O configuration is related to short terminal TM–O bond which could also be stable for the first row TM including Ti, V, Cr, and Mn[51]. Thus, we preliminarily predict that the telescopic O–TM–O mechanism is also applicable for the first row light TM based Li-rich layered cathode materials. The structural response to oxygen redox would be alter from O–O dimerization to telescopic O–TM–O configuration when the local symmetry is tuned, avoiding O$_2$ release and thus enhancing the cycling stability of oxygen redox reaction involved charging/discharging processes in Li-rich layered cathode materials.

In conclusion, a new structural response mode other than O–O dimerization for the oxygen redox reaction was explored based on the local symmetry tuning around oxygen ions to suppress oxygen loss. ID-Li$_2$RuO$_3$ was synthesized, in which the local symmetry around the oxygen ions was tuned successfully via an TM/Li-intralayer disordered arrangement in the transition metal layer. Compared with R-Li$_2$RuO$_3$, the cycling stability and voltage stability of local-symmetry-tuned ID-Li$_2$RuO$_3$ was significantly

enhanced. EXAFS analyses and first-principles calculations indicated that the structural response to the oxygen redox reaction in local-symmetry-tuned Li$_2$RuO$_3$ involved a telescopic O–Ru–O configuration rather than O–O dimerization. DEMS analyses during the charge and discharge processes showed that no oxygen gas was released. This research highlights the importance of the local symmetry tuning in fabricating better Li-rich layered oxide cathode materials and provides a new structural accommodation mechanism to oxygen redox reaction for better cycling stability of Li-rich layered oxide cathode, which is expected to promote the practical application of such cathode materials in LIBs.

## Methods

**Sample preparation**. The Ru/Na-intralayer disordered (ID)-Na$_2$RuO$_3$ sample was synthesized via the solid-state route previously reported by Yamada et al.[42,52] The Na$_2$CO$_3$ and RuO$_2$ precursors were calcined at 900 °C for 10 h under an argon atmosphere. The ID-Li$_2$RuO$_3$ sample was obtained by Li/Na-ion exchange of the ID-Na$_2$RuO$_3$ sample in molten LiNO$_3$ at 280 °C for 4 h under argon atmosphere. The regular (R)-Li$_2$RuO$_3$ sample was synthesized via a solid-state route according to the literature[35,38,44]. RuO$_2$ and Li$_2$CO$_3$ (5% excess) were ground and mixed homogeneously, and the mixture was heated at 900 °C for 12 h in air, cooled, ground, and then heated at 1000 °C for 12 h in air.

**Materials characterization**. X-ray diffraction (XRD) patterns were collected using a Bruker D8 Advance diffractometer (Bruker, Germany) equipped with a Cu Kα radiation source ($\lambda = 1.5406$ Å) and operated at 40 kV and 40 mA. The R-$Li_2RuO_3$ and ID-$Li_2RuO_3$ spectra were recorded in the range of $2\theta = 10°–90°$ with a step of 0.02° and a constant counting time of 8 s. Neutron powder diffraction (NPD) measurements were performed on a time-of-flight general purpose powder diffractometer at the China Spallation Neutron Source (CSNS), Dongguan, China. The samples were loaded in 9.1 mm diameter vanadium cans and neutron diffraction patterns were recorded at room temperature. Rietveld refinements of the XRD and NPD patterns were performed using the GSAS software. The in situ XRD patterns were collected as the cell was slowly charged and discharged at a current density of 30 mA $g^{-1}$ to capture static or quasi-static structural evolution. The cathodes for the in situ XRD tests were prepared by mixing 80 wt% active materials, 10 wt% super-P as the conducting medium, and 10 wt% polytetrafluoroethylene as the binder, followed by rolling the mixture into a piece, and slicing into discs that fit into in situ XRD cell. The morphologies of the R-$Li_2RuO_3$ and ID-$Li_2RuO_3$ samples were characterized by cold-field emission scanning electron microscope (SEM, Hitachi S-4800). High-angle annular dark-field scanning transmission electron microscopy (HAADF-STEM) and annular bright-field scanning transmission electron microscopy (ABF-STEM) images were obtained using an aberration-corrected Jeol JEM-ARM200F Dual-X transmission electron microscope at an accelerating voltage of 200 kV at the Toray Research Center (Tokyo, Japan). The HAADF-STEM and ABF-STEM samples were prepared by Ar-ion milling, as the micrometer-scale particle sizes of our samples obstructed the measurements.

**Electrochemical measurements**. Each cathode was prepared by mixing 80 wt% active materials, 10 wt% super-P as the conductive additive, and 10 wt% polyvinylidene fluoride as the binder in N-methylpyrrolidone. Then, the obtained slurry was coated on Al foil and dried at 100 °C for at least 10 h. CR2032-type coin cells with Li metal as the anode and a Whatman GF/D glass microfiber filter as the separator were fabricated in a glove box at moisture and oxygen levels below 0.1 ppm. The proprietary high-voltage electrolyte was purchased from the Beijing Institute of Chemical Reagents. The galvanostatic charge/discharge tests were performed using a NEWARE tester (China) at room temperature.

**XAS**. The ex situ Ru K-edge XAS spectra were collected in transmission mode at beamline BL14W1 of the Shanghai Synchrotron Radiation Facility (SSRF), China, using a Si (311) double-crystal monochromator[53]. The ex situ O K-edge XAS spectra were collected in transmission mode at beamline BL10B of the National Synchrotron Radiation Laboratory (NSRL). The XAS samples were prepared by charging or discharging the electrode to the required voltage at a current density of 30 mA $g^{-1}$ and then transferring the electrode to the test station using an argon-filled bag for protection from atmospheric moisture and oxygen. All data processing performed prior to analysis, including energy calibration, background removal, normalization, and Fourier transformation, was performed using the Athena software. The first-shell extended X-ray absorption fine structure (EXAFS) fittings were performed using the Artemis program.

**In situ DEMS measurements**. The in situ DEMS system was constructed in-house based on the design reported by McCloskey et al.[54] A quadrupole mass spectrometer with a secondary electron multiplier (Hiden HPR-40 with Hiden HAL 201 RC) was used for mass spectra analysis. Measurements were performed using a Swagelok cell and argon as the carrier gas. The DEMS cells, which were prepared in an argon-filled glovebox (< 0.1 ppm of $H_2O$ and $O_2$), comprised Li foil as the negative electrode and the DEMS positive electrodes, separated by a polypropylene separator (Celgard). The DEMS measurement was started 4–5 h before the cell was operated to obtain a stable gas evolution background. The electrochemical measurements were carried out at a current density of 30 mA $g^{-1}$ for charge and discharge with a time interval of 8.5 s between each DEMS sequence.

**Computational details**. All the first-principles calculations in this work were performed in the Vienna ab initio simulation package (VASP) 5.4.4[55], which is based on a DFT framework. The projector augmented wave method[56] was used to expand the wave functions. The energy cutoff was set to 550 eV. The exchange-correlation functional was described using the spin-polarized Perdew–Burke–Ernzerhof (PBE) functional[57]. The strong correlation effect of the 4d state of the Ru ion was taken into account using the GGA + U method[58] with an effective U value of 4.0 eV, according to references[35]. The regular $Li_2RuO_3$ system was calculated with a cell containing 8 $Li_2RuO_3$ formula units (16 Li atoms, 8 Ru atoms, and 24 O atoms). The ID-$Li_2RuO_3$ system was modeled by exchanging Ru ion with Li ion within the Ru/Li layers in a cell containing 16 $Li_2RuO_3$ formula units. The Monkhorst–Pack scheme[59] with $5 \times 3 \times 3$ and $3 \times 3 \times 3$ k-point meshes was used for R-$Li_2RuO_3$ and ID-$Li_2RuO_3$, respectively. The total energies were converged to within $10^{-5}$ eV per formula unit. The final forces on all atoms were less than 0.02 eV $\text{Å}^{-1}$.

## Data availability

The data that support the findings of this study are available from the corresponding author upon reasonable request. Source data are provided with this paper.

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

## Acknowledgements

This work was financially supported by the International Science & Technology Cooperation of China under 2019YFE0100200, the National Key R & D Program of China (No. 2016YFB0100200), the National Natural Science Foundation of China (No. U1764255), and the Beijing Municipal Natural Science Foundation (No. 2181001). The first-principles calculations were supported by High-performance Computing Platform of Peking University. All support for our work is gratefully acknowledged.

## Author contributions

F.N. and D.X. conceived the idea. F.N., Z.Z., and B.L. carried out the sample synthesis, characterization and performance measurement. F.N. and B.L performed the DFT simulation and theoretical analyses. F.N and K.Z. carried out the in situ DEMS measurement. W.C., Z.Y, Y.Z and G.F. helped with the XAS measurements and discussion. J.S. and H.S. helped with the NPD measurements and discussion. The paper was written by F.N. and D.X.; D.X. and X.W. edited the paper. All authors contributed to discussing and revising the paper.

## Competing interests

The authors declare no competing interests.
