## [Peer Review File · Nature Communications]

Reviewers' Comments:

Reviewer #1:

Remarks to the Author:

In this work the authors investigate how the presence of disorder in model Li-ion cathode material Li_2RuO_3 affects the electrochemical performance and nature of anionic redox. Understanding the nature of anionic redox in Li-excess cathode materials is an important topic in the battery community as irreversible structural transitions and loss of oxygen from the structure have plagued the electrochemical performance of these potentially high capacity materials. In this study, the authors use two different synthetic routes to produce Li_2RuO_3 with a honeycomb ordered $\text{Li}_{1/3}\text{Ru}_{2/3}$ arrangement (R- Li_2RuO_3) and a metastable Li_2RuO_3 structure with a disordered Li and Ru arrangement (ID- Li_2RuO_3). The authors demonstrate that the capacity retention, voltage decay and rate performance of the ID- Li_2RuO_3 phase is superior to the R- Li_2RuO_3 phase, which they argue is related to the difference in the anionic redox mechanism at high states of charge. Using a combination of X-ray absorption spectroscopy (XAS), in-situ X-ray diffraction and density functional theory (DFT) calculations, they propose that the reversible anionic redox in the ID- Li_2RuO_3 structure comes from the presence of reversible 'telescopic O-Ru-O' interactions as opposed to the formation of O-O dimers in the R- Li_2RuO_3 phase, which leads to O_2 loss.

The majority of the work is carried out well and is presented clearly. The electrochemistry is clearly improved in the ID- Li_2RuO_3 material which adds an interesting piece to the puzzle of anionic redox in Li-excess cathodes. However, as outlined below, there are issues with the methodology and analysis that need to be addressed before this manuscript is suitable for publication.

Throughout the manuscript the authors argue that novel 'telescopic O-Ru-O' configurations with one short and one long Ru-O bond, as shown in Figure 2b, are responsible for the reversible anionic redox of the ID- Li_2RuO_3 structure. However, this idea is very similar to recent work by Hong et al. (reference 36), in which it was proposed that terminal Ir-O n bonds in the analogous system, $\text{Li}_2\text{Ir}_{1-x}\text{Sn}_x\text{O}_3$, were responsible for oxygen redox. From the structure in Figure 2b, it appears that disorder of the Li-Ru sites leads to similar vacancy configurations in the current system that were present in the previous work after Sn migration. The authors should discuss their findings in the context of this previous study and highlight whether their O-Ru-O environments are related to the formation of analogous Ru-O n bonds.

In Figure 2c, the authors highlight that the oxygen release energy is less favourable for the Ru_1Li_5 and Ru_3Li_3 configurations of the ID- Li_2RuO_3 structure than the Ru_2Li_4 environments in the R- Li_2RuO_3 structure, which leads to less O_2 loss in the former case. However, the authors do not include the oxygen release energy for the Ru_2Li_4 configuration of the ID- Li_2RuO_3 structure. From the schematic in Figure 2b and TEM in Figure 3, these environments Ru_2Li_4 should also be present in the ID- Li_2RuO_3 structure. The authors should show whether there is a difference in the driving force for oxygen evolution between the Ru_2Li_4 environments in the ID- Li_2RuO_3 and R- Li_2RuO_3 structures and if there is not, explain why the ID- Li_2RuO_3 structure shows considerably less O_2 evolution.

The authors use a combination of X-ray diffraction and neutron diffraction to refine the crystal structures of the ID- Li_2RuO_3 and R- Li_2RuO_3 phases, however, they do not provide details of the fitting procedure which are necessary to properly assess the data. The occupancy of the 4e Li sites are fixed at 1. Did the authors try and refine the occupancy of Ru on the 4e site, i.e. antisite disorder? In the power X-ray diffraction pattern of ID- Li_2RuO_3 in Figure S2, there is a broad bump between $20-30^\circ$ 2θ which is characteristic of superstructure ordering of the $\text{Li}_{1/3}\text{Ru}_{2/3}$ honeycomb layers. Was this region included in the refinement of the structures? Where the thermal parameters refined for each structure?

The structure of R-Li₂RuO₃ has been reported in a number of studies (Wang, J.C., Cao, G. et al, 2014. Physical Review B, 90(16), p.161110.) where it has been shown that neighbouring Ru ions form Ru-Ru dimers which form ordered arrangements that lower the symmetry from C2/c to P21/m or C2/m. The presence of Ru-Ru dimers has also been suggested to be linked to the decrease in the voltage and electrochemical performance in Ru doped Li₂MnO₃ (Knight, J.C., Manthiram, A., et al. 2015. Journal of Materials Chemistry A, 3(5), pp.2006-2011). Did the authors consider the P21/m or C2/m structures during their refinement? The authors should consider the presence of Ru-Ru dimers in their structure and how the disordered arrangement of Li and Ru may alter the long-range dimer structure and electrochemistry.

As highlighted in the introduction by the present authors, there is a link between cation migration and anionic redox in a number of Li-excess cathode materials. Did the authors calculate the energy to form Ru anti-site defects in the delithiated ID-Li₂-xRuO₃ and R-Li₂-xRuO₃ structures analogous to the calculations of Ir and Sn antisite disorder in reference 36?

The properties of a range of delithated Li₂-xRuO₃ structures are calculated with DFT calculations, but the authors do not give details about how the delithated structures were generated. Were multiple Li orderings considered for each structure? The authors should provide more details about the structures were generated and ideally include the lowest energy structures in the SI.

Reviewer #2:

Remarks to the Author:

The author reported a new structural response mechanism that may inhibit O-O dimerization and then improve cycling and voltage stabilities without oxygen release. Solving oxygen release is an important but challenging task for Li-rich cathode. This is a very important piece of this work and may impact on a wider field.

While there were some interesting findings and good improvements, such as suppressing O-O dimerization, the writing should be significantly improved, mainly in the quality of figures before publication. Here are a few issues to fix.

1) The telescopic O-TM-O configuration seems be something similar to the idea of electron hole used by Tarascon's, Bruce's and Ceder's groups. Please elaborate the conceptive novelty in the telescopic O-TM-O configuration with respect to or simply a realization of electron hole?

2) The quality of Figure 2 hinders understanding their discussions on oxygen release. They should increase all fonts in Figure 2 larger enough to render notes readable. Additional DFT data of the total energy should be provided for all the three kinds of oxygen sites at A, B, C in ID-Li₂TMO₃ systems and the original Li₂RuO₃ as well. In the calculation of oxygen release energy, clarify what reference have been used for delithiated ID-Li₂TMO₃ system and the original Li₂RuO₃.

3) I do not understand "The Rietveld refinement indicates that the long-range structure of ID-Li₂RuO₃ is consistent with that of ideal ID-Li₂RuO₃." On page 9. What is ideal ID-Li₂RuO₃ referring to? Because they used ionic exchange method to synthesize the Ru/Li mixing layer, clarify if there is any residual Na in the sample that makes the electrochemistry different?

4) The quality of Figure 3 makes their XRD and TEM results difficult to understand at this stage.

5) Figure 4 indicates that R-Li₂RuO₃ has significantly more discharge capacity 300 mAh g⁻¹ than does ID-Li₂RuO₃ at 230 mAh g⁻¹. Does this mean without oxygen dimerization, the repondance mechanism cannot achieve high capacity in ID-Li₂RuO₃? Were all the benefits in rate and voltages in the cost of high capacity? Suggest to turn the inset of charge/discharge profiles into a full figure 4a and compare both cycling plots in figure 4b. They should also mention what the initial charge capacities were for both samples.

6) The quality of Figure 5/6 makes it impossible to guess what oxidation occurs in Ru or O.

7) While DEMS measurements are valued techniques to detect gas generation, they cannot tell what gas generated by their own. So, they should be cautious to claim the telescopic O–Ru–O configuration to suppress oxygen release for two reasons. a) the actual effect of telescopic O–Ru–O configuration seemed suppress the dimer formation. If no oxygen dimer, there is no suppressing at all. b) the figure 6d and 6e were not exactly the same in the region of gassing. ID-Li₂RuO₃ showed higher increases after CO₂ detected at 4.1V. The charging curves were hardly comparable between 6d and 6e. So, need to clarify the difference in the gas generation. Both 6d and e indeed showed some gas generated once charging to high voltage. Be fairly enough, the 6d only showed reduced gassing in the end of charging comparing to 6e. But this cannot rule out whether it is oxygen or not without further analysis. The figure quality should be improved as well.

8) Provide some discussions or guidance how their proposed repondance mechanism would like to work for light elements other than heavy elements like Ru in practical cathode. Will such a mechanism only work for second row transitional metals? It seems the telescopic O–TM–O configuration unlikely stable for first row TM.

Reviewer #3:

Remarks to the Author:

This paper reports comparative studies on ordered and disordered Li₂RuO₃ as oxygen-redox cathode materials, which I don't recommend for publication. The main claim 'telescopic O-Ru-O' with short and long Ru-O bonds of 1.6 and 3.0 Å, which is against the classic but fundamental concept of 'ionic radius',

is just a speculation based on the hypothetical calculations and (subjective, in my opinion) fittings of EXAFS. I could not find any convincing experimental evidence for their hypothesis to deny the fundamental concept of inorganic chemistry. The followings are my serious concerns.

1. Concerning the DFT part, the authors found a specific local structure, that is, short and long Ru-O bonds in disordered Li₂RuO₃. I'm highly suspicious of such a chemically counterintuitive short and long Ru-O bonds of 1.6 and 3.0 Å, which are completely against the simple concept of ionic radius. I believe that it is necessary for the authors to re-consider the validity of the calculation models, especially for disordered one.

2. Figure 3: Please confirm the disordered and ordered cation arrangements of two materials using SAED rather than HAADF-STEM. More importantly, Figure 3d shows ordering of Li and Ru within the Ru layer, which is not consistent with the claim 'the intralayer disordered Li₂RuO₃ was achieved successfully.'

In this situation, Figure 3a (the XRD pattern for ID-Li₂RuO₃ shows no superstructure peaks) is not reasonable.

3. Please compare the charge-discharge curves of ID-Li₂RuO₃ and R-Li₂RuO₃ in Figure 4, rather than

only focusing on the cycle stability. I believe that the comparison of dQ/dV plots would also be of interest to readers.

4. P13 'This behavior differs from that for the reductive coupling mechanism induced by the formation of O–O dimers in R-Li₂RuO₃, meaning that O–O dimers (O₂n⁻) may not be formed in ID-Li₂RuO₃.' I'm not convinced by this speculation. Whether O-O dimer is formed or not is apparently the key question in this work. Therefore, it is mandatory for the authors to prove it by experiments.

5. Figure 6: Please compare the difference of the phase evolution of R-Li₂RuO₃ and ID-Li₂RuO₃, and explain why the differences occur. Then, the authors would be able to discuss the origin of the better cycle stability of ID-Li₂RuO₃. The present Figure 6 and relating part only report 'results'.

Responses to Referees' Comments

Manuscript title: Inhibition of oxygen dimerization by local symmetry tuning in Li-rich layered oxides for improved stability

Manuscript number: NCOMMS-20-10074A

Corresponding author name(s): Dingguo Xia

Reply to Reviewer 1

General Comment

In this work the authors investigate how the presence of disorder in model Li-ion cathode material Li_2RuO_3 affects the electrochemical performance and nature of anionic redox. Understanding the nature of anionic redox in Li-excess cathode materials is an important topic in the battery community as irreversible structural transitions and loss of oxygen from the structure have plagued the electrochemical performance of these potentially high capacity materials. In this study, the authors use two different synthetic routes to produce Li_2RuO_3 with a honeycomb ordered $\text{Li}_{1/3}\text{Ru}_{2/3}$ arrangement (R- Li_2RuO_3) and a metastable Li_2RuO_3 structure with a disordered Li and Ru arrangement (ID- Li_2RuO_3). The authors demonstrate that the capacity retention, voltage decay and rate performance of the ID- Li_2RuO_3 phase is superior to the R- Li_2RuO_3 phase, which they argue is related to the difference in the anionic redox mechanism at high states of charge. Using a combination of X-ray absorption spectroscopy (XAS), in-situ X-ray diffraction and density functional theory (DFT) calculations, they propose that the reversible anionic redox in the ID- Li_2RuO_3 structure comes from the presence of reversible 'telescopic O-Ru-O' interactions as opposed to the formation of O-O dimers in the R- Li_2RuO_3 phase, which leads to O_2 loss. The majority of the work is carried out well and is presented clearly. The electrochemistry is clearly improved in the ID- Li_2RuO_3 material which adds an interesting piece to the puzzle of anionic redox in Li-excess cathodes. However, as outlined below, there are issues with the methodology and analysis that need to be addressed before this manuscript is suitable for publication.

Response to the General Comment

Thanks for the referee's positive affirmation of our work. The comments are of great benefit to improve our work. We have revised the manuscript accordingly.

Comment 1

Throughout the manuscript the authors argue that novel ‘telescopic O–Ru–O’ configurations with one short and one long Ru–O bond, as shown in Figure 2b, are responsible for the reversible anionic redox of the ID-Li₂RuO₃ structure. However, this idea is very similar to recent work by Hong et al. (reference 36), in which it was proposed that terminal Ir–O π bonds in the analogous system, Li₂Ir_{1-x}Sn_xO₃, were responsible for oxygen redox. From the structure in Figure 2b, it appears that disorder of the Li-Ru sites leads to similar vacancy configurations in the current system that were present in the previous work after Sn migration. The authors should discuss their findings in the context of this previous study and highlight whether their O–Ru–O environments are related to the formation of analogous Ru–O π bonds.

Response to Comment 1

Thanks for referee’s constructive suggestions. To confirm whether the O–Ru–O environments in our work are related to the formation of analogous Ru–O π bonds, the crystal orbital overlap population (COOP) analysis was performed to study the characteristics of short Ru–O bond in delithiated state ID-Li₀RuO₃ and normal Ru–O bond in R-Li₀RuO₃, as shown in Figure S1. The integrated COOP below Fermi level of the short Ru–O bonds in ID-Li₀RuO₃ increases by 51% when compared with normal Ru–O bonds in R-Li₀RuO₃, implying that the net bond order of the short Ru–O bonds in ID-Li₀RuO₃ is higher than that of Ru–O bonds in R-Li₀RuO₃. Considering the higher net bond order and the bond length of 1.67 Å that is close to the previously reported bond lengths of Ru⁵⁺=O double bond (1.63 Å,¹ 1.676 Å,² 1.697 Å,² and 1.70 Å³), this terminal Ru–O short bond can be regarded as quasi Ru⁵⁺–O double bond with a π -type hybridization between Ru (t_{2g}) and O (2p). This is similar to the previous proposed Ir–O π bonds in Li₂Ir_{1-x}Sn_xO₃ system after TM ions migration to Li layer.⁴ Thus, the telescopic O–Ru–O configurations also involve the formation of Ru–O π bonds. In the revised manuscript, the Ru–O π bonds were highlighted for the short Ru–O bond in O–Ru–O configuration and the previously proposed Ir–O π bonds⁴ were discussed.

Although both Li₂Ir_{1-x}Sn_xO₃ in Hong’s work⁴ and ID-Li₂RuO₃ in our work involve drastic TM–O bond shortening, these two works are different from each other in the following aspects: 1) the significance and implication of the main findings; 2) the structural triggering mechanism of TM–O π bonds; 3) the reversibility of local-range structural change; 4) the presence of O–O dimerization; 5) the reversibility of long-range structural evolution.

1) First, our findings aim to propose a strategy, namely intralayer Li/TM disordering, to tune

the local symmetry around oxygen ions and thus enhance the robustness of oxygen networks by inhibiting the O-O dimerization/O₂ release, which finally promotes the reversibility of anionic redox a lot as evidenced by the high cycling stability of ID-Li₂RuO₃. Although similar phenomenon (TM-O π bonds) was proposed by Hong et al.,⁴ their work mainly focuses on explaining why anionic redox occurs frequently with local structural disordering (TM migration) and how such a point-defect mechanism affects the unusual electrochemical behavior induced by anionic redox, such as voltage fade and hysteresis. Specially, two forms of anionic redox behavior, i.e. O-O dimer and Ir-O π bonds, were proposed by theoretical DFT calculation while lack of solid experimental evidences. Besides, they didn't mention at all how the existence of Ir-O π bonds influence the reversibility of anionic redox but which is highly relevant with our work. Therefore, for significance and implications, we have completely different initiatives when we were doing such a work.

2) Second, regarding the structural triggering mechanism, the TM-O bond shortening in Li₂Ir_{1-x}Sn_xO₃ was caused by antisite-vacancy introduced by TM migration during delithiation, which is undesirable since it causes capacity and voltage irreversibility. Moreover, it is also uncontrollable since TM migration is coupled with anionic redox itself. However, in our ID-Li₂RuO₃ system, the telescopic O-TM-O configuration during charging was caused by designing special local symmetry around oxygen ions, which is highly controllable and beneficial for the reversibility of anionic redox. TM migration does not occur in our ID-Li₂RuO₃ system, as demonstrated by XRD refinement of charged ID-Li₂RuO₃ (Figure S16, Table S8-9), and also supported by the results of formation energy of Ru antisite defect from DFT calculations (Figure S5), which is a good news for cycling stability.

3) Third, the reversibility of local-range structural change, i.e., Ir-O bond evolution, was not confirmed in Hong's work. However, in our work, the local-range structural response of the telescopic O-TM-O evolution is visualized by ABF-STEM image (Figure 7a), which proved to be reversible in ID-Li₂RuO₃ by EXAFS (Figure 7h, Figure S23a, and Table S11).

4) Fourth, Hong's work didn't eliminate the O-O dimerization which occurs simultaneously with Ir-O short bond. However, O-O dimerization disappeared in our ID-Li₂RuO₃ as indicated by Raman spectra (Figure 7f). Furthermore, our ID-Li₂RuO₃ showed no O₂ release during charging/discharging even at a higher voltage of 5.0 V as verified by in situ DEMS (Figure 6d,

Figure S19). In contrast, as for the R-Li₂RuO₃ served as a control group, the irreversible local-range structural response with O₂ release that caused by O-O dimerization during oxygen redox processes is proved by in situ DEMS (Figure 6e), in accordance with previous studies.⁵

5) Fifth, the long-range structural evolution is irreversible for Li₂Ir_{1-x}Sn_xO₃ in Hong's work, as indicated by XRD patterns. However, in our work, the long-range structural evolution is reversible for ID-Li₂RuO₃ (Figure 6a-c, Figure S17), whereas the long-range structural evolution is irreversible for R-Li₂RuO₃ (Figure S18), as observed by XRD patterns. Combining with the long- and local-range structural evolution, the ID-Li₂RuO₃ shows excellent cycling stability.

Changes in the revised manuscript:

The detailed data and discussions have been added to the revised manuscript (Line 2-13, Page 8) and the Supplementary Materials as follows:

As for the short Ru-O bonds, the crystal orbital overlap population (COOP) analysis was performed to study the interaction between Ru and O, as shown in Figure S1. The integrated COOP of the short Ru-O bonds in ID-Li₀RuO₃ below Fermi level increases by 51% when compared with Ru-O bonds in R-Li₀RuO₃, implying that the net bond order of the short Ru-O bonds in ID-Li₀RuO₃ is higher than that of Ru-O bonds in R-Li₀RuO₃. Considering the higher net bond order and the bond length of 1.67 Å that is close to the previously reported bond lengths of Ru⁵⁺=O double bond (1.63 Å,³⁹ 1.676 Å,⁴⁰ 1.697 Å,⁴⁰ and 1.70 Å⁴¹), this terminal Ru-O short bond can be regarded as quasi Ru⁵⁺=O double bond with a π-type hybridization between with Ru (*t_{2g}*) and O (*2p*). This is similar to the previous proposed Ir-O π bonds in Li₂Ir_{1-x}Sn_xO₃ system after TM ions migration to Li layer.³⁶

Figure S1. The crystal orbital overlap population (COOP) of the short Ru–O bond in ID-Li₀RuO₃ and normal Ru–O bond in R-Li₀RuO₃.

Comment 2

In Figure 2c, the authors highlight that the oxygen release energy is less favourable for the Ru1Li5 and Ru3Li3 configurations of the ID-Li₂RuO₃ structure than the Ru2Li4 environments in the R-Li₂RuO₃ structure, which leads to less O₂ loss in the former case. However, the authors do not include the oxygen release energy for the Ru2Li4 configuration of the ID-Li₂RuO₃ structure. From the schematic in Figure 2b and TEM in Figure 3, these environments Ru2Li4 should also be present in the ID-Li₂RuO₃ structure. The authors should show whether there is a difference in the driving force for oxygen evolution between the Ru2Li4 environments in the ID-Li₂RuO₃ and R-Li₂RuO₃ structures and if there is not, explain why the ID-Li₂RuO₃ structure shows considerably less O₂ evolution.

Response to Comment 2

Thanks for referee's kind suggestion. The oxygen release energies for the O_{center}[Ru₂Li₄] environments in the ID-Li_{2-x}RuO₃ systems are supplemented in Figure 2c. Generally, the oxygen release energies for the O_{center}[Ru₂Li₄] environments in ID-Li_{2-x}RuO₃ systems are between the two values for O_{center}[Ru₁Li₅] environments and O_{center}[Ru₃Li₃] environments in ID-Li_{2-x}RuO₃. Within the same ID-Li_{2-x}RuO₃ system, the oxidation extent for oxygen ions in O_{center}[Ru₂Li₄] environments is between that for oxygen ions with O_{center}[Ru₁Li₅] environments and O_{center}[Ru₃Li₃]

environments (Figure S15b). The oxygen release energies at deep delithiation states ($x > 1$ for $\text{Li}_{2-x}\text{RuO}_3$) for $\text{O}_{\text{center}}[\text{Ru}_2\text{Li}_4]$ environments in the ID- Li_2RuO_3 (positive values) are higher than that for $\text{O}_{\text{center}}[\text{Ru}_2\text{Li}_4]$ environments in the R- Li_2RuO_3 (negative values), which is related to the total energy influenced by overall structural evolution of the systems. Thus, the driving force for oxygen evolution is weaker and oxygen is more stable in ID- Li_2RuO_3 .

Changes in the revised manuscript

The oxygen release energies for the $\text{O}_{\text{center}}[\text{Ru}_2\text{Li}_4]$ environments in the ID- Li_2RuO_3 structure are supplemented in Figure 2c. And the corresponding discussions were given in the revised manuscript (Line 1-5, Page 9) as follows:

the oxygen release energies for $\text{O}_{\text{center}}[\text{Ru}_1\text{Li}_5]$ coordination (green dashed line), $\text{O}_{\text{center}}[\text{Ru}_2\text{Li}_4]$ coordination (purple dashed line) and $\text{O}_{\text{center}}[\text{Ru}_3\text{Li}_3]$ coordination (blue dashed line) in ID- Li_2RuO_3 are all more positive than that for R- Li_2RuO_3 after deep delithiation, which is related to the total energy influenced by overall structural evolution of the systems, indicating that the oxygen is more stable in ID- Li_2RuO_3 .

Figure 2. Optimized crystal structures and local RuO_6 octahedrons of Li_2RuO_3 and the corresponding delithiated state (Li_0RuO_3) for R- Li_2RuO_3 (a) and ID- Li_2RuO_3 (b). The values (in

angstrom) on the local structures are the Ru–O bond lengths and O–O distances. (c) Oxygen release energy for R-Li_{2-x}RuO₃ and ID-Li_{2-x}RuO₃ systems.

Comment 3

The authors use a combination of X-ray diffraction and neutron diffraction to refine the crystal structures of the ID-Li₂RuO₃ and R-Li₂RuO₃ phases, however, they do not provide details of the fitting procedure which are necessary to properly assess the data. The occupancy of the 4e Li sites are fixed at 1. Did the authors try and refine the occupancy of Ru on the 4e site, i.e. antisite disorder? In the power X-ray diffraction pattern of ID-Li₂RuO₃ in Figure S2, there is a broad bump between 20–30° 2θ which is characteristic of superstructure ordering of the Li_{1/3}Ru_{2/3} honeycomb layers. Was this region included in the refinement of the structures? Where the thermal parameters refined for each structure?

The structure of R-Li₂RuO₃ has been reported in a number of studies (Wang, J.C., Cao, G. et al, 2014. *Physical Review B*, 90(16), p.161110.) where it has been shown that neighbouring Ru ions form Ru–Ru dimers which form ordered arrangements that lower the symmetry from C2/c to P21/m or C2/m. The presence of Ru–Ru dimers has also been suggested to be linked to the decrease in the voltage and electrochemical performance in Ru doped Li₂MnO₃ (Knight, J.C., Manthiram, A., et al. 2015. *Journal of Materials Chemistry A*, 3(5), pp.2006-2011). Did the authors consider the P21/m or C2/m structures during their refinement? The authors should consider the presence of Ru–Ru dimers in their structure and how the disordered arrangement of Li and Ru may alter the long-range dimer structure and electrochemistry.

Response to Comment 3

Thanks for referee's careful check and constructive suggestions.

i) First, as for the structure of Li₂RuO₃, the space groups of C2/c,⁶⁻¹¹ C2/m¹²⁻¹⁴ and P21/m¹²⁻¹⁵ were reported. The structure of Li₂RuO₃ at room temperature is often described as adopting the C2/c unit cell, as was reported in most of the literature⁶⁻¹¹. Owing to the significant influence of the Ru dimerization on the magnetism, heat capacity, and resistivity, its effect on the structure been more thoroughly explored by the physics community. Kakurai et al.¹² suggested that the structure adopts the space group of P21/m at low temperatures (< 540 K) and C2/m at high

temperatures (> 540 K) by using the powder neutron diffraction. Wang et al.¹⁴ had proposed that either C2/m- or P21/m-type single crystals, as well as P21/m-type polycrystalline Li_2RuO_3 at temperatures below 300 K exist, but the synthesis conditions of C2/m- and P21/m-type single crystals described in the main text are the same. Maeno et al.¹⁶ believed that the Rietveld refinement of neutron diffraction is equally good for the space groups C2/m and C2/c after comparing his work with Kakurai's work¹². Indeed, due to the similarity of XRD patterns, further research is needed to confirm the space group of Li_2RuO_3 . Considering the complexity of the space group mentioned above, the space group of Li_2RuO_3 at room temperature still cannot be ascertained simply from temperature. All the three space group of C2/m, C2/c, and P21/m are possible for R- Li_2RuO_3 . Thus, the possibility of all the three space groups of C2/m, C2/c, and P21/m are considered for both the R- Li_2RuO_3 and ID- Li_2RuO_3 samples here. As for the R- Li_2RuO_3 sample, XRD patterns of the R- Li_2RuO_3 structures copied from reference with space groups of C2/m¹⁴, P21/m¹⁴, and C2/c⁶ are simulated by Materials Studio software for comparison, as shown in Figure R1. The peaks at $\sim 18.8^\circ$ (corresponding to $(10\bar{1})$ peak of P21/m) and $\sim 44.6^\circ$ (corresponding to (202) peak of C2/c) are absence for the R- Li_2RuO_3 sample. R- Li_2RuO_3 sample was fitted best to C2/m space group, although the refinements of the R- Li_2RuO_3 are equally good for the space groups C2/m and C2/c, as is highlighted in references^{12,16}. Similarly, the refinements of the ID- Li_2RuO_3 are equally good for the space groups C2/m and C2/c. The space group of ID- Li_2RuO_3 sample is confirmed by comparing the observed and simulated patterns of the selected area electron diffraction (SAED). Figure R2 shows the observed SAED patterns of ID- Li_2RuO_3 sample (a, b), the simulated SAED patterns of C2/m-type ID- Li_2RuO_3 (c, d) and C2/m-type R- Li_2RuO_3 (e, f) along $[100]$ and $[\bar{1}00]$ zone axes, C2/c-type ID- Li_2RuO_3 (g, h) and C2/c-type R- Li_2RuO_3 (i, j) along $[100]$ and $[\bar{1}01]$ zone axes. The simulated SAED patterns of C2/c and C2/m cells viewing along $[100]$ zone axis are different. The simulated SAED of C2/c cell along $[100]$ zone axis contains additional spots (marked with blue arrows), compared with that of the simulated SAED of C2/m cell. It can be found that the observed SAED patterns in Figure R2a is the same with the simulated SAED patterns of C2/m-type ID- Li_2RuO_3 along $[100]$ zone axis (Figure R2c). If viewed along the $[001]$ zone axis, the observed image is the same with the simulated image of $[001]$ zone axis of C2/m ID- Li_2RuO_3 . Thus, ID- Li_2RuO_3 sample are confirmed to be C2/m structure with Ru/Li intralayer disordering arrangement within TM layer.

ii) Second, the antisite of Ru on Li site in Li layer is considered in the revised manuscript. The antisite of Ru in Li layer was refined to be $\sim 0.02\%$ of the total Li content in Li layer, as listed in Table S5 and Table S7 for XRD refinement and NPD refinement, respectively. Thus, the antisite of Ru was generally absent. The XRD patterns with different antisite concentrations were also simulated by using Materials studio software, as shown in Figure R3a. The XRD patterns were normalized by the intensity of (001) peak. As the antisite concentration increases, the peak A decreases while the peak B and C increases. The relative intensity of peak A and peak C (Figure R3b) show that the ratio of the antisite absent system fitted best with that of the observed case. Thus, the antisite Ru on Li site in Li layer is almost negligible.

iii) Third, the broad bump between $20\text{--}30^\circ$ 2θ was included in the refinement of the structures in the revised manuscript. The occupancy of Ru and Li in TM layers are carefully refined. The results of XRD (Figure 3a and Table S4-5) and NPD (Figure S9 and Table S4, S7) refinement show that the occupancies of Ru are ~ 0.70 and 0.60 at 4h and 2d site, and occupancies of Li are ~ 0.30 and 0.40 at 4h and 2d site, which are close to the Ru and Li occupancies of the ideal intralayer disordered Li_2RuO_3 with 0.667 (Ru) and 0.333 (Li) at both 4h and 2d site. Thus, the structure of ID- Li_2RuO_3 sample was similar to the ideal intralayer disordered Li_2RuO_3 . In order to evaluate the extent of intralayer disordering, two phase including regular Li_2RuO_3 and ideal intralayer disordered Li_2RuO_3 were used for refinement, which shows that the ratio of regular Li_2RuO_3 and ideal intralayer disordered Li_2RuO_3 phases is about 35: 1. The percentage of the ideal intralayer disordered Li_2RuO_3 phase is 97.1%, confirming that the ID- Li_2RuO_3 sample is almost the idea intralayer disordered Li_2RuO_3 phase.

iv) Fourth, the thermal parameters refined in the revised manuscript, the final values are shown in Table S5 and S7.

v) Fifth, regarding the effect of Ru–Ru dimer on electrochemical performance. Both the ID- Li_2RuO_3 and R- Li_2RuO_3 samples with C2/m space group don't contain Ru–Ru dimer. The absence of Ru-Ru dimer in ID- Li_2RuO_3 might be related to the disruption of long term hexagon arrangement of Ru ions within the TM layer, because both the zigzag- and armchair-type of Ru–Ru dimer arrangement are based on long term hexagon arrangement.¹⁶ Considering that the space groups of P2₁/m with Ru-Ru dimer and C2/m without Ru-Ru dimer are both possible for R- Li_2RuO_3 at room temperature, as discussed above, the relatively pure effect of the Ru–Ru dimer

on electrochemical performance can be studied by comparing C2/m- and P2₁/m-type R-Li₂RuO₃ samples. Fortunately, P2₁/m type R-Li₂RuO₃ was additionally synthesized here (XRD patterns shown in Figure R4a) to preliminary investigate the effect of Ru–Ru dimer on electrochemical performance. The main difference of synthesis condition is that the RuO₂ and RuO₂·xH₂O were used to prepare the C2/m- and P2₁/m-type R-Li₂RuO₃, respectively. The galvanostatic charge/discharge tests of the two type of R-Li₂RuO₃ were performed. As shown in Figure R4b, charge-discharge curves for C2/m- and P2₁/m-type R-Li₂RuO₃ cathodes are quite different. Both the two kinds of charge-discharge curves had been reported, the charge-discharge curves in some references¹¹ similar to our C2/m-type charge-discharge curve, while some other references¹⁷ agree well with our P2₁/m-type charge-discharge curve. However, the two kind of charge-discharge curves had not been compared in references. This difference might be related to Ru–Ru dimer formation, as the effect of Ru–Ru dimer on electrochemical performance was reported by Knight et al.¹⁸ However, the cycling stability of both the C2/m type R-Li₂RuO₃ sample and the P2₁/m type R-Li₂RuO₃ sample are poor, as shown in Figure R4c. The effect of Ru–Ru dimer on electrochemical performance might be a good topic for next work. In the present work, both the ID-Li₂RuO₃ and R-Li₂RuO₃ sample with C2/m space group do not contain Ru–Ru dimer, the effect of local symmetry tuned by intralayer disordering was the focus of our work.

Figure R1. Simulated XRD patterns of the R-Li₂RuO₃ structures copied from reference with space groups of C2/m¹⁴, P2₁/m¹⁴, and C2/c⁶, alongside with the observed XRD patterns of R-Li₂RuO₃ sample.

Figure R2. The observed SAED patterns of ID-Li₂RuO₃ sample (a, b), the simulated SAED patterns of C2/m-type ID-Li₂RuO₃ (c, d), C2/m-type R-Li₂RuO₃ (e, f) along [100] and [100] zone axes, C2/c-type ID-Li₂RuO₃ (g, h) and C2/c-type R-Li₂RuO₃ (i, j) along [100] and [101] zone axes.

Figure R3. (a) The simulated XRD patterns with different antisite concentrations based on C2/m space group idea ID-Li₂RuO₃ structure, along with the observed XRD pattern of ID-Li₂RuO₃ sample; (b) The relative intensity of peak A (I_A) and peak C (I_B), I_A/I_B. The idea ID-Li₂RuO₃ structure was obtained by disordering the distribution of Ru and Li ion in TM layer with 66.667% Ru and 33.333% Li occupancy for all sites based on regular Li₂RuO₃ structure from reference¹⁴.

Figure R4. (a) The XRD patterns of C2/m- and P2₁/m-type R-Li₂RuO₃; The charge–discharge curves (b) and the cycling performance (c) of C2/m- and P2₁/m-type R-Li₂RuO₃ in the voltage range of 2.0–4.8 V at a current density of 30 mA/g (0.1 C).

Changes in the revised manuscript

The XRD and NPD refinements (Figure 3a-b, Figure S9, Table S4-7) were revised with careful checks on the space group, antisite defects, broad bump, and thermal parameters. The observed and simulated selected area electron diffraction (SAED) patterns (Figure S8) were provided. Discussions are added in the revised manuscript (Line 9-17, Page 10, and Line 9-16, Page 11). Charges are as follows:

Unlike R-Li₂RuO₃, the ID-Li₂RuO₃ sample exhibit negligible superstructure reflection peaks (such as the peaks in the 2θ range of 20°–35°, highlighted in Figure 3b), which suggests that TM/Li-intralayer disordering within TM layer exists in ID-Li₂RuO₃ sample. Specifically, according to refinement results of ID-Li₂RuO₃, the Ru and Li occupancies are 0.701515 (Ru) and 0.298485 (Li) at 4h site, and 0.596268 (Ru) and 0.403732 (Li) at 2d site, which are close to 0.667 (Ru) and 0.333 (Li) of the Ru and Li occupancies at both 4h and 2d site in the ideal TM/Li-intralayer disordered Li₂RuO₃. Thus, the structure of ID-Li₂RuO₃ sample was similar to the ideal intralayer disordered Li₂RuO₃. In order to evaluate the extent of intralayer disordering, two phase including regular Li₂RuO₃ and ideal intralayer disordered Li₂RuO₃ were used for refinement, which shows that the ratio of regular Li₂RuO₃ and idea intralayer disordered Li₂RuO₃ phases is

about 35: 1. The percentage of the ideal intralayer disordered Li_2RuO_3 phase is 97.1%, confirming that the ID- Li_2RuO_3 sample is almost the ideal intralayer disordered Li_2RuO_3 phase. Thus, the intralayer disordered Li_2RuO_3 was achieved successfully.

The observed and simulated selected area electron diffraction (SAED) patterns (Figure S8) were also given to analyze the structure on long-range scale. The ID- Li_2RuO_3 and R- Li_2RuO_3 structures with C2/m space group used for SAED simulation are taken from the XRD refinements. The observed SAED patterns of the as-prepared ID- Li_2RuO_3 sample shown in Figure S8a and b that characterized with the marked weaker diffraction spots (red circles) are consistent with the simulated SAED patterns of ID- Li_2RuO_3 structure model along [100] (Figure S8c) and [001] (Figure S8d) zone axis, respectively. Therefore, the intralayer disordering is verified by SAED patterns on long-range scale. Neutron powder diffraction (NPD) patterns were also obtained to further analyze the structural properties of the ID- Li_2RuO_3 sample. As shown in Figure S9, the results of NPD refinement (details are listed in Table S4, and S7) show Ru/Li-intralayer disordering, which is similar to XRD refinement. Hence, the TM/Li-intralayer disordered arrangement in the ID- Li_2RuO_3 sample was further confirmed by NPD results.

Figure 3. XRD patterns of ID- Li_2RuO_3 (a) and R- Li_2RuO_3 (b). The insets show the corresponding crystal models after refinement. HAADF-STEM images of the ID- Li_2RuO_3 sample along the [100] (c) and [001] (d) zone axes.

Table S4. Crystallographic parameters and structure determination details for ID-Li₂RuO₃ and R-Li₂RuO₃ from XRD and NPD refinement.

	ID-Li ₂ RuO ₃		R-Li ₂ RuO ₃
	XRD refinement	NPD refinement	XRD refinement
Space group	C2/m (No. 12)	C2/m (No. 12)	C2/m (No. 12)
a (Å)	4.9938	5.0065	5.039858
b (Å)	8.6844	8.7082	8.749701
c (Å)	5.17981	5.1848	5.141971
$\alpha = \gamma$ (°)	90.000	90.000	90.000
β (°)	108.587	108.640	109.0488
Volume (Å ³)	212.922	214.19	214.330
R _{wp} (%)	5.11	3.58	11.04
R _p (%)	3.88	2.74	8.61
χ^2	4.137	3.897	1.836

Table S5. Atomic coordinates of ID-Li₂RuO₃ from XRD refinement.

Atom	Site	Coordinates			Occupation	Uiso (Å ²)
		x	y	z		
Ru(1)	4h	0.000000	0.166728	0.500000	0.701515	0.056554
Li(1)	4h	0.000000	0.166728	0.500000	0.298485	0.029000
Li(2)	2d	-0.500000	0.000000	0.500000	0.403732	0.029000
Ru(2)	2d	-0.500000	0.000000	0.500000	0.596268	0.052420
Li(3)	4g	-0.500000	-0.166720	0.000000	0.999757	0.020071
Ru(3)	4g	-0.500000	-0.166720	0.000000	0.000243	0.055812
Li(4)	2a	0.000000	0.000000	0.000000	0.999784	0.028834
Ru(4)	2a	0.000000	0.000000	0.000000	0.000216	0.05695
O(1)	8j	0.254592	0.168280	0.270732	1.000000	0.023340
O(2)	4i	-0.241671	0.000000	0.272298	1.000000	0.031790

Table S6. Atomic coordinates of R-Li₂RuO₃ sample from XRD refinement.

Atom	Site	Coordinates			Occupation	Uiso (Å ²)
		x	y	z		
Ru	4h	0.000000	0.167859	0.500000	1.000000	0.053716
Li(1)	2d	-0.500000	0.000000	0.500000	1.000000	0.019000
Li(2)	4g	-0.500000	-0.158000	0.000000	1.000000	0.021024
Li(3)	2a	0.000000	0.000000	0.000000	1.000000	0.018437
O(1)	8j	0.248934	0.172385	0.267545	1.000000	0.039705
O(2)	4i	-0.233513	0.000000	0.267525	1.000000	0.031569

Figure S8. The observed SAED patterns (a-b) and simulated SAED patterns of ID-Li₂RuO₃ and R-Li₂RuO₃ structure models along [100] (c-d) and [001] (e-f) zone axes, respectively. The ID-Li₂RuO₃ and R-Li₂RuO₃ structure models with C2/m space group used for SAED simulation are taken from the XRD refinements.

Figure S9. NPD patterns of ID-Li₂RuO₃. The impurity peaks of the vanadium pot are omitted within the marked ranges.

Table S7. Atomic coordinates of ID-Li₂RuO₃ from NPD refinement.

Atom	Site	Coordinates			Occupation	Uiso (Å ²)
		x	y	z		
Ru(1)	4h	0.000000	0.166732	0.500000	0.701204	0.031036
Li(1)	4h	0.000000	0.166732	0.500000	0.298796	0.023590
Li(2)	2d	-0.500000	0.000000	0.500000	0.403153	0.030742
Ru(2)	2d	-0.500000	0.000000	0.500000	0.596847	0.042297
Li(3)	4g	-0.500000	-0.166719	0.000000	0.999751	0.027274
Ru(3)	4g	-0.500000	-0.166719	0.000000	0.000249	0.047763
Li(4)	2a	0.000000	0.000000	0.000000	0.999753	0.027849
Ru(4)	2a	0.000000	0.000000	0.000000	0.000247	0.047832
O(1)	8j	0.254588	0.168285	0.270731	1.000000	0.024047
O(2)	4i	-0.241668	0.000000	0.272302	1.000000	0.028501

Comment 4

As highlighted in the introduction by the present authors, there is a link between cation migration and anionic redox in a number of Li-excess cathode materials. Did the authors calculate the energy to form Ru anti-site defects in the delithiated ID-Li_{2-x}RuO₃ and R-Li_{2-x}RuO₃ structures analogous to the calculations of Ir and Sn antisite disorder in reference 36?

Response to Comment 4

Cation migration is linked with anionic redox especially in the case of oxygen release. The migration is evaluated from DFT calculation and XRD refinement in the revised manuscript. The

energy to form Ru antisite defects (Ru migrate to octahedral sites of Li layer) in deep delithiated ID-Li_{2-x}RuO₃ and R-Li_{2-x}RuO₃ ($x=1.5, 1.75, 2$) were calculated at the same defect concentration (1 out of 16 Ru ion migrated to octahedral site Li layer), as shown in Figure S5. Generally, the anti-site defects formation energy of ID-Li_{2-x}RuO₃ are much higher than that of R-Li_{2-x}RuO₃, which means the Ru migration in ID-Li_{2-x}RuO₃ is much harder than that in R-Li_{2-x}RuO₃. Moreover, the formation energy become negative at a low Li content ($x \sim 2.0$) for R-Li₂RuO₃, which means Ru ion is easy to migrate, thus harm to electrochemical performance such as cycling stability.

On the other hand, the XRD refinement results of delithiated ID-Li_{2-x}RuO₃ is further analyzed to confirm cation migration (Figure S16, Table S8, S9). The antisite defects of Ru in Li layer are about 0.023% and 0.025% of the total Li site in Li layer for pristine (Table S5) and charged (4.8 V) ID-Li_{2-x}RuO₃ (Table S9), respectively, which means the antisite of Ru in Li layer in both pristine and charged (4.8 V) ID-Li₂RuO₃ are almost absent. Thus, the Ru migration during delithiation is negligible. The negligible Ru migration is consistent with the excellent cycling stability.

Changes in the revised manuscript

Figure S5 was added to predict the TM migration from DFT calculation. Figure S16, Table S8-S9 were given to confirm the absence of TM migration XRD refinement. Discussions were added in the revised manuscript (Line 9-13, Page 9; Line 12-22, Page 20; Line 1-2, Page 21) of page 18. Changes are as follows:

In addition, since TM migration to Li layer would be promoted by oxygen release, the energy to form antisite defects of Ru in Li layer is calculated (Figure S5), which shows a much higher formation energy in ID-Li₂RuO₃ than in R-Li₂RuO₃. Thus, the Ru migration should be much more difficult in ID-Li₂RuO₃ than in R-Li₂RuO₃.

According to the refinement of XRD pattern of the 4.8 V charged ID-Li₂RuO₃, we find that ID-Li₂RuO₃ kept in C2/m phase with lattice parameter changed during delithiation, as shown in Figure S16, Table S8 and S9. The β was changed from 108.5870° to 90.0097°, indicating that the layered structure was altered from O3- to O1-type C2/m phase.^{12,36} As shown clearly in Figure S17, the phase changed gradually from O3- to O1-type structure during charge process, then

almost returned back to O3-type structure of the pristine during discharge process. Hence, the long-range structure of ID-Li₂RuO₃ is reversible during charge and discharge processes. In addition, the migration of Ru to Li layer is almost absent according to the XRD refinement as the occupancies of Ru in Li layer are about 0.023% and 0.025% of the total Li site in Li layer for pristine and charged (4.8 V) ID-Li_{2-x}RuO₃, respectively, which is consistent with the results of the formation energy of Ru anti-site defects (Figure S5).

Figure S5. The formation energy (ΔE) of Ru anti-site defects (Ru migrate to octahedral sites of Li layer) in deep delithiated ID-Li_{2-x}RuO₃ and R-Li_{2-x}RuO₃ ($x=1.5, 1.75, 2$). The same Ru anti-site defects concentration of 1/16 (6% of the total Ru content) is considered for both ID-Li_{2-x}RuO₃ and R-Li_{2-x}RuO₃.

Figure S16. The refinement of XRD patterns of ID-Li₂RuO₃ that charged to 4.8 V.

Table S8. Crystallographic parameters and structure determination details for 4.8V charged ID-Li₂RuO₃ from XRD refinement.

Sample	4.8Vcharged ID-Li ₂ RuO ₃
Space group	C2/m (No. 12)
a (Å)	5.037718
b (Å)	8.718863
c (Å)	4.712646
$\alpha = \gamma$ (°)	90.000
β (°)	90.0097
Volume (Å ³)	206.99
R _{wp} (%)	4.22
R _p (%)	3.22
χ^2	4.757

Table S9. Atomic coordinates of 4.8 V charged ID-Li₂RuO₃ from XRD refinement.

Atom	Site	Coordinates			Occupation	Uiso (Å ²)
		x	y	z		
Ru(1)	4h	0.000000	0.168110	0.500000	0.701547	0.045805
Li(1)	4h	0.000000	0.168110	0.500000	0.030000	0.039030
Li(2)	2d	-0.500000	0.000000	0.500000	0.030000	0.039030
Ru(2)	2d	-0.500000	0.000000	0.500000	0.596161	0.047284
Li(3)	4g	-0.500000	-0.158000	0.000000	0.100000	0.038628
Ru(3)	4g	-0.500000	-0.158000	0.000000	0.000254	0.045635
Li(4)	2a	0.000000	0.000000	0.000000	0.100000	0.038670
Ru(4)	2a	0.000000	0.000000	0.000000	0.000237	0.048176
O(1)	8j	0.330751	0.170456	0.273423	1.000000	0.039911
O(2)	4i	-0.150312	0.000000	0.276555	1.000000	0.037253

Comment 5

The properties of a range of delithated Li_{2-x}RuO₃ structures are calculated with DFT calculations, but the authors do not give details about how the delithated structures were generated. Were multiple Li orderings considered for each structure? The authors should provide more details about the structures were generated and ideally include the lowest energy structures in the SI.

Response to Comment 5

The multiple Li ordering of each delithated Li_{2-x}RuO₃ structures had been tested. The final

structures are the lowest energy structures among the multiple Li ordering. The formation energies for Li removal in R-Li_{2-x}RuO₃ and ID-Li_{2-x}RuO₃ (x = 0, 0.5, 1.0, 1.5, 1.75, 2.0) are given in Figure S4, and the lowest energy structures for R-Li_{2-x}RuO₃ and ID-Li_{2-x}RuO₃ are shown in Figure S2 and S3, respectively.

Changes in the revised manuscript

Detailed structures (Figure S2, S3), formation energies (Figure S4), and corresponding descriptions have been added to the revised manuscript (Line 18-20, Page 7). Changes are as follows:

The final structures for R-Li_{2-x}RuO₃ and ID-Li_{2-x}RuO₃ (x = 0, 0.5, 1, 1.5, 1.75, 2) are shown in Figure S2 and S3, which were tested to be the lowest energy structures among the multiple Li ordering (Figure S4).

Figure S2. The lowest energy structures of R-Li_{2-x}RuO₃ for x=0, 0.5, 1.0, 1.5, 1.75, 2.0. The purple, dark cyan, and red spheres are Li, Ru, and O, respectively.

Figure S3. The lowest energy structures of ID- $\text{Li}_{2-x}\text{RuO}_3$ for $x=0, 0.5, 1.0, 1.5, 1.75, 2.0$. The purple, dark cyan, and red spheres are Li, Ru, and O, respectively.

Figure S4. The delithiation formation energy of ID- $\text{Li}_{2-x}\text{RuO}_3$ (a) and R- $\text{Li}_{2-x}\text{RuO}_3$ (b). The formation energy is defined as $E_f(\text{Li}_{2-x}\text{RuO}_3) = E(\text{Li}_{2-x}\text{RuO}_3) - 0.5(2-x) E(\text{Li}_{2-x}\text{RuO}_3) - 0.5 x E(\text{RuO}_3)$.

Reply to Reviewer 2

General Comment

The author reported a new structural response mechanism that may inhibit O–O dimerization and then improve cycling and voltage stabilities without oxygen release. Solving oxygen release is an important but challenging task for Li-rich cathode. This is a very important piece of this work and may impact on a wider field. While there were some interesting findings and good improvements, such as suppressing O–O dimerization, the writing should be significantly improved, mainly in the quality of figures before publication. Here are a few issues to fix.

Response to the General Comment

We thank the referee for considering that “there were some interesting findings and good improvements”. And thank the referee for his/her useful comments on our work. We have revised the manuscript according to the referee’s advices. The quality of figures was improved.

Comment 1

The telescopic O–TM–O configuration seems be something similar to the idea of electron hole used by Tarascon's, Bruce's and Ceder's groups. Please elaborate the conceptive novelty in the telescopic O–TM–O configuration with respect to or simply a realization of electron hole?

Response to Comment 1

We agree with the reviewer that there are two kinds of anionic redox behavior in reported Li-rich compounds, i.e. $2O^{2-} \rightarrow O_2^{n-}$ (O–O dimer or O_2 release)¹⁹ and $O^{2-} \rightarrow O^{n-}$ (hole state on O-2p orbitals).^{20,21} The local structural response mode for the $2O^{2-} \rightarrow O_2^{n-}$ behavior is widely acknowledged, which is realized via O–O dimerization or even O_2 gas generation, as has been evidenced directly by ABF-STEM image,¹⁹ Raman spectra,²² and DEMS²³ results.

For the second kind of behavior ($O^{2-} \rightarrow O^{n-}$), it’s existence has been previously proposed by ruling out O–O dimer formation based on spectroscopy results like RIXS, DEMS, Raman,^{20,21} which, however, only revealed a different charge compensation behavior related to hole state on O-2p

orbitals from electronic structural considerations. The specific local structural response mode of $O^{2-} \rightarrow O^{n-}$ have not been declared previously. Moreover, the validity of the local electron hole in O 2p band still requires more direct and competent experimental evidences, and more theoretical analysis is needed to account for the rationality of why and when this behavior ($O^{2-} \rightarrow O^{n-}$) can occur.

In our work, the clearly defined local structural response mode related to specific telescopic O–TM–O configuration for $O^{2-} \rightarrow O^{n-}$ process is proposed for the first time. The specific local structural response mode of telescopic O–Ru–O configuration show excellent reversibility. More importantly, based on the comparative studies on R- Li_2RuO_3 and ID- Li_2RuO_3 , we for the first time propose to tune the local structural asymmetry (Li/M intralayer disordering) to trigger this $O^{2-} \rightarrow O^{n-}$ process in order to have better reversibility of anionic redox, which is realized by means of a specific telescopic O–TM–O configuration as we proposed, of great paramount importance is this work first exemplifies how we can utilize structural design approach to implement $O^{2-} \rightarrow O^{n-}$ process, rather than proposing a conceptual idea that is similar with previous work. Besides, what differs lot from previous work is that we also supply various experimental evidences, such as ABF-STEM (Figure 7a-c), Raman (Figure 7f), EXAFS (Figure 7g-h, Figure S21-23, Table S10-11), DEMS (Figure 6d-e), and cycling stability (Figure 4), to either directly or indirectly prove the existence of such electron-hole mechanism in opposite to O-O dimerization in R- Li_2RuO_3 . Lastly, as started from DFT, this work also gives theoretical instructions and understandings on the realization of electron-hole anionic redox, which are totally absent in previous works.

Comment 2

The quality of Figure 2 hinders understanding their discussions on oxygen release. They should increase all fonts in Figure 2 larger enough to render notes readable. Additional DFT data of the total energy should be provided for all the three kinds of oxygen sites at A, B, C in ID- Li_2TMO_3 systems and the original Li_2RuO_3 as well. In the calculation of oxygen release energy, clarify what reference have been used for delithiated ID- Li_2TMO_3 system and the original Li_2RuO_3 .

Response to Comment 2

Thank you for prompting us to improve the readability of the results. The total energies of the systems with oxygen loss at the three kinds of oxygen sites in ID-Li₂RuO₃ and original R-Li₂RuO₃ systems, as well as the reference energy for the calculation of oxygen release energy for delithiated ID-Li_{2-x}RuO₃ and R-Li_{2-x}RuO₃ are listed in Table S1 and Table S2, respectively. The Gibbs free energy of oxygen release was calculated as Equation S1.

$$\Delta G_{(O_2 \text{ release})} = \Delta E_{(O_2 \text{ release})} + (-TS_{(O_2)}) + ZPE_{(O_2)} \quad (S1)$$

where $-TS_{(O_2)}$ and $ZPE_{(O_2)}$ of the O₂ gas phase under standard conditions were taken from previous studies.^[5-6] and $\Delta E_{(O_2 \text{ release})}$ are defined as Equation S2

$$\Delta E_{(O_2 \text{ release})} = 2(E_{(X-O)} + \frac{1}{2}E_{(O_2)} - E_{(X)}) \quad (S2)$$

where, $E_{(X-O)}$ and $E_{(X)}$ indicate the calculated total energy of Li_{2-x}RuO₃ with and without oxygen vacancies, respectively, where $E_{(O_2)}$ was modified by the experimental formation energy of water as the binding energy of O₂ molecules from DFT calculations is overestimated.^[4, 5] The energy of the O₂ molecule was calculated based on Equation S3 and S4. Values used in the calculation of the total energy of the O₂ molecule were shown in Table S1.

$$\Delta G = \Delta(E - TS + ZPE) = \left[E(H_2O) - E(H_2) - \frac{1}{2}E(O_2) \right] + \left[-TS(H_2O) + TS(H_2) + \frac{1}{2}TS(O_2) \right] + \left[ZPE(H_2O) - ZPE(H_2) - \frac{1}{2}ZPE(O_2) \right] \quad (S4)$$

Changes in the revised manuscript

All fonts in Figure 2 were increased. The oxygen release energy for three kinds of oxygen site in ID-Li₂RuO₃ are supplemented in Figure 2c. Detailed definition of ΔG for oxygen release are given in Supporting information, and Table S1-S3 are add in the supporting information.

Figure 2. Optimized crystal structures and local RuO₆ octahedrons of Li₂RuO₃ and the corresponding delithiated state (Li₀RuO₃) for R-Li₂RuO₃ (a) and ID-Li₂RuO₃ (b). The values (in angstrom) on the local structures are the Ru–O bond lengths and O–O distances. (c) Oxygen release energy for R-Li_{2-x}RuO₃ and ID-Li_{2-x}RuO₃ systems.

Definition of ΔG for oxygen release: The total energy of the systems with and without oxygen vacancies used in the calculation of oxygen release energy for ID-Li_{2-x}RuO₃ and R-Li_{2-x}RuO₃ are listed in Table S1 and Table S2, respectively. The same oxygen vacancy concentration of 1/48 (2% of the total oxygen content) is considered for both ID-Li_{2-x}RuO₃ and R-Li_{2-x}RuO₃. The Gibbs free energy of oxygen release was calculated as Equation S1.

$$\Delta G_{(O_2 \text{ release})} = \Delta E_{(O_2 \text{ release})} + (-TS_{(O_2)}) + ZPE_{(O_2)} \quad (S1)$$

where $-TS_{(O_2)}$ and $ZPE_{(O_2)}$ of the O₂ gas phase under standard conditions were taken from previous studies.^{1,2} and $\Delta E_{(O_2 \text{ release})}$ are defined as Equation S2

$$\Delta E_{(O_2 \text{ release})} = 2(E_{(X-O)} + \frac{1}{2}E_{(O_2)} - E_{(X)}) \quad (S2)$$

where, $E_{(x-o)}$ and $E_{(x)}$ indicate the calculated total energy of $\text{Li}_{2-x}\text{RuO}_3$ with and without oxygen vacancies, respectively, where $E_{(O_2)}$ was modified by the experimental formation energy of water as the binding energy of O_2 molecules from DFT calculations is overestimated.^{1,3} The energy of the O_2 molecule was calculated based on Equation S3 and S4. Values used in the calculation of $E_{(O_2)}$ were shown in Table S3.

$$\Delta G = \Delta(E - TS + ZPE) = \left[E(\text{H}_2\text{O}) - E(\text{H}_2) - \frac{1}{2}E(\text{O}_2) \right] + \left[-TS(\text{H}_2\text{O}) + TS(\text{H}_2) + \frac{1}{2}TS(\text{O}_2) \right] + \left[ZPE(\text{H}_2\text{O}) - ZPE(\text{H}_2) - \frac{1}{2}ZPE(\text{O}_2) \right] \quad (\text{S4})$$

Table S1. The total energies of the systems with and without oxygen loss at the three kinds of oxygen sites for ID- $\text{Li}_{2-x}\text{RuO}_3$.

Vacancy sites	$\text{O}_{\text{center}}[\text{Ru}_1]$	$\text{O}_{\text{center}}[\text{Ru}_2]$	$\text{O}_{\text{center}}[\text{Ru}_3]$
	Li ₅]	Li ₄]	Li ₃]
$\text{E}(\text{Li}_2\text{RuO}_3-$	-532.050	-531.661	-531.660
$\text{E}(\text{Li}_2\text{RuO}_3)$		-539.671	
$\text{E}(\text{Li}_{1.5}\text{RuO}_3-$	-492.821	-491.853	-491.639
$\text{E}(\text{Li}_{1.5}\text{RuO}_3)$		-499.557	
$\text{E}(\text{Li}_1\text{RuO}_3-$	-453.183	-451.212	-450.621
$\text{E}(\text{Li}_1\text{RuO}_3)$		-458.800	
$\text{E}(\text{Li}_{0.5}\text{RuO}_3-$	-405.751	-403.612	-403.285
$\text{E}(\text{Li}_{0.5}\text{RuO}_3)$		-411.281	
$\text{E}(\text{Li}_{0.25}\text{RuO}_3-$	-380.719	-379.946	-378.729
$\text{E}(\text{Li}_{0.25}\text{RuO}_3)$		-386.417	
$\text{E}(\text{Li}_0 \text{ RuO}_3-$	-355.806	-355.471	-355.043
$\text{E}(\text{Li}_0\text{RuO}_3)$		-360.711	

Table S2. The total energies of the systems with and without oxygen loss for R-Li_{2-x}RuO₃.

Vacancy sites	O _{center} [Ru ₂ Li ₄]
E(Li ₂ RuO ₃ -	-533.365
E(Li ₂ RuO ₃)	-541.172
E(Li _{1.5} RuO ₃ -	-493.482
E(Li _{1.5} RuO ₃)	-501.158
E(Li ₁ RuO ₃ -	-455.764
E(Li ₁ RuO ₃)	-463.509
E(Li _{0.5} RuO ₃ -	-406.474
E(Li _{0.5} RuO ₃)	-411.305
E(Li _{0.25} RuO ₃ -	-380.565
E(Li _{0.25} RuO ₃)	-385.533
E(Li ₀ RuO ₃ -	-355.622
E(Li ₀ RuO ₃)	-358.250

Table S3. Values used in the calculation of the total energy of the O₂ molecule. The energies of the H₂O and H₂ molecules were obtained from our first-principles study, while TS and ZPE were taken from the literature.¹

Molecule type	H ₂ O	H ₂	O ₂
E (eV)	-14.273	-6.800	-9.426 ^a
TS (eV)	0.67	0.41	0.64
ZPE (eV)	0.56	0.27	0.10

^a The E(O₂) is obtained from Equation S4.

Comment 3

I do not understand “The Rietveld refinement indicates that the long-range structure of ID-Li₂RuO₃ is consistent with that of ideal ID-Li₂RuO₃.” On page 9. What is ideal ID-Li₂RuO₃ referring to? Because they used ionic exchange method to synthesize the Ru/Li mixing layer, clarify if there is any residual Na in the sample that makes the electrochemistry different?

Response to Comment 3

The “ideal ID-Li₂RuO₃” structure was obtained by intralayer disordering the distribution of Ru and Li ion in TM layer with 66.667% Ru and 33.333% Li occupancy for all sites in TM layer based on regular Li₂RuO₃ structure from reference. The XRD pattern of “ideal ID-Li₂RuO₃” was the simulated patterns of the “ideal ID-Li₂RuO₃” structure by using Materials Studio software. The structural details from the refinement were shown in Table S4, S5. Specifically, according to refinement results of ID-Li₂RuO₃, the Ru and Li occupancies are 0.701515 (Ru) and 0.298485 (Li) at 4h site, and 0.596268 (Ru) and 0.403732 (Li) at 2d site, which are close to 0.667 (Ru) and 0.333 (Li) of the Ru and Li occupancies at both 4h and 2d site in the ideal TM/Li-intralayer disordered Li₂RuO₃. Thus, the structure of ID-Li₂RuO₃ sample was similar to the ideal intralayer disordered Li₂RuO₃. In order to evaluate the extent of intralayer disordering, two phase including regular Li₂RuO₃ and ideal intralayer disordered Li₂RuO₃ were used for refinement, which shows that the ratio of regular Li₂RuO₃ and ideal intralayer disordered Li₂RuO₃ phases is about 35: 1. The percentage of the ideal intralayer disordered Li₂RuO₃ phase is 97.1%, confirming that the ID-Li₂RuO₃ sample is almost the ideal intralayer disordered Li₂RuO₃ phase. Thus, the intralayer disordered Li₂RuO₃ was achieved successfully.

During the ionic exchange processes, 400% excess LiNO₃ was mixed with Na₂RuO₃, and the reacted in molten LiNO₃ at 280 °C for 4 h. Considering the absence of residue peaks, the excess LiNO₃ and the long heating time, we think there was no residual Na in our ID-Li₂RuO₃. Previous works demonstrated that the residue peaks resulted from Na contained compound are sensitive to the incomplete ion exchange, and such as the obvious residue peaks between 14~17°, as show in Figure R5 (copy from reference²⁴). In the XRD pattern of our ID-Li₂RuO₃ sample, no residue

peaks from Na_2RuO_3 , as shown in Figure 3a and Figure S7. Thus we consider that there are no residual Na substantially. The ICP results can estimate the element ratio more exactly. However, our sample need to be digested by alkaline NaOH as Ru cannot be digested by acid solution (even for aqua regia). Since the ICP sample preparation would undergo a NaOH alkaline digestion process, the Na would be taken into the sample during ICP test. It makes the ICP test to estimate Na content very tricky. Nevertheless, in order to detect Na content in our ID- Li_2RuO_3 sample, the ICP test was performed after an acid (aqua regia) digestion process (100 mg sample in 100 ml solution). Keeping in mind that the Ru is hard to be digested, the ICP results show that Li and Ru were detected to be 7.11 wt% (7.11 mg) and 6.82 wt% (6.82 mg) of the 100 mg sample, respectively, which are both lower than the theoretical values of 62% for Ru and 8.5% for Li based on Li_2RuO_3 . However, Na was not detected. It means that the Na content is under the minimum detectable range of 0.05 ppm, i.e., less than 0.005 mg in 100 ml solution, which is under 0.005 wt% of the 100 mg sample. Thus, there was no residual Na in our ID- Li_2RuO_3 sample.

Figure R5. XRD pattern of different lithium manganese oxides obtained by ion exchange. (A) LiMnO_2 from $\alpha\text{-NaMnO}_2$ (O3), (B) $\text{Li}_{2/3}[\text{Li}_{1/6}\text{Mn}_{5/6}]\text{O}_2$ obtained from $\text{Na}_{2/3}[\text{Li}_{1/6}\text{Mn}_{5/6}]\text{O}_2$ (P2), (C) $\text{Li}_{2/3}[\text{Li}_{1/6}(\text{Mn}_{0.82}\text{Co}_{0.18})_{5/6}]\text{O}_2$ obtained from $\text{Na}_{2/3}[\text{Li}_{1/6}(\text{Mn}_{0.82}\text{Co}_{0.18})_{5/6}]\text{O}_2$ (P2), and (D) $\text{Li}_{0.7}\text{MnO}_{2+y}$ obtained from low temperature $\text{Na}_{0.7}\text{MnO}_{2+y}$. (copy from previous work of Dahn et al.²⁴)

Changes in the revised manuscript

Descriptions on refinement results of ID-Li₂RuO₃ were given more specifically (Line 10-21, Page 11) as follows:

Specifically, according to refinement results of ID-Li₂RuO₃, the Ru and Li occupancies are 0.701515 (Ru) and 0.298485 (Li) at 4h site, and 0.596268 (Ru) and 0.403732 (Li) at 2d site, which are close to 0.667 (Ru) and 0.333 (Li) of the Ru and Li occupancies at both 4h and 2d site in the ideal TM/Li-intralayer disordered Li₂RuO₃. Thus, the structure of ID-Li₂RuO₃ sample was similar to the ideal intralayer disordered Li₂RuO₃. In order to evaluate the extent of intralayer disordering, two phase including regular Li₂RuO₃ and ideal intralayer disordered Li₂RuO₃ were used for refinement, which shows that the ratio of regular Li₂RuO₃ and ideal intralayer disordered Li₂RuO₃ phases is about 35: 1. The percentage of the ideal intralayer disordered Li₂RuO₃ phase is 97.1%, confirming that the ID-Li₂RuO₃ sample is almost the ideal intralayer disordered Li₂RuO₃ phase. Thus, the intralayer disordered Li₂RuO₃ was achieved successfully.

Comment 4

The quality of Figure 3 makes their XRD and TEM results difficult to understand at this stage.

Response to Comment 4

Thank you for your kind reminder, the quality of Figure 3 was improved in the revised manuscript. XRD refinement (Figure 3a, Table S4-5) of ID-Li₂RuO₃ sample demonstrated the intralayer disorder arrangement of Ru and Li within TM layer on long-range scale. Furthermore, Neutron powder diffraction (Figure S9, Table S4, S7) and the SAED pattern (Figure S8) confirmed the intralayer disorder arrangement from the long-range scale. HAADF-STEM images (Figure 3c-d) confirmed the disordered arrangement of the TM/Li intralayer in ID-Li₂RuO₃ on short-range scale.

Changes in the revised manuscript

The quality of Figure 3 is improved.

Figure 3. XRD patterns of ID-Li₂RuO₃ (a) and R-Li₂RuO₃ (b). The insets show the corresponding crystal models after refinement. HAADF-STEM images of the ID-Li₂RuO₃ sample along the [100] (c) and [001] (d) zone axes.

Comment 5

Figure 4 indicates that R-Li₂RuO₃ has significantly more discharge capacity 300 mAh g⁻¹ than does ID-Li₂RuO₃ at 230 mAh g⁻¹. Does this mean without oxygen dimerization, the repondance

mechanism cannot achieve high capacity in ID-Li₂RuO₃? Were all the benefits in rate and voltages in the cost of high capacity? Suggest to turn the inset of charge/discharge profiles into a full figure 4a and compare both cycling plots in figure 4b. They should also mention what the initial charge capacities were for both samples.

Response to Comment 5

Thanks for referee's kind suggestions. The layout of Figure 4 is tuned in the revised manuscript. The charge-discharge profiles are changed into a full Figure 4a, and the charge-discharge profiles of R-Li₂RuO₃ are given in Figure 4b for comparison. ID-Li₂RuO₃ delivers a specific capacity of 230 mAh g⁻¹ in the first discharge, which is larger than the theoretical capacity of 164 mAh g⁻¹, estimated through the redox reaction of Ru⁴⁺/Ru⁵⁺. The voltage platform at ~ 4.55 V for the first charge may be related with the oxygen redox as reported from previous studies. The extra capacity could be assigned to the contribution of the oxygen redox. ID-Li₂RuO₃ can deliver an initial discharge capacity of 230mAhg⁻¹ with an average discharge voltage of 3.33 V in the voltage range of 2-4.8V, while R-Li₂RuO₃ can deliver an initial discharge capacity of 289 mAhg⁻¹ with a lower average discharge of 3.24V in the voltage range of 2-4.8V. Both the midpoint and average discharge voltage of ID-Li₂RuO₃ is higher than that of R-Li₂RuO₃, as is shown in Figure 4c. The lower initial discharge capacity can be explained by the higher voltage platform of ID-Li₂RuO₃. ID-Li₂RuO₃ demonstrates a discharge capacity of 221 mAh/g with a capacity retention of 96% after 80 cycles, which are significantly higher than the 57 mAh/g discharge capacity and 20% capacity retention of R-Li₂RuO₃. In addition, the voltage decay of ID-Li₂RuO₃ based on the midpoint discharge voltages is only 0.07 V after 80 cycles, which is much less than that (1.13 V) of R-Li₂RuO₃, as shown in Figure 4d. That means the voltage decay in ID-Li₂RuO₃ is significantly suppressed. As show in Figure 4e-f, ID-Li₂RuO₃ showed a batter rate capacity with 145 mAh/g at 5C, while R-Li₂RuO₃ can deliver only 93 mAh/g at 5C. Furthermore, the capacity retention for the cycle at 0.1C after the progressive charging and discharging test was 100% and 78.8% in the ID-Li₂RuO₃ (Figure 4e) and R-Li₂RuO₃ (Figure 4f) systems, respectively, further confirming the excellent cycling stability of ID-Li₂RuO₃.

In order to compare the cycling stability of ID-Li₂RuO₃ and R-Li₂RuO₃ in the comparable initial discharge capacity, several voltage range had been tested for both R-Li₂RuO₃ and

ID-Li₂RuO₃. As is shown in Figure S11(a), capacity retentions of ID-Li₂RuO₃ in the voltage range of 2.0–4.8V, 2.0–5.0V, 1.5–4.8V are all higher than that of R-Li₂RuO₃ in the voltage range of 2.0–4.8V, 2.0–4.6V, 2.0–4.2V. The initial discharge capacity of R-Li₂RuO₃ in the voltage range of 2.0–4.2V is lower than that of the ID-Li₂RuO₃ in the voltage range of 2.0–5.0V, while the capacity retention of R-Li₂RuO₃ is still lower than ID-Li₂RuO₃ even when the capacity of the first cycle of R-Li₂RuO₃ is lower than ID-Li₂RuO₃. Moreover, the voltage decay of ID-Li₂RuO₃ and R-Li₂RuO₃ electrodes within several voltage ranges shown in Figure S11b also indicate less voltage decay in ID-Li₂RuO₃ in all cases. This is also true for comparison between higher capacity case of ID-Li₂RuO₃ and lower capacity case of R-Li₂RuO₃. Thus, the benefits in cycling stability and voltages decay are not in the cost of high capacity.

Changes in the revised manuscript

The layout of Figure 4 is tuned in the revised manuscript. The charge-discharge profiles are changed into a full Figure 4a, and the charge-discharge profiles of R-Li₂RuO₃ are given in Figure 4b for comparison. Descriptions are given in revised manuscript (Line 8-16) as follows:

The charge–discharge curves of R-Li₂RuO₃ in the voltage range of 2.0–4.8 V at a current density of 30 mA/g that agrees well with previous reports^{43,44} were given for comparison (Figure 4b), showing an initial specific discharge capacity of 289 mAh g⁻¹. The initial specific discharge capacity of ID-Li₂RuO₃ with average discharge voltage of 3.33 V is lower than that of R-Li₂RuO₃ with average discharge voltage of 3.24 V within the same voltage range of 2.0–4.8 V, which can be explained by the higher voltage platform of ID-Li₂RuO₃.

As shown in Figure S11a, the capacity retention of ID-Li₂RuO₃ is significantly higher than that of R-Li₂RuO₃ in all cases, even when the initial specific discharge capacity of ID- Li₂RuO₃ (260 mAh/g for 2.0–5.0 V) turns higher than that of R-Li₂RuO₃ (246 mAh/g for 2.0–4.2 V).

Figure 4. The comparative electrochemical performance of ID-Li₂RuO₃ and R-Li₂RuO₃. The charge–discharge profiles of ID-Li₂RuO₃ (a) and R-Li₂RuO₃ (b); (c) Cycling performance of ID-Li₂RuO₃ and R-Li₂RuO₃ in the voltage range of 2.0–4.8 V at a current density of 30 mA/g (0.1 C); (d) Midpoint discharge voltages of the ID-Li₂RuO₃ and R-Li₂RuO₃ during cycling. The progressive charging and discharging of the ID-Li₂RuO₃ (e) and R-Li₂RuO₃ (f) electrode in serial stages at various current rates from 0.1C (30 mA/g) to 5C (1500 mA/g) in the voltage range of 2.0–4.8 V.

Figure S11. Cycling performance (a) and midpoint discharge voltage (b) of ID-Li₂RuO₃ in the voltage ranges of 2.0–5.0 V, 2.0–4.8 V, and 1.5–4.8 V compared with those of R-Li₂RuO₃ in the voltage ranges of 2.0–4.6 V, 2.0–4.8 V, and 2.0–4.2 V at a current density of 30 mA/g.

Comment 6

The quality of Figure 5/6 makes it impossible to guess what oxidation occurs in Ru or O.

Response to Comment 6

Thanks for referee's suggestions. The quality of Figure 5/6 was improved. As shown in Figure 5a, during the charging process, the absorption edge of Ru continuously shifts to a higher energy below 4.3 V, indicating continuous oxidation of Ru, whereas the Ru K-edge remains unchanged when charging from 4.3 V to 4.8 V. The O K-edge XANES spectra of ID-Li₂RuO₃ in Figure 5b show a continuous increase in intensity of the first peak for the charge processes, which corresponds to the hybridization of the 2*p* orbital of O and the 4*d*-*t*_{2*g*} orbital of Ru. As no Ru oxidation occurred above ~4.3 V, this continuous increase in intensity of the O K-edge above ~4.3 V can be attributed to the anionic oxygen redox reaction. During the discharging process, the absorption edge of Ru continuously shifts back to lower energy, and O K-edge XANES spectra show a continuous decrease in intensity of the first peak. Thus, a joint of Ru and O charge compensation is occurred during discharging.

The charge variations on the Ru ions and O ions during the delithiation processes for the

R-Li₂RuO₃ and ID-Li₂RuO₃ systems obtained from Bader charge analysis are shown in Figure 5c and d, respectively. Generally, the electronic structure variations are similar for R-Li₂RuO₃ and ID-Li₂RuO₃. The average charge on the Ru ions in Li_{2-x}RuO₃ decreases for $x < 1$, then remains almost unchanged for $x > 1$. The average charge on the O ions in Li_{2-x}RuO₃ decreases with a higher slope for $x > 1$ than for $x < 1$. We conclude that Ru in Li_{2-x}RuO₃ mainly participates in charge compensation at $x < 1$, whereas charge compensation can mainly be attributed to the oxygen redox reaction at $x > 1$ in both the R-Li₂RuO₃ and ID-Li₂RuO₃ systems, which is consistent with the XAS results.

Changes in the revised manuscript

The quality of Figure 5/6 was improved.

Figure 5. *Ex situ* Ru K-edge (a) and O K-edge (b) XANES spectra of ID-Li₂RuO₃. Charge and average charge on Ru ions and O ions in R-Li₂RuO₃ (c) and ID-Li₂RuO₃ (d) with respect to the Li content.

Figure 6. (a) Voltage profiles used for in situ XRD analysis for ID-Li₂RuO₃ at a current density of 30 mA/g. (b) Contour plot of in situ XRD patterns in the range of 2θ = 16°–19°. (c) in situ XRD patterns from the direct observations. Gas evolution at a current density of 30 mA/g in the ID-Li₂RuO₃ (d) and R-Li₂RuO₃ (e) vs. Li cells from *in situ* DEMS analyses.

Comment 7

While DEMS measurements are valued techniques to detect gas generation, they cannot tell what gas generated by their own. So, they should be cautious to claim the telescopic O–Ru–O configuration to suppress oxygen release for two reasons. a) the actual effect of telescopic O–Ru–O configuration seemed suppress the dimer formation. If no oxygen dimer, there is no suppressing at all. b) the figure 6d and 6e were not exactly the same in the region of gassing. ID-Li₂RuO₃ showed higher increases after CO₂ detected at 4.1V. The charging curves were hardly comparable between 6d and 6e. So, need to clarify the difference in the gas generation. Both 6d and e indeed showed some gas generated once charging to high voltage. Be fairly enough, the 6d only showed reduced gassing in the end of charging comparing to 6e. But this cannot rule out whether it is oxygen or not without further analysis. The figure quality should be improved as well.

Response to Comment 7

In situ DEMS measurements were carried out to detect gas generation and what kind of gas is produced, similar to previous literature reports.^{5,23,25,26} From the blue curve ($m/z = 44$) in Figure 6d and 6e, it can be seen that CO₂ release occurred once the charge voltage reached 4.1 V for both ID-Li₂RuO₃ (5.600 mg active material) and R-Li₂RuO₃ (4.356 mg active material), which is similar to the DEMS results in previous reports.^{5,23,25,26} No oxygen was detected until the charge reached 4.8 volts for ID-Li₂RuO₃, as shown in the red curve in Figure 6d. Even when charged to a high voltage of 5.0 V, no oxygen is detected (Figure S19), confirming the absence of oxygen release from ID-Li₂RuO₃. However, evolution of O₂ from R-Li₂RuO₃ was observed during charging when the charge voltage approached ~ 4.2 V, as shown in the red curve in Figure 6e, which is consistent with a previous *in situ* DEMS result for R-Li₂RuO₃.⁵

Indeed, the *in situ* DEMS results show some differences in CO₂ evolution for ID-Li₂RuO₃ and R-Li₂RuO₃, but these differences are another evidence for the absence of O₂ release in ID-Li₂RuO₃. In the earlier stage of CO₂ evolution, CO₂ evolution of ID-Li₂RuO₃ was higher than R-Li₂RuO₃, while in the later stage of CO₂ evolution, CO₂ evolution of ID-Li₂RuO₃ was higher

than R-Li₂RuO₃. First, as for the earlier stage of CO₂ evolution, the average voltage of ID-Li₂RuO₃ during this stage is higher than that of R-Li₂RuO₃. The electrolyte decomposition rate would increase at a higher voltage during charging, thus the average gas generation rate of ID-Li₂RuO₃ is higher than that of R-Li₂RuO₃. Second, as for the later stage of CO₂ evolution, although the average voltage of ID-Li₂RuO₃ is still higher than that of R-Li₂RuO₃, the CO₂ evolution of ID-Li₂RuO₃ turns lower than that of R-Li₂RuO₃. Since O₂ in the cell could promote the electrolyte decomposition,²⁵ a rapid release of CO₂ would be observed once O₂ evolution rate reached a certain value.²⁵ Thus, the sharp increase of CO₂ for R-Li₂RuO₃ at ~ 4.3 V is caused by O₂ generated in the cell. In contrast, the ID-Li₂RuO₃ without O₂ release do not show sharp increase of CO₂ evolution rate. Then, the CO₂ evolution of ID-Li₂RuO₃ turns lower than that of R-Li₂RuO₃. This confirms the absence of O₂ release from ID-Li₂RuO₃.

Changes in the revised manuscript

The quality of Figure 6 was improved. Further analysis on DEMS results were added in the revised manuscript (Lines 11-15, Page 23) as follows:

In addition, a sharp increase of CO₂ generation at ~ 4.3 V for R-Li₂RuO₃ was occurred as the electrolyte decomposition was promoted by O₂ that generated in the cell once O₂ evolution reached a certain high rate, as reported previously⁴⁸. The O₂ release demonstrated here is in accordance with the irreversible XRD evolution of R-Li₂RuO₃ during charging/discharging.

Figure 6. (a) Voltage profiles used for in situ XRD analysis for ID-Li₂RuO₃ at a current density of 30 mA/g. (b) Contour plot of in situ XRD patterns in the range of $2\theta = 16^{\circ}$ – 19° . (c) in situ XRD patterns from the direct observations. Gas evolution at a current density of 30 mA/g in the ID-Li₂RuO₃ (d) and R-Li₂RuO₃ (e) vs. Li cells from in situ DEMS analyses.

Comment 8

Provide some discussions or guidance how their proposed repondance mechanism would like to work for light elements other than heavy elements like Ru in practical cathode. Will such a

mechanism only work for second row transitional metals? It seems the telescopic O–TM–O configuration unlikely stable for first row TM.

Response to Comment 8

Thank you for the good advice. Theoretically, since the structural response to oxygen redox would be alter from O–O dimerization to telescopic O–TM–O configuration when the local symmetry is tuned, as indicated in Figure 1. In order to evaluate the possibility of telescopic O–TM–O in light TM based layered material, the Li_2MnO_3 system is investigated by DFT calculation. As shown in Figure S24, the delithiated state of local symmetry tuned ID- Li_2MnO_3 also responds with telescopic O–Mn–O configuration. Thus, we preliminarily predict that the telescopic O–TM–O mechanism is also applicable for the first row light elements based layer structures. The telescopic O–TM–O configuration is related to short terminal TM–O bonds which could be stable for the first row TM row TM including Ti, V, Cr, and Mn²⁷. Thus, the telescopic O–TM–O configuration is stable for the first row light TM based Li-rich layered cathode materials. The structural response to oxygen redox would be altered from O–O dimerization to telescopic O–TM–O configuration when the local symmetry is tuned, avoiding O₂ release and thus enhancing the cycling stability of oxygen redox reaction involved charging/discharging processes in Li-rich layered cathode materials.

Changes in the revised manuscript

Figure S24 related to the prediction of the O–Mn–O from DFT calculation are supplemented. Discussions are added in revised manuscript (Lines 13-15, Page 27; Lines 1-13, Page 28) as follows:

Based on all the above results, the theoretical prediction of local symmetry tuning as a strategy to achieve a structural response of telescopic O–TM–O configuration that avoiding oxygen dimerization upon charging/discharging is confirmed in a model Li-rich layered cathode material, Li_2RuO_3 . In order to verify whether this telescopic O–TM–O mechanism works for the other cathode Li-rich layered cathode material related to first row TM, the Li_2MnO_3 system is investigated by DFT calculation. As shown in Figure S24, similar to the ID- Li_2RuO_3 system, the delithiated state of local symmetry tuned ID- Li_2MnO_3 also responds with telescopic O–Mn–O

configurations. The O–TM–O configuration is related to short terminal TM–O bond which could also be stable for the first row TM including Ti, V, Cr, and Mn⁵¹. Thus, we preliminarily predict that the telescopic O–TM–O mechanism is also applicable for the first row light TM based Li-rich layered cathode materials. The structural response to oxygen redox would be alter from O–O dimerization to telescopic O–TM–O configuration when the local symmetry is tuned, avoiding O₂ release and thus enhancing the cycling stability of oxygen redox reaction involved charging/discharging processes in Li-rich layered cathode materials.

Figure S24. Optimized crystal structures and local MnO₆ octahedrons of Li₂MnO₃ and the corresponding delithiated state (Li₀MnO₃) for ID-Li₂MnO₃. The values (in angstrom) on the local structures are the Ru–O bond lengths.

Reply to Reviewer 3

General Comment

This paper reports comparative studies on ordered and disordered Li_2RuO_3 as oxygen-redox cathode materials, which I don't recommend for publication. The main claim 'telescopic O-Ru-O' with short and long Ru-O bonds of 1.6 and 3.0 Å, which is against the classic but fundamental concept of 'ionic radius', is just a **speculation** based on the hypothetical calculations and (subjective, in my opinion) fittings of EXAFS. I could not find any convincing experimental evidence for their hypothesis to deny the fundamental concept of inorganic chemistry. The followings are my serious concerns.

Response to the General Comment

Thanks for referee's careful and serious review. We checked the validity of the calculation models again and discussed in depth with other experts in inorganic chemistry and crystallography. To further confirm the O-Ru-O configurations with short bond and long bond, we added the results of ABF-STEM measurement and EXAFS fitting in the revised manuscript. Both unambiguously confirm again the presence of the short and long Ru-O bonds in the delithated ID- Li_0RuO_3 . Therefore, we thought that the short and long Ru-O bonds is rational in ID- Li_0RuO_3 , based on the complementary results of DFT calculation, EXAFS fitting and ABF-STEM measurement. These results are also consistent with the previous reports about the length of Ru-O bond, suggesting such O-Ru-O configuration is not against the fundamental concept of inorganic chemistry.

Comment 1

Concerning the DFT part, the authors found a specific local structure, that is, short and long Ru-O bonds in disordered Li_2RuO_3 . I'm highly suspicious of such a chemically counterintuitive short and long Ru-O bonds of 1.6 and 3.0 Å, which are completely against the simple concept of ionic radius. I believe that it is necessary for the authors to re-consider the validity of the calculation

models, especially for disordered one.

Response to Comment 1

Thanks for referee's careful review and kind suggestion. We checked the validity of the calculation models again and discussed in depth with other experts in inorganic chemistry and crystallography. To further confirm the O–Ru–O configurations with short bond and long bond, we added the results of ABF-STEM measurement and EXAFS fitting in the revised manuscript. Both of them unambiguously confirm again the presence of the short and long Ru–O bonds in the delithated ID-Li₀RuO₃. Therefore, we thought that the short and long Ru–O bonds is rational in ID-Li₀RuO₃, based on the complementary results of DFT calculation, EXAFS fitting and ABF-STEM measurement. These results are also consistent with the previous reports about the length of Ru–O bond, suggesting such O-Ru-O configuration is not against the basic concept of ionic radius.

To start with, we want to reconcile the ionic radius problem. Table R1 listed the Shannon Effective Ionic Radii (copy from references^{28,29}). The ionic radius for specific element is related to the coordination number and the net charge number. For example, when the coordination number are 2 and 6, the ionic radii of O²⁻ are 1.21 Å and 1.40 Å, respectively. And the ionic radii of Ru⁴⁺ is 0.62 Å and the ionic radii of Ru⁵⁺ is 0.57 Å when the coordination number is 6. If the Ru⁵⁺–Oⁿ⁻ is pure ionic bond and n=2, the bond length is ~1.78 Å, which is larger than 1.67 Å of the proposed short bond.

However, there are three factors resulting in 'shortening' of the Ru⁵⁺–Oⁿ⁻ bond length as compared with the ~ 1.78 Å bond length. First, the ionic radius of Oⁿ⁻ resulting from the oxygen redox will be smaller than O²⁻ as the net charge is reduced²⁸. Second, the coordination number of the corresponding O ion is approximately one. The ionic radius would be smaller than two-coordinated Oⁿ⁻. Third, the covalency of the Ru–O bond increased in ID-Li₀RuO₃ based on the crystal orbital overlap population (COOP) analysis (Figure S1). The "covalent bond" effect would shorten the bonds²⁸. Thus, the Ru–O bond length should be further decreased. Table R2 listed the covalent radii of O element (copy from references^{30,31}) which are 0.63, 0.57, and 0.53 Å for single-, double-, and triple-bond cases. That means the bond length of 1.67 Å does not violate the concept of 'ionic radius'.

On the other hand, as for the short 1.67 Å Ru–O bond in the local RuO₆ octahedron of the delithiated ID-Li₀RuO₃ system, the crystal orbital overlap population (COOP) analysis was performed to study the difference of the bond interaction between short Ru–O bond in ID-Li₀RuO₃ and normal Ru–O bond in R-Li₀RuO₃, as shown in Figure S1. The integrated COOP of the short Ru–O bonds below Fermi level in ID-Li₀RuO₃ increases by 51% when compared with Ru–O bonds in R-Li₀RuO₃, implying that the net bond order of the short Ru–O bonds in ID-Li₀RuO₃ is much higher than that of Ru–O bonds in R-Li₀RuO₃. Considering the higher net bond order and the bond length of 1.67 Å that is close to the previously reported bond lengths of Ru⁵⁺=O double bond (1.63 Å,¹ 1.676 Å,² 1.697 Å,² and 1.70 Å³), this terminal Ru–O short bond can be regarded as quasi Ru⁵⁺–O double bond with π-type hybridization between Ru (t_{2g}) and O (2p). This is similar to the previous proposed Ir–O π bonds in Li₂Ir_{1-x}Sn_xO₃ system after TM ions migration to Li layer.⁴ Besides the reports about the length of Ru⁵⁺=O double bond in previous works, there are also different reports about the length of Ru⁴⁺=O bond, such as 1.705 Å,³² 1.718 Å,³² 1.732 Å³³. It can be found that the short Ru–O bonds of 1.67 Å in the delithiated ID-Li₀RuO₃ is very consistent with the length of Ru⁵⁺=O double bond in previous works. Thus, these short Ru–O bonds of 1.67 Å are rational.

As for the long Ru–O bond of 3.015 Å in the local RuO₆ octahedron, marked as Ru1-O2 in Figure R5. The 3.015 Å-long bond essentially reflects the very weak interaction between Ru and oxygen. Thus, it can also be called the distance between ruthenium Ru-ion and O-ion. Considering the Ru–O distance would reversible shorten back to range of normal Ru–O bond length of 1.9–2.0 Å during discharge processes, we call it Ru–O bonds in the whole processes of charging and discharging. The Ru–Ru distance of 3.05 Å in previous work¹⁵ was also called as bond length. As shown in Figure R5, the coordination number of the O ion in O2 site is not one. Oxygen ion in O2 site is additionally bonded with another Ru (marked as Ru2 in Figure R5), with a bond length of 1.77 Å, which is close to the bond length of ~1.78 Å Ru⁵⁺–O²⁻ simply based on the sum of ionic radii in Table R1. As discussed above, the bond lengths are affected by complex factors including the net charge on Oⁿ⁻, coordination number, and the “covalent bond” effect.²⁸ Therefore, the Ru–O bond lengths in our proposal are theoretically rational based on above discussions.

Table R1. The Shannon Effective Ionic Radii of Ru and O ions. (copy from references^{28,29})

Ion	Coordination number	Effective ionic radii (Å)
Ru ⁸⁺	4	0.36
Ru ⁷⁺	4	0.38
Ru ⁵⁺	6	0.57
Ru ⁴⁺	6	0.62
Ru ³⁺	6	0.68
O ²⁻	2	1.21
O ²⁻	6	1.40
O ²⁻	8	1.42

Table R2. The covalent radii of O element. (copy from references^{30,31})

Single-bond radius	0.63 Å
Double-bond radius	0.57 Å
Triple-bond radius	0.53 Å

Figure R5. The local structure of telescopic O–Ru–O configurations in fully delithiated state

Moreover, experimental proof is more persuasive, hence we performed ABF-STEM analysis, as shown in Figure 7a. The telescopic O–Ru–O configurations with short bond and long bond are observed directly by ABF-STEM image of 4.8V charged ID-Li₂RuO₃ (Figure 7a). It should be noted that the viewing direction is ascertained by the SEAD and FFT patterns, securing the reliability of such analysis. According to the XRD refinement results of 4.8 V charged ID-Li_{2-x}RuO₃ (Figure S16, Table S8, Table S9), the 4.8V charged ID-Li_{2-x}RuO₃ kept in C2/m space group with a beta angle of about 90°, indicating a O1-type layered structure.^{4,19} The SEAD of the O1-type structure with Li/Ru disordering arrangement obtained from XRD refinement of 4.8V charged ID-Li₂RuO₃ was simulated. The observed SEAD (Figure S20a) and FFT (Figure S20b) patterns are consistent well with the simulated SEAD of the O1-type ID-Li_{2-x}RuO₃ along the [001] zone axis (Figure S20c). Thus, the [001] zone axis is confirmed. The theoretical atomic structure along the [001] zone axis is shown in Figure 7d and e. Within the ABF-STEM image (Figure 7a), TM atoms appear as dark black dots, and oxygen and lithium atoms appear as light black dots. There are regular honeycomb domains, Li/vacancy concentrated domains, and Ru concentrated domains. If the structural response of the charged ID-Li_{2-x}RuO₃ behaves in a similar manner with the R-Li₂TMO₃, i.e., O–O dimerization which have been demonstrated by ABF image, and Raman spectroscopy previously^{19,22}, we should observe it directly from the Ru–O arrangement along the [001] zone axis that is schematically presented in Figure 7e, where the Ru–O bond are rotated slightly with six equal projected distances, with the O–O dimerization being nicely visualized. However, the ABF image of the charged ID-Li₂RuO₃ shows a very different projected Ru–O arrangement with the R-Li₂TMO₃ case. The projected distances of the Ru–O bonds along b1, b2, and b3 directions (marked with white dotted arrows) were evaluated by the gray value of the ABF image, as shown in Figure 7b (b1–b3). The corresponding projected Ru–O distances of the red hexagon marked RuO₆ are shown in Figure 7c, where the two Ru–O projected distances along the b1 and b2 directions are not equal, and the two Ru–O projected distances along the b3 direction are equal. Therefore, the asymmetric Ru–O bonds with specific O–Ru–O configuration around the Ru ions are observed, in contrast to the symmetric Ru–O bonds with O–O dimerization that would take place in R-Li₂RuO₃. The telescopic O–Ru–O configuration of ID-Li₂RuO₃ was visualized by ABF image.

Furthermore, the Figure S21, S22, S23 show the fitting results for the magnitude of the

Fourier transforms performed on k^2 -weighted EXAFS oscillations of both the R-Li₂RuO₃ and ID-Li₂RuO₃ during the first charge and discharge processes. Based on the presence of two crests in the Ru K-edge XANES spectra shown in Figure 5a, two group of Ru–O bonds were considered during fitting for both the R-Li₂RuO₃ and ID-Li₂RuO₃. As for R-Li₂RuO₃, the variation in the Ru–O shell from the fitting results (Figure S21) is given in Figure S23b with the detailed values listed in Table S10. Generally, the Ru–O bond length decreased during charging then increased during discharging. The total coordination number of the Ru–O bonds dramatically decreased when charged to high voltage. However, as shown in Figure 7g, the total coordination number of the Ru–O shell was not recovered to the pristine during the discharge process, indicating that the structural variation is irreversible during charge and discharge processes. This irreversible coordination number is related to O₂ release during charging, as is demonstrated by in situ DEMS measurement in Figure 6e.

As for the ID-Li₂RuO₃, the variation in the Ru–O shell from the fitting results (Figure S22) is given in Figure S23b with the detailed values listed in Table S11. Generally, the Ru–O bond length decreased during charging then increased during discharging. Furthermore, the coordination number of the long bonds dramatically decreased whereas that of the short bonds increased slightly during charging from 4.3 V to 4.8 V. We infer that a small portion of the long bonds was shortened and some long bonds were stretched to such an extent that the stretched bonds were no longer counted as part of the first Ru–O shell. Furthermore, the difference between two group of Ru–O bond length is much larger than that in R-Li₂RuO₃, showing more inhomogeneous Ru–O bond lengths. Thus, the telescopic O–Ru–O configuration, including both shortened and stretched portions, occurs in response to the oxygen redox reaction during the charge process, which agrees well with the results of the DFT calculation and ABF image. More importantly, unlike the irreversible coordination variation in R-Li₂RuO₃, the total coordination number of the Ru–O shell for ID-Li₂RuO₃ (Figure 7h) was recovered during the discharge process, indicating that the telescopic O–Ru–O configuration is reversible. This structural response in ID-Li₂RuO₃ based on the reversible telescopic O–Ru–O configuration show absence of O₂ release during charging, as demonstrated by in situ DEMS measurement in Figure 6d, which is responsible for the enhanced cycling stability of ID-Li₂RuO₃.

Changes in the revised manuscript

COOP analysis (Figure S1), ABF-STEM image (Figure 7a), and the EXAFS fitting of the control group R-Li₂RuO₃ (Figure S21, S23b, Table S10) were supplemented. Corresponding discussions are given in revised manuscript (Line 2-13, Page 8; Line 1-22, Page 24; Line 1-9, Page 25; Line 18-22, Page 25; Line 1-22, Page 26; Line 1-3, Page 27) as follows:

As for the short Ru–O bonds, the crystal orbital overlap population (COOP) analysis was performed to study the interaction between Ru and O, as shown in Figure S1. The integrated COOP of the short Ru–O bonds in ID-Li₀RuO₃ below Fermi level increases by 51% when compared with Ru–O bonds in R-Li₀RuO₃, implying that the net bond order of the short Ru–O bonds in ID-Li₀RuO₃ is higher than that of Ru–O bonds in R-Li₀RuO₃. Considering the higher net bond order and the bond length of 1.67 Å that is close to the previously reported bond lengths of Ru⁵⁺=O double bond (1.63 Å,³⁹ 1.676 Å,⁴⁰ 1.697 Å,⁴⁰ and 1.70 Å⁴¹), this terminal Ru–O short bond can be regarded as quasi Ru⁵⁺=O double bond with a π -type hybridization between with Ru (t_{2g}) and O ($2p$). This is similar to the previous proposed Ir–O π bonds in Li₂Ir_{1-x}Sn_xO₃ system after TM ions migration to Li layer.³⁶

In order to reveal the structural evolution on the local-range scale, annular bright-field scanning transmission electron microscopy (ABF-STEM) image of 4.8 V charged ID-Li₂RuO₃ along [001] zone axis was obtained (Figure 7a). It should be noted that the viewing direction is ascertained by the SAED and FFT patterns (Figure S20a-b), securing the reliability of such analysis. Based on the structure model of a O1-type layered structure with a space group of C2/m obtained from the XRD refinement of 4.8 V charged ID-Li_{2-x}RuO₃ as mentioned above, the theoretical SAED patterns are simulated (Figure S20c). The observed SAED (Figure S20a) and FFT Patterns (Figure S20b) are consistent well with the simulated SAED of this O1-type ID-Li_xRuO₃ along the [001] zone axis (Figure S20c). Thus, the [001] zone axis is confirmed. The theoretical atomic structure along the [001] zone axis is shown in Figure 7d and e. Within the ABF-STEM image (Figure 7a), Ru ions appear as dark black dots, and oxygen and lithium ions appear as light black dots. There are regular honeycomb domains, Li/vacancy concentrated domains, and Ru concentrated domains, as marked in Figure 7a. If the structural response of the charged ID-Li₂RuO₃ behaves in a similar manner with the R-Li₂TMO₃, i.e., O–O dimerization which have been demonstrated by ABF image, and Raman spectroscopy previously,^{12,14} we should

observe it directly from the Ru–O arrangement along the [001] zone axis that is schematically presented in Figure 7e, where the Ru–O bond are rotated slightly with six equal projected distances, with the O–O dimerization being nicely visualized. However, the ABF image of the charged ID-Li₂RuO₃ shows a very different projected Ru–O arrangement when compared with the R-Li₂TMO₃ case. The projected distances of the Ru–O bonds along b1, b2, and b3 directions (marked with white dotted arrows) were evaluated by the gray value of the ABF image, as shown in Figure 7b (b1–b3). The corresponding projected Ru–O distances of the red hexagon marked RuO₆ are shown in Figure 7c, where the two Ru–O projected distances along the b1 and b2 directions are not equal, and the two Ru–O projected distances along the b3 direction are equal. Therefore, the inhomogeneous Ru–O bonds with specific O–Ru–O configuration around the Ru ions are observed, in contrast to the homogeneous Ru–O bonds with O–O dimerization that would take place in R-Li₂RuO₃. Thus, the telescopic O–Ru–O configuration of ID-Li₂RuO₃ was visualized by ABF image.

The magnitude of the Fourier transform of the k^2 -weighted extended X-ray absorption fine structure (EXAFS) oscillations, $|\chi(R)|$, along with the fitting results of ID-Li₂RuO₃ and R-Li₂RuO₃ are both given for comparison. Based on the presence of two crests in the Ru K-edge XANES spectra shown in Figure 5a, two group of Ru–O bonds were considered during fitting. The variation in the Ru–O shell from the fitting results of R-Li₂RuO₃ (Figure S21) is given in Figure S23a with the detailed values listed in Table S10. The Ru–O bond length decreases during charge process then increased during discharge process. The total coordination number of the Ru–O bonds dramatically decreased when charged to high voltage (4.1–4.6 V). However, the total coordination number of the first Ru–O shell was not recovered to the pristine during the discharge process (Figure 7g), indicating that the structural variation is irreversible during charge and discharge processes. This irreversible coordination number might be related to O₂ release during charging, which is inconsistent with the irreversible XRD and *in situ* DEMS results. In contrast, the fitting results of ID-Li₂RuO₃ show a reversible variation, as shown in Figure S22, S23a and Table S11. Generally, the Ru–O bond length decreased during charging then increased during discharging. The coordination number of the long bonds dramatically decreased whereas that of the short bonds increased slightly during charging from 4.3 V to 4.8 V. We infer that a small portion of the long bonds was shortened and some long bonds were stretched to such an extent

that the stretched bonds were no longer counted as part of the first Ru–O shell. Furthermore, as shown in Figure S23a and b the difference between two group of Ru–O bond length is much larger than that in R-Li₂RuO₃, showing more inhomogeneous Ru–O bond lengths. Thus, the telescopic O–Ru–O configuration, including both shortened and stretched portions, occurs in response to the oxygen redox reaction during the charge process, which agrees well with the results of the DFT calculation, ABF-STEM image. The total coordination number of the first Ru–O shell was recovered during the discharge process (Figure 7h), indicating that the telescopic O–Ru–O configuration is reversible. As is mentioned above, this structural response based on the reversible telescopic O–Ru–O configuration is responsible for the enhanced cycling stability of ID-Li₂RuO₃.

Figure S1. The crystal orbital overlap population (COOP) of the short Ru–O bond in ID-Li₀RuO₃ and normal Ru–O bond in R-Li₀RuO₃.

Figure 7. The local structure of ID-Li₂RuO₃ and R-Li₂RuO₃ upon charging and discharging. (a) The ABF-STEM image of 4.8 V charged ID-Li₂RuO₃ along [001] zone axis; (b) The gray value variation of ABF-STEM image along b1, b2, and b3 directions (marked with white dotted arrows); (c) the enlarged image of the of the red hexagon marked RuO₆, the value between dark black dot and light black dot are the corresponding projected Ru–O distances; (f) The Raman spectra of 4.8 V charged ID-Li₂RuO₃ and R-Li₂RuO₃; The variation of total coordination number of R-Li₂RuO₃ (g) and ID-Li₂RuO₃ (h) during charge and discharge processes, obtain from EXAFS fitting.

Figure S21. Fitting results for the magnitude of the Fourier transforms performed on k^2 -weighted EXAFS oscillations of $R\text{-Li}_2\text{RuO}_3$ during the first charge (a) and discharge (b) processes in the R -range of 1–2 Å (first peak only). E_0 was set to -0.7 eV, and the amplitude reduction factor (S_0^2) was fixed at 0.95.

Figure S23. Fitted Ru–O shell of the Fourier-transformed k^2 -weighted EXAFS oscillations of $\text{ID-Li}_2\text{RuO}_3$ (a), and $R\text{-Li}_2\text{RuO}_3$ (b).

Table S10. Detailed fitting results (coordination number (CN), radial distance (R), and Debye-Waller factor (σ^2) for the first shell) for the EXAFS oscillations of R-Li₂RuO₃ during the first charge and discharge processes. E_0 was set to -0.7 eV, and the amplitude reduction factor (S_0^2) was fixed at 0.95.

Charge of state	Pair	CN	R (Å)	σ^2 ($\times 10^{-3}$ Å ²)
Pristine	Ru-O1	1.8	1.97	4.5
	Ru-O2	4.0	2.01	4.2
1C 3.7 V	Ru-O1	2.8	1.94	3.0
	Ru-O2	3.0	1.98	3.4
1C 4.1 V	Ru-O1	2.8	1.94	4.5
	Ru-O2	3.0	1.97	4.5
1C 4.6 V	Ru-O1	1.6	1.94	6.8
	Ru-O2	3.0	1.97	7.0
1D 3.3 V	Ru-O1	1.6	1.96	4.0
	Ru-O2	3.3	1.98	4.3
1D 2.0 V	Ru-O1	1.6	1.98	4.1
	Ru-O2	3.3	2.01	4.3

Comment 2

Figure 3: Please confirm the disordered and ordered cation arrangements of two materials using SAED rather than HAADF-STEM. More importantly, Figure 3d shows ordering of Li and Ru within the Ru layer, which is not consistent with the claim 'the intralayer disordered Li₂RuO₃ was

achieved successfully.'. In this situation, Figure 3a (the XRD pattern for ID-Li₂RuO₃ shows no superstructure peaks) is not reasonable.

Response to Comment 2

Thanks for referee's kind suggestions. HAADF-STEM image can give direct visualization of the local-range Ru/Li arrangement. The HAADF image of ID-Li₂RuO₃ sample along the [100] zone axis (Figure 3c) holds regular domains characterized by a periodic arrangement with one dark spot followed by two bright dots, Li concentrated domains with continuous dark spots and Ru concentrated domains with continuous bright dots, indicating TM/Li-intralayer disorder in the transition metal layer. Moreover, the HAADF image of ID-Li₂RuO₃ sample along the [001] zone axis (Figure 3d) also shows regular honeycomb domains, Li concentrated domains, and Ru concentrated domains, respectively. Thus, the HAADF images confirmed the disordered arrangement of the TM/Li intralayer on short-range scale in the as-prepared ID-Li₂RuO₃ sample

Also, the intralayer disorder arrangement of Ru and Li within TM layer are confirmed based on the XRD patterns, NPD patterns, SAED patterns, respectively. The SAED patterns are able to characterize the intralayer disordering on the long-range scale. The observed SAED patterns of ID-Li₂RuO₃ along [001] and [100] zone axes that characterized with the marked weaker diffraction spots (red cycles) are shown in Figure S8a and b, respectively, which are consistent with the simulated SEAD pattern of ID-Li₂RuO₃ with intralayer disorder arrangement of Ru and Li within TM layer along [001] (Figure S8c) and [100] (Figure S8d) zone axes. Therefore, the intralayer disordering is verified by SAED patterns on long-range scale. XRD and NPD patterns are also the convincing method to characterize the structure in the long-range scale. The XRD (Figure 3a, Table S4 and Table S5) and NPD (Figure S9, Table S4, and S7) refinement results for ID-Li₂RuO₃ both show Ru/Li-intralayer disordering.

Thus, the intralayer disorder arrangement of Ru and Li within TM layer are confirmed from both the long-range scale and local-range scale by XRD, NPD, SAED patterns, and HAADF-STEM images.

Changes in the revised manuscript

SAED patterns (Figure S8) were supplemented. Corresponding discussions were added in revised manuscript (Line 14-22, Page 8) as follows:

The observed and simulated selected area electron diffraction (SAED) patterns (Figure S8) were also given to analyze the structure on long-range scale. The ID-Li₂RuO₃ and R-Li₂RuO₃ structures with C2/m space group used for SAED simulation are taken from the XRD refinements. The observed SAED patterns of the as-prepared ID-Li₂RuO₃ sample shown in Figure S8a and b that characterized with the marked weaker diffraction spots (red circles) are consistent with the simulated SAED patterns of ID-Li₂RuO₃ structure model along [100] (Figure S8c) and [001] (Figure S8d) zone axis, respectively. Therefore, the intralayer disordering is verified by SAED patterns on long-range scale.

Figure S8. The observed SAED patterns (a-b) and simulated SAED patterns of ID-Li₂RuO₃ and R-Li₂RuO₃ structure models along [100] (c-d) and [001] (e-f) zone axes, respectively. The ID-Li₂RuO₃ and R-Li₂RuO₃ structure models with C2/m space group used for SAED simulation are taken from the XRD refinements.

Comment 3

Please compare the charge-discharge curves of ID-Li₂RuO₃ and R-Li₂RuO₃ in Figure 4, rather than only focusing on the cycle stability. I believe that the comparison of dQ/dV plots would also be of interest to readers.

Response to Comment 3

Thank you for the good suggestions. The charge-discharge curves and dQ/dV plots of ID-Li₂RuO₃ and R-Li₂RuO₃ are compared in Figure 4a-b and Figure S10 in the revised manuscript, respectively. The charge-discharge curves of ID-Li₂RuO₃ and R-Li₂RuO₃ was tested by galvanostatic discharge-charge in the voltage range of 2.0–4.8 V at a current density of 30 mA/g, as shown in Figure 4a and 4b, respectively. ID-Li₂RuO₃ delivers a specific capacity of 230 mAh g⁻¹ in the first discharge, which is larger than the theoretical capacity of 164 mAh g⁻¹, estimated through the redox reaction of Ru⁴⁺/Ru⁵⁺. The extra capacity could be assigned to the contribution of the oxygen redox. The charge-discharge curves of R-Li₂RuO₃ is consistent well with the previous work.^{11,34} The charge-discharge curves of ID-Li₂RuO₃ largely differs from that of R-Li₂RuO₃. Generally, there are two stages and three stages in the initial charge processes for ID-Li₂RuO₃ and R-Li₂RuO₃ respectively. Besides, the voltage of ID-Li₂RuO₃ are higher than that of R-Li₂RuO₃ in general, which can be seen clearly in dQ/dV curves (Figure S10).

Changes in the revised manuscript

The charge-discharge curves and dQ/dV plots of ID-Li₂RuO₃ and R-Li₂RuO₃ were compared in Figure 4a-b and Figure S10, and corresponding discussions were given in the revised manuscript (Line 2-17) as follows:

The electrochemical performance of the ID-Li₂RuO₃ was tested by galvanostatic charge-discharge in the voltage range of 2.0–4.8 V at a current density of 30 mA/g, as shown in Figure 4a. It delivers a specific capacity of 230 mAh g⁻¹ in the first discharge, which is larger than the theoretical capacity of 164 mAh g⁻¹, estimated through the redox reaction of Ru⁴⁺/Ru⁵⁺. The voltage platform at ~ 4.55 V for the first charge may be related with the oxygen redox as reported from previous studies. The extra capacity could be assigned to the contribution of the oxygen redox. The charge-discharge curves of R-Li₂RuO₃ in the voltage range of 2.0–4.8 V at a current density of 30 mA/g that agrees well with previous reports^{43,44} were given for comparison (Figure 4b), showing an initial specific discharge capacity of 289 mAh g⁻¹. The initial specific discharge capacity of ID-Li₂RuO₃ with average discharge voltage of 3.33 V is lower than that of R-Li₂RuO₃ with average discharge voltage of 3.24 V within the same voltage range of 2.0–4.8 V, which can

be explained by the higher voltage platform of ID-Li₂RuO₃. Indeed, the dQ/dV curves (Figure S10) indicate that charge and discharge voltage platform of ID-Li₂RuO₃ are both higher than that of R-Li₂RuO₃. Figure 4c compare the cycling performance of the ID-Li₂RuO₃ and R-Li₂RuO₃ electrodes. ID-Li₂RuO₃ demonstrates a discharge capacity of 221 mAh/g with a capacity retention of 96% after 80 cycles, which are significantly higher than the 57 mAh/g discharge capacity and 20% capacity retention of R-Li₂RuO₃.

Figure 4. The comparative electrochemical performance of ID-Li₂RuO₃ and R-Li₂RuO₃. The charge–discharge profiles of ID-Li₂RuO₃ (a) and R-Li₂RuO₃ (b); (c) Cycling performance of ID-Li₂RuO₃ and R-Li₂RuO₃ in the voltage range of 2.0–4.8 V at a current density of 30 mA/g (0.1 C); (d) Midpoint discharge voltages of the ID-Li₂RuO₃ and R-Li₂RuO₃ during cycling. The progressive charging and discharging of the ID-Li₂RuO₃ (e) and R-Li₂RuO₃ (f) electrode in serial stages at various current rates from 0.1C (30 mA/g) to 5C (1500 mA/g) in the voltage range of 2.0–4.8 V.

Figure S10. The dQ/dV plots of ID- Li_2RuO_3 and R- Li_2RuO_3 cathodes tested in the voltage range of 2.0–4.8 V at a current density of 30 mA/g.

Comment 4

P13 'This behavior differs from that for the reductive coupling mechanism induced by the formation of O–O dimers in R- Li_2RuO_3 , meaning that O–O dimers (O_2^{n-}) may not be formed in ID- Li_2RuO_3 .' I'm not convinced by this speculation. Whether O–O dimer is formed or not is apparently the key question in this work. Therefore, it is mandatory for the authors to prove it by experiments.

Response to Comment 4

In response to this comment, Raman analysis was performed to confirm whether O–O dimer is formed or not, as it is sensitive to O–O peroxo vibrations. The Raman spectra of R- Li_2RuO_3 and ID- Li_2RuO_3 that charged to 4.8 V were obtained with excitation light of a He-Ne laser at 633 nm wavelength, as shown in Figure 7f. The Raman stretch of O–O dimer (O_2^{n-}) at 847 cm^{-1} (previous reported to be $\sim 850\text{ cm}^{-1}$ in charged cathode material²²) was observed in charged R- Li_2RuO_3 sample while not in charged ID- Li_2RuO_3 . Hence, unlike the R- Li_2RuO_3 , the O–O dimerization didn't occur in ID- Li_2RuO_3 during charge, coinciding with our prediction.

Besides, as presented before (Response to Comment 1) in Figure 7, the ABF-STEM analysis gives a clear evidence that cooperative O-O dimerization that happens in R-Li₂RuO₃ is absent in ID-Li₂RuO₃, whereas a structural response associated with telescopic O-Ru-O configuration is observed.

Furthermore, Ru K-edge X-ray absorption near edge structure (XANES) spectra of R-Li₂RuO₃ (Figure S12) and ID-Li₂RuO₃ (Figure 5a) are given for further information in terms of the O-O coupling. As for R-Li₂RuO₃, the absorption edge shifts to a higher energy level during the earlier stage of charging, well corresponding to the change of Ru⁴⁺ oxidation to Ru⁵⁺. However, instead of shifting further, the absorption edge shifts back gradually to a lower energy level during later stage of charging. Such abnormal behavior is called as a reductive coupling mechanism (RCM), as reported previously for Li₂Ru_{0.75}Sn_{0.25}O₃ and regular Li₂RuO₃ material,^{35,36} which is known as a process where anionic redox is triggered that O ions are oxidized and structurally accommodated by O-O dimerization. However, for ID-Li₂RuO₃, the Ru K-edge shifts to a higher energy without shifting back during charging, showing the absence of RCM and thus O-O dimerization in ID-Li₂RuO₃.

In addition, *in situ* DEMS measurement (Figure 6d and e) shows no O₂ gas generation during charge process of ID-Li₂RuO₃ (even for charge voltage of 5.0 V) in opposite to R-Li₂RuO₃ that has substantial O₂ release. Considering that in both structures the first charge delivered extra capacities that calls for both oxidations of Ru and oxygen, there is no reason to have ID-Li₂RuO₃ to be free of oxygen release while R-Li₂RuO₃ isn't. The only possible explanation could be that ID-Li₂RuO₃ has a different oxygen redox behavior from O-O dimerization which is highly prone to form O₂ gas.

Overall, we demonstrate that the complementary evidences of Raman, ABF-STEM, XANES, and DEMS can convincingly prove ID-Li₂RuO₃ is free of O-O dimerization during its oxygen redox.

Changes in the revised manuscript

Raman spectra (Figure 7f), ABF-STEM (Figure 7a), Ru K-edge X-ray absorption near edge structure (XANES) spectra of R-Li₂RuO₃ served as control group (Figure S12) were supplemented and corresponding discussions were given in the manuscript (Line 10-17, Page 25; Line 19-22,

Page 16; Line 1-8, Page 17) as follows:

Raman analysis was also performed to confirm the structural response mode. The Raman spectra of the 4.8V charged ID-Li₂RuO₃ and R-Li₂RuO₃ were obtained with excitation light of a He-Ne laser at 633 nm wavelength, as shown in Figure 7f. The Raman stretch of O–O dimer (O₂)^{•-} at 847 cm⁻¹ (in accordance with ~ 850 cm⁻¹ reported previously¹⁴) was observed in charged R-Li₂RuO₃ sample while not in charged ID-Li₂RuO₃. Hence, unlike the R-Li₂RuO₃, the O–O dimerization didn't occur in ID-Li₂RuO₃ during charge process, coinciding with our prediction from DFT calculation and Ru K-edge XANES spectra.

Changes in the Ru oxidation state in ID-Li₂RuO₃ were determined by examining the ex situ X-ray absorption near edge structure (XANES) spectra of the Ru K-edge, as shown in Figure 5a. The Ru K-edge continuously shifts to a higher energy below 4.3 V, indicating continuous oxidation of Ru, whereas the Ru K-edge remains unchanged when charging from 4.3 V to 4.8 V. This behavior differs from the Ru K-edge XANES spectra of R-Li₂RuO₃ (Figure S12). R-Li₂RuO₃ presents a shift of absorption edge back to lower energy at the end charging (4.1–4.6 V), i.e., the reductive coupling mechanism (RCM), as reported previously for Li₂Ru_{0.75}Sn_{0.25}O₃ and regular Li₂RuO₃ material,^{35,38} which is known as a process where anionic redox is triggered that O ions are oxidized and structurally accommodated by O-O dimerization. However, for ID-Li₂RuO₃, the Ru K-edge shifts to a higher energy without shifting back during charging, showing the absence of RCM and thus O-O dimerization in ID-Li₂RuO₃.

Figure 7. The local structure of ID-Li₂RuO₃ and R-Li₂RuO₃ upon charging and discharging. (a) The ABF-STEM image of 4.8 V charged ID-Li₂RuO₃ along [001] zone axis; (b) The gray value variation of ABF-STEM image along b1, b2, and b3 directions (marked with white dotted arrows); (c) the enlarged image of the of the red hexagon marked RuO₆, the value between dark black dot and light black dot are the corresponding projected Ru–O distances; (f) The Raman spectra of 4.8 V charged ID-Li₂RuO₃ and R-Li₂RuO₃; The variation of total coordination number of R-Li₂RuO₃ (g) and ID-Li₂RuO₃ (h) during charge and discharge processes, obtain from EXAFS fitting.

Figure S12. Ru K-edge XANES spectra of R-Li₂RuO₃ during charge and discharge processes.

Comment 5

Figure 6: Please compare the difference of the phase evolution of R-Li₂RuO₃ and ID-Li₂RuO₃, and explain why the differences occur. Then, the authors would be able to discuss the origin of the better cycle stability of ID-Li₂RuO₃. The present Figure 6 and relating part only report 'results'.

Response to Comment 5

The phase evolution of both ID-Li₂RuO₃ and R-Li₂RuO₃ are analyzed from XRD patterns. According to the refinement of XRD pattern of the 4.8 V charged ID-Li₂RuO₃, we find that ID-Li₂RuO₃ kept in C2/m phase with lattice parameter changed during delithiation, as shown in Figure S16, Table S8, and Table S9. The β was changed from 108.5870° to 90.0097°, indicating

that the layered structure was altered from O3- to O1-type C2/m phase.^{4,19} As shown clearly in Figure S17, the phase changed from O3- to O1-type structure gradually during charge process, then almost return back to O3-type structure of the pristine during discharge process. In addition, the migration of Ru to Li layer is almost absent according to the refinement. Hence, the long-range structure of ID-Li₂RuO₃ is reversible during charge and discharge processes. More importantly, O₂ release was not occurred, demonstrated by *in situ* DEMS measurement (Figure 6d).

In contrast, the R-Li₂RuO₃ undergoes an irreversible phase transition, as shown in Figure S18. The XRD patterns variation of our R-Li₂RuO₃ during charge and discharge processes are similar to the results that reported by Inaguma et al.¹¹ As revealed by Inaguma et al., the structure changed from C2/c (or C2/m) phase to a mixed phase of R $\bar{3}$ and C2/c when charged to 3.8 V, then the structural transition with oxygen evolution occurs when further charged to 4.8 V, and the corresponding structure is unknown.¹¹ Similar to the reference,¹¹ the structure of R-Li₂RuO₃ cannot be recovered to the pristine during discharge processes. The crystallinity is lowered after the first charge-discharge cycle. Hence, the long-range structure of R-Li₂RuO₃ is irreversible during charging and discharging, which should be related to the oxygen evolution and TM migration. Indeed, the *in situ* DEMS result of R-Li₂RuO₃ (Figure 6e) show considerable O₂ release, in accordance with the irreversible processes revealed by XRD patterns, resulting in a poor cycling stability.

Combining the results of the XRD patterns and DEMS, the main difference of phase transition between ID-Li₂RuO₃ and R-Li₂RuO₃ is reversibility. The irreversible phase transition of R-Li₂RuO₃ is induced by the structural degradation associated with oxygen loss. In contrast, the phase transition of ID-Li₂RuO₃ is reversible with no oxygen loss occurred. The origin of these difference is related to the local structural response mode of O–O dimerization and telescopic O–Ru–O configuration for R-Li₂RuO₃ and ID-Li₂RuO₃, respectively. Therefore, ID-Li₂RuO₃ shows much better cycling stability than R-Li₂RuO₃.

Figure R6. XRD patterns for Li_2RuO_3 during charging and discharging. 1C, 1D, 2C and 2D represent 1st charge and discharge, 2nd charge and discharge, respectively. (copy from reference¹¹)

Figure R7. Schematic drawings of charge–discharge reaction model for Li_2RuO_3 upon the initial cycling. (copy from reference¹¹)

Changes in the revised manuscript

Comparative *ex situ* XRD analysis of ID- Li_2RuO_3 (Figure S16, S17, Table S8, S9) and R- Li_2RuO_3 (Figure S18) during charging/discharging was supplemented, corresponding discussions were given in the manuscript (Line 12-22, Page 20; Line 1-12, Page 21). Changes are as follows:

According to the refinement of XRD pattern of the 4.8 V charged ID- Li_2RuO_3 , we find that ID- Li_2RuO_3 kept in C2/m phase with lattice parameter changed during delithiation, as shown in

Figure S16, Table S8 and S9. The β was changed from 108.5870° to 90.0097° , indicating that the layered structure was altered from O3- to O1-type C2/m phase.^{12,36} As shown clearly in Figure S17, the phase changed gradually from O3- to O1-type structure during charge process, then almost returned back to O3-type structure of the pristine during discharge process. Hence, the long-range structure of ID-Li₂RuO₃ is reversible during charge and discharge processes. In addition, the migration of Ru to Li layer is almost absent according to the XRD refinement as the occupancies of Ru in Li layer are about 0.023% and 0.025% of the total Li site in Li layer for pristine and charged (4.8 V) ID-Li_{2-x}RuO₃, respectively, which is consistent with the results of the formation energy of Ru anti-site defects (Figure S5). In contrast, the R-Li₂RuO₃ undergoes an irreversible phase transition, as shown in Figure S18. The XRD patterns variation of our R-Li₂RuO₃ during charge and discharge processes are similar to the results that reported by Inaguma et al.⁴³ As revealed by Inaguma et al., the structure changed from C2/c phase to a mixed phase of R $\bar{3}$ and C2/c when charged to 3.8 V, then the structural transition with oxygen evolution occurs when further charged to 4.8 V, and the corresponding structure is unknown.⁴³ Similar to the reference,⁴³ the structure of R-Li₂RuO₃ cannot be recovered to the pristine case during discharge processes. In short, the long-range structure of ID-Li₂RuO₃ is reversible during charge and discharge processes, in contrast to the irreversible processes of ID-Li₂RuO₃, resulting in better cycling stability.

Figure S16. The refinement of XRD patterns of ID-Li₂RuO₃ that charged to 4.8 V.

Table S8. Crystallographic parameters and structure determination details for 4.8V charged ID-Li₂RuO₃ from XRD refinement.

Sample	4.8Vcharged ID-Li ₂ RuO ₃
Space group	C2/m (No. 12)
a (Å)	5.037718
b (Å)	8.718863
c (Å)	4.712646
$\alpha = \gamma$ (°)	90.000
β (°)	90.0097
Volume (Å ³)	206.99
R _{wp} (%)	4.22
R _p (%)	3.22
χ^2	4.757

Table S9. Atomic coordinates of 4.8 V charged ID-Li₂RuO₃ from XRD refinement.

Atom	Site	Coordinates			Occupation	Uiso (Å ²)
		x	y	z		
Ru(1)	4h	0.000000	0.168110	0.500000	0.701547	0.045805
Li(1)	4h	0.000000	0.168110	0.500000	0.030000	0.039030
Li(2)	2d	-0.500000	0.000000	0.500000	0.030000	0.039030
Ru(2)	2d	-0.500000	0.000000	0.500000	0.596161	0.047284
Li(3)	4g	-0.500000	-0.158000	0.000000	0.100000	0.038628
Ru(3)	4g	-0.500000	-0.158000	0.000000	0.000254	0.045635
Li(4)	2a	0.000000	0.000000	0.000000	0.100000	0.038670
Ru(4)	2a	0.000000	0.000000	0.000000	0.000237	0.048176
O(1)	8j	0.330751	0.170456	0.273423	1.000000	0.039911
O(2)	4i	-0.150312	0.000000	0.276555	1.000000	0.037253

Figure S17. The *ex situ* XRD patterns of ID-Li₂RuO₃. 1C, 1D represent the first charge and discharge, respectively.

Figure S18. The *ex situ* XRD patterns of R-Li₂RuO₃ during charging and discharging. 1C, 1D represent the first charge and discharge, respectively.

References

- 1 Sun, X., Zhou, S., Yue, L., Schlangen, M. & Schwarz, H. Thermal Activation of CH₄ and H₂ as Mediated by the Ruthenium Oxide Cluster Ions [RuO_x](+) (x=1-3): On the Influence of Oxidation States. *Chemistry* **25**, 3550-3559, doi:10.1002/chem.201806187 (2019).
- 2 Dengel, A. C., Griffith, W. P., O'Mahoney, C. A. & Williams, D. J. A stable ruthenium(V) oxo complex. X-Ray crystal structure and oxidising properties of tetra-n-propylammonium bis-2-hydroxy-2-ethylbutyrate(oxo)-ruthenate(V). *Journal of the Chemical Society, Chemical Communications*, 1720-1721, doi:10.1039/C39890001720 (1989).
- 3 Moonshiram, D. *et al.* Structure and electronic configurations of the intermediates of water oxidation in blue ruthenium dimer catalysis. *J Am Chem Soc* **134**, 4625-4636, doi:10.1021/ja208636f (2012).
- 4 Hong, J. *et al.* Metal-oxygen decoordination stabilizes anion redox in Li-rich oxides. *Nat Mater* **18**, 256-265, doi:10.1038/s41563-018-0276-1 (2019).
- 5 Yu, Y. *et al.* Revealing Electronic Signatures of Lattice Oxygen Redox in Lithium Ruthenates and Implications for High-Energy Li-Ion Battery Material Designs. *Chemistry of Materials* **31**, 7864-7876, doi:10.1021/acs.chemmater.9b01821 (2019).
- 6 Kobayashi, H. *et al.* STRUCTURE AND LITHIUM DEINTERCALATION OF Li₂-XRuO₃. *Solid State Ionics* **82**, 25-31, doi:10.1016/0167-2738(95)00135-s (1995).
- 7 James, A. C. W. P. & Goodenough, J. B. Structure and bonding in lithium ruthenate, Li₂RuO₃. *Journal of Solid State Chemistry* **74**, 287-294, doi:[https://doi.org/10.1016/0022-4596\(88\)90357-X](https://doi.org/10.1016/0022-4596(88)90357-X) (1988).
- 8 Kobayashi, H., Kanno, R., Kawamoto, Y., Tabuchi, M. & Nakamura, O. Physical properties of the de-lithiated Li₂-xRuO₃ with the layered structure. *Solid State Ionics* **86-8**, 859-863, doi:10.1016/0167-2738(96)00194-4 (1996).
- 9 Sarkar, S., Mahale, P. & Mitra, S. Lithium Rich Composition of Li₂RuO₃ and Li₂Ru_{1-x}Ir_xO₃ Layered Materials as Li-Ion Battery Cathode. *Journal of the Electrochemical Society* **161**, A934-A942, doi:10.1149/2.030406jes (2014).

- 10 Yao, Y. *et al.* High capacity and rate capability of a layered Li₂RuO₃ cathode utilized in hybrid Na⁺/Li⁺ batteries. *Journal of Materials Chemistry A* **3**, 18273-18278, doi:10.1039/c5ta03632a (2015).
- 11 Mori, D. *et al.* XRD and XAFS study on structure and cation valence state of layered ruthenium oxide electrodes, Li₂RuO₃ and Li₂Mn_{0.4}Ru_{0.6}O₃, upon electrochemical cycling. *Solid State Ionics* **285**, 66-74, doi:10.1016/j.ssi.2015.09.025 (2016).
- 12 Miura, Y., Yasui, Y., Sato, M., Igawa, N. & Kakurai, K. New-Type Phase Transition of Li₂RuO₃ with Honeycomb Structure. *Journal of the Physical Society of Japan* **76**, 033705, doi:10.1143/jpsj.76.033705 (2007).
- 13 Miura, Y., Sato, M., Yamakawa, Y., Habaguchi, T. & Ōno, Y. Structural Transition of Li₂RuO₃ Induced by Molecular-Orbit Formation. *Journal of the Physical Society of Japan* **78**, 094706, doi:10.1143/jpsj.78.094706 (2009).
- 14 Wang, J. C. *et al.* Lattice-tuned magnetism of Ru⁴⁺(4d⁴) ions in single crystals of the layered honeycomb ruthenates Li₂RuO₃ and Na₂RuO₃. *Physical Review B* **90**, doi:10.1103/PhysRevB.90.161110 (2014).
- 15 Jimenez-Segura, M.-P. *et al.* Effect of delithiation on the dimer transition of the honeycomb-lattice ruthenate Li_{2-x}RuO₃. *Physical Review B* **94**, doi:10.1103/PhysRevB.94.115163 (2016).
- 16 Jimenez-Segura, M.-P., Ikeda, A., Yonezawa, S. & Maeno, Y. Effect of disorder on the dimer transition of the honeycomb-lattice compound Li₂RuO₃. *Physical Review B* **93**, doi:10.1103/PhysRevB.93.075133 (2016).
- 17 Park, M.-S. *et al.* Li₂RuO₃ as an Additive for High-Energy Lithium-Ion Capacitors. *The Journal of Physical Chemistry C* **117**, 11471-11478, doi:10.1021/jp4005828 (2013).
- 18 Knight, J. C., Nandakumar, P., Kan, W. H. & Manthiram, A. Effect of Ru substitution on the first charge-discharge cycle of lithium-rich layered oxides. *Journal of Materials Chemistry A* **3**, 2006-2011, doi:10.1039/c4ta05178e (2015).
- 19 McCalla, E. *et al.* Visualization of O-O peroxo-like dimers in high-capacity layered oxides for Li-ion batteries. *Science* **350**, 1516-1521, doi:10.1126/science.aac8260 (2015).
- 20 Luo, K. *et al.* Charge-compensation in 3d-transition-metal-oxide intercalation cathodes

- through the generation of localized electron holes on oxygen. *Nat Chem* **8**, 684-691, doi:10.1038/nchem.2471 (2016).
- 21 Luo, K. *et al.* Anion Redox Chemistry in the Cobalt Free 3d Transition Metal Oxide Intercalation Electrode $\text{Li}[\text{Li}_{0.2}\text{Ni}_{0.2}\text{Mn}_{0.6}]\text{O}_2$. *J Am Chem Soc* **138**, 11211-11218, doi:10.1021/jacs.6b05111 (2016).
- 22 Li, X. *et al.* Direct Visualization of the Reversible $\text{O}(2^-)/\text{O}(-)$ Redox Process in Li-Rich Cathode Materials. *Adv Mater* **30**, e1705197, doi:10.1002/adma.201705197 (2018).
- 23 Armstrong, A. R. *et al.* Demonstrating oxygen loss and associated structural reorganization in the lithium battery cathode $\text{LiNi}_{0.2}\text{Li}_{0.2}\text{Mn}_{0.6}\text{O}_2$. *Journal of the American Chemical Society* **128**, 8694-8698, doi:10.1021/ja062027+ (2006).
- 24 Paulsen, J. M. Layered Li-Mn-Oxide with the O_2 Structure: A Cathode Material for Li-Ion Cells Which Does Not Convert to Spinel. *Journal of The Electrochemical Society* **146**, 3560, doi:10.1149/1.1392514 (1999).
- 25 Gueguen, A. *et al.* Decomposition of LiPF_6 in High Energy Lithium-Ion Batteries Studied with Online Electrochemical Mass Spectrometry. *Journal of the Electrochemical Society* **163**, A1095-A1100, doi:10.1149/2.0981606jes (2016).
- 26 Renfrew, S. E. & McCloskey, B. D. Quantification of Surface Oxygen Depletion and Solid Carbonate Evolution on the First Cycle of $\text{LiNi}_{0.6}\text{Mn}_{0.2}\text{Co}_{0.2}\text{O}_2$ Electrodes. *ACS Applied Energy Materials* **2**, 3762-3772, doi:10.1021/acsaem.9b00459 (2019).
- 27 Trnka, T. M. & Parkin, G. A survey of terminal chalcogenido complexes of the transition metals: Trends in their distribution and the variation of their $\text{M}=\text{E}$ bond lengths. *Polyhedron* **16**, 1031-1045, doi:10.1016/s0277-5387(96)00411-1 (1997).
- 28 Shannon, R. Revised Effective Ionic Radii and Systematic Study of Inter Atomic Distances in Halides and Chalcogenides. *Acta Crystallographica Section A* **32**, 751-767, doi:10.1107/s0567739476001551 (1976).
- 29 Shannon, R. D. & Prewitt, C. T. Effective ionic radii in oxides and fluorides. *Acta Crystallographica Section B* **25**, 925-946, doi:10.1107/s0567740869003220 (1969).
- 30 Pyykko, P. & Atsumi, M. Molecular single-bond covalent radii for elements 1-118. *Chemistry* **15**, 186-197, doi:10.1002/chem.200800987 (2009).
- 31 Pyykko, P. Additive covalent radii for single-, double-, and triple-bonded molecules and

- tetrahedrally bonded crystals: a summary. *J Phys Chem A* **119**, 2326-2337, doi:10.1021/jp5065819 (2015).
- 32 Mak, T. C. W., Che, C.-M. & Wong, K.-Y. High-valent dioxo-ruthenium(VI) complexes of macrocyclic tetradentate tertiary amines: X-ray crystal structures of trans-[Ru(15-tmc)O₂](ClO₄)₂(15-tmc = 1,4,8,12-tetramethyl-1,4,8,12-tetra-azacyclopentadecane) and trans-[Ru(16-tmc)O₂](ClO₄)₂(16-tmc = 1,5,9,13-tetramethyl-1,5,9,13-tetra-azacyclohexadecane). *Journal of the Chemical Society, Chemical Communications*, 986-988, doi:10.1039/C39850000986 (1985).
- 33 El-Hendawy, A. M., Griffith, W. P., Piggott, B. & Williams, D. J. Studies on transition-metal oxo and nitrido complexes. Part 9. Periodato and tellurato oxo-ruthenium complexes as organic oxidants. X-Ray crystal structure of trans-NaK₅[RuO₂(HIO₆)₂]·8H₂O. *Journal of the Chemical Society, Dalton Transactions*, 1983-1988, doi:10.1039/DT9880001983 (1988).
- 34 Li, B., Yan, H., Zuo, Y. & Xia, D. Tuning the Reversibility of Oxygen Redox in Lithium-Rich Layered Oxides. *Chemistry of Materials* **29**, 2811-2818, doi:10.1021/acs.chemmater.6b04743 (2017).
- 35 Sathiya, M. *et al.* Reversible anionic redox chemistry in high-capacity layered-oxide electrodes. *Nat Mater* **12**, 827-835, doi:10.1038/nmat3699 (2013).
- 36 Li, B. *et al.* Understanding the Stability for Li-Rich Layered Oxide Li₂RuO₃ Cathode. *Advanced Functional Materials* **26**, 1330-1337, doi:10.1002/adfm.201504836 (2016).

Reviewers' Comments:

Reviewer #1:

Remarks to the Author:

In the revised manuscript the authors have carefully and satisfactorily addressed all of the issues that I raised in the previous draft and have put in an extensive amount of additional work to answer outstanding scientific questions. The updated version of the manuscript provides compelling evidence for the importance of cation disorder on the nature of oxygen redox, which should be of significant interest for the wider battery community. It is my pleasure to recommend this paper for publication in Nature Communications as is.

Reviewer #2:

Remarks to the Author:

All figures have been clearly presented with full consideration. No more question, it can be published as is.

Reviewer #3:

Remarks to the Author:

The authors revised the manuscript appropriately according to the comments from reviewers. I'm still against drawing a bond between Ru and O in a long distance of 3 Å. However, the manuscript is now suitable for publication in Nat. Commun.

Responses to Referees' Comments

Manuscript title: Inhibition of oxygen dimerization by local symmetry tuning in Li-rich layered oxides for improved stability

Manuscript number: NCOMMS-20-10074B

Corresponding author name(s): Dingguo Xia

Reply to Reviewer 1

General Comment

In the revised manuscript the authors have carefully and satisfactorily addressed all of the issues that I raised in the previous draft and have put in an extensive amount of additional work to answer outstanding scientific questions. The updated version of the manuscript provides compelling evidence for the importance of cation disorder on the nature of oxygen redox, which should be of significant interest for the wider battery community. It is my pleasure to recommend this paper for publication in Nature Communications as is.

Response to the General Comment

We are grateful for the positive evaluation of our work. We thank the referee for the contribution to our work.

Reply to Reviewer 2

General Comment

All figures have been clearly presented with full consideration. No more question, it can be published as is.

Response to the General Comment

We thank the referee for the time and expertise. And greatly appreciate the referee for the positive evaluation of our work.

Reply to Reviewer 3

General Comment

The authors revised the manuscript appropriately according to the comments from reviewers. I'm still against drawing a bond between Ru and O in a long distance of 3 Å. However, the manuscript is now suitable for publication in Nat. Commun.

Response to the General Comment

Thanks for the referee's positive evaluation of our work. The bond drawn between Ru and O in a distance of 3 Å was removed. We thank the referee for the contribution in improving our work.